# Provable Guarantees for Nonlinear Feature Learning in Three-Layer Neural Networks

**Eshaan Nichani**
Princeton University
eshnich@princeton.edu

**Alex Damian**
Princeton University
ad27@princeton.edu

**Jason D. Lee**
Princeton University
jasonlee@princeton.edu

## Abstract

One of the central questions in the theory of deep learning is to understand how neural networks learn hierarchical features. The ability of deep networks to extract salient features is crucial to both their outstanding generalization ability and the modern deep learning paradigm of pretraining and finetuneing. However, this feature learning process remains poorly understood from a theoretical perspective, with existing analyses largely restricted to two-layer networks. In this work we show that three-layer neural networks have provably richer feature learning capabilities than two-layer networks. We analyze the features learned by a three-layer network trained with layer-wise gradient descent, and present a general purpose theorem which upper bounds the sample complexity and width needed to achieve low test error when the target has specific hierarchical structure. We instantiate our framework in specific statistical learning settings – single-index models and functions of quadratic features – and show that in the latter setting three-layer networks obtain a sample complexity improvement over all existing guarantees for two-layer networks. Crucially, this sample complexity improvement relies on the ability of three-layer networks to efficiently learn *nonlinear* features. We then establish a concrete optimization-based depth separation by constructing a function which is efficiently learnable via gradient descent on a three-layer network, yet cannot be learned efficiently by a two-layer network. Our work makes progress towards understanding the provable benefit of three-layer neural networks over two-layer networks in the feature learning regime.

## 1 Introduction

The success of modern deep learning can largely be attributed to the ability of deep neural networks to decompose the target function into a hierarchy of learned features. This feature learning process enables both improved accuracy [29] and transfer learning [21]. Despite its importance, we still have a rudimentary theoretical understanding of the feature learning process. Fundamental questions include understanding what features are learned, how they are learned, and how they affect generalization.

From a theoretical viewpoint, a fascinating question is to understand how depth can be leveraged to learn more salient features and as a consequence a richer class of hierarchical functions. The base case for this question is to understand which features (and function classes) can be learned efficiently by three-layer neural networks, but not two-layer networks. Recent work on feature learning has shown that two-layer neural networks learn features which are *linear* functions of the input (see Section 1.2 for further discussion). It is thus a natural question to understand if three-layer networks can learn *nonlinear* features, and how this can be leveraged to obtain a sample complexity improvement. Initial learning guarantees for three-layer networks [16, 5, 4], however, consider simplified model and function classes and do not discern specifically what the learned features are or whether a sample complexity improvement can be obtained over shallower networks or kernel methods.

37th Conference on Neural Information Processing Systems (NeurIPS 2023).

On the other hand, the standard approach in deep learning theory to understand the benefit of depth has been to establish "depth separations" [51], i.e. functions that cannot be efficiently approximated by shallow networks, but can be via deeper networks. However, depth separations are solely concerned with the representational capability of neural networks, and ignore the optimization and generalization aspects. In fact, depth separation functions such as [51] are often not learnable via gradient descent [36]. To reconcile this, recent papers [45, 44] have established *optimization-based* depth separation results, which are functions which cannot be efficiently learned using gradient descent on a shallow network but can be learned with a deeper network. We thus aim to answer the following question:

*What features are learned by gradient descent on a three-layer neural network, and can these features be leveraged to obtain a provable sample complexity guarantee?*

## 1.1 Our contributions

We provide theoretical evidence that three-layer neural networks have provably richer feature learning capabilities than their two-layer counterparts. We specifically study the features learned by a three-layer network trained with a layer-wise variant of gradient descent (Algorithm 1). Our main contributions are as follows.

- Theorem 1 is a general purpose sample complexity guarantee for Algorithm 1 to learn an arbitrary target function $f^*$. We first show that Algorithm 1 learns a feature roughly corresponding to a low-frequency component of the target function $f^*$ with respect to the random feature kernel $\mathbb{K}$ induced by the first layer. We then derive an upper bound on the population loss in terms of the learned feature. As a consequence, we show that if $f^*$ possesses a hierarchical structure where it can be written as a 1D function of the learned feature (detailed in Section 3), then the sample complexity for learning $f^*$ is equal to the sample complexity of learning the feature. This demonstrates that three-layer networks indeed perform hierarchical learning.

- We next instantiate Theorem 1 in two statistical learning settings which satisfy such hierarchical structure. As a warmup, we show that Algorithm 1 learns single-index models (i.e $f^*(x) = g^*(w \cdot x)$) in $d^2$ samples, which is comparable to existing guarantees for two-layer networks and crucially has $d$-dependence not scaling with the degree of the link function $g^*$. We next show that Algorithm 1 learns the target $f^*(x) = g^*(x^T A x)$, where $g^*$ is either Lipschitz or a degree $p = O(1)$ polynomial, up to $o_d(1)$ error with $d^4$ samples. This improves on all existing guarantees for learning with two-layer networks or via NTK-based approaches, which all require sample complexity $d^{\Omega(p)}$. A key technical step is to show that for the target $f^*(x) = g^*(x^T A x)$, the learned feature is approximately $x^T A x$. This argument relies on the universality principle in high-dimensional probability, and may be of independent interest.

- We conclude by establishing an explicit optimization-based depth separation: We show that the target function $f^*(x) = \text{ReLU}(x^T A x)$ for appropriately chosen $A$ can be learned by Algorithm 1 up to $o_d(1)$ error in $d^4$ samples, whereas any two layer network needs either superpolynomial width or weight norm in order to approximate $f^*$ up to comparable accuracy. This implies that such an $f^*$ is not efficiently learnable via two-layer networks.

The above separation hinges on the ability of three-layer networks to learn the nonlinear feature $x^T A x$ and leverage this feature learning to obtain an improved sample complexity. Altogether, our work presents a general framework demonstrating the capability of three-layer networks to learn nonlinear features, and makes progress towards a rigorous understanding of feature learning, optimization-based depth separations, and the role of depth in deep learning more generally.

## 1.2 Related Work

**Neural Networks and Kernel Methods.** Early guarantees for neural networks relied on the Neural Tangent Kernel (NTK) theory [31, 50, 23, 17]. The NTK theory shows global convergence by coupling to a kernel regression problem and generalization via the application of kernel generalization bounds [7, 15, 5]. The NTK can be characterized explicitly for certain data distributions [27, 39, 38], which allows for tight sample complexity and width analyses. This connection to kernel methods has also been used to study the role of depth, by analyzing the signal propagation and evolution of

the NTK in MLPs [42, 48, 28], convolutional networks [6, 55, 56, 40], and residual networks [30]. However, the NTK theory is insufficient as neural networks outperform their NTK in practice [6, 32]. In fact, [27] shows that kernels cannot adapt to low-dimensional structure and require $d^k$ samples to learn any degree $k$ polynomials in $d$ dimensions. Ultimately, the NTK theory fails to explain generalization or the role of depth in practical networks not in the kernel regime. A key distinction is that networks in the kernel regime cannot learn features [57]. A recent goal has thus been to understand the feature learning mechanism and how this leads to sample complexity improvements [53, 22, 58, 3, 25, 26, 20, 54, 33, 35]. Crucially, our analysis is *not* in the kernel regime, and shows an improvement of three-layer networks over two-layer networks in the feature-learning regime.

**Feature Learning.** Recent work has studied the provable feature learning capabilities of two-layer neural networks. [9, 1, 2, 8, 13, 10] show that for isotropic data distributions, two-layer networks learn linear features of the data, and thus efficiently learn functions of low-dimensional projections of the input (i.e targets of the form $f^*(x) = g(Ux)$ for $U \in \mathbb{R}^{r \times d}$). Here, $x \mapsto Ux$ is the "linear feature." Such target functions include low-rank polynomials [18, 2] and single-index models [8, 13] for Gaussian covariates, as well as sparse boolean functions [1] such as the $k$-sparse parity problem [10] for covariates uniform on the hypercube. [43] draws connections from the mechanisms in these works to feature learning in standard image classification settings. The above approaches rely on layerwise training procedures, and our Algorithm 1 is an adaptation of the algorithm in [18].

Another approach uses the quadratic Taylor expansion of the network to learn classes of polynomials [9, 41] This approach can be extended to three-layer networks. [16] replace the outermost layer with its quadratic approximation, and by viewing $z^p$ as the hierarchical function $(z^{p/2})^2$ show that their three-layer network can learn low rank, degree $p$ polynomials in $d^{p/2}$ samples. [5] similarly uses a quadratic approximation to improperly learn a class of three-layer networks via sign-randomized GD. An instantiation of their upper bound to the target $g^*(x^T A x)$ for degree $p$ polynomial $g^*$ yields a sample complexity of $d^{p+1}$. However, [16, 5] are proved via opaque landscape analyses, do not concretely identify the learned features, and rely on nonstandard algorithmic modifications. Our Theorem 1 directly identifies the learned features, and when applied to the quadratic feature setting in Section 4.2 obtains an improved sample complexity guarantee independent of the degree of $g^*$.

**Depth Separations.** [51] constructs a function which can be approximated by a poly-width network with large depth, but not with smaller depth. [24] is the first depth separation between depth 2 and 3 networks, with later works [46, 19, 47] constructing additional such examples. However, such functions are often not learnable via three-layer networks [34]. [36] shows that approximatability by a shallow (depth 3 network) is a necessary condition for learnability via a deeper network.

These issues have motivated the development of optimization-based, or algorithmic, depth separations, which construct functions which are learnable by a three-layer network but not by two-layer networks. [45] shows that certain ball indicator functions $\mathbf{1}(\|x\| \geq \lambda)$ are not approximatable by two-layer networks, yet are learnable via GD on a special variant of a three-layer network with second layer width equal to 1. However, their network architecture is tailored for learning the ball indicator, and the explicit polynomial sample complexity ($n \gtrsim d^{36}$) is weak. [44] shows that a multi-layer mean-field network with a 1D bottleneck layer can learn the target $\text{ReLU}(1 - \|x\|)$, which [47] previously showed was inaproximatable via two-layer networks. However, their analysis relies on the rotational invariance of the target function, and it is difficult to read off explicit sample complexity and width guarantees beyond being $\text{poly}(d)$. Our Section 4.2 shows that three-layer networks can learn a larger class of features ($x^T A x$ versus $\|x\|$) and functions on top of these features (any Lipschitz $q$ versus ReLU), with explicit dependence on the width and sample complexity needed ($n, m_1, m_2 = \tilde{O}(d^4)$).

## 2 Preliminaries

### 2.1 Problem Setup

**Data distribution.** Our aim is to learn the target function $f^* : \mathcal{X}_d \to \mathbb{R}$, with $\mathcal{X}_d \subset \mathbb{R}^d$ the space of covariates. We let $\nu$ be some distribution on $\mathcal{X}_d$, and draw two independent datasets $\mathcal{D}_1, \mathcal{D}_2$, each with $n$ samples, so that each $x \in \mathcal{D}_1$ or $x \in \mathcal{D}_2$ is sampled i.i.d as $x \sim \nu$. Without loss of generality, we normalize so $\mathbb{E}_{x \sim \nu}[f^*(x)^2] \leq 1$. We make the following assumptions on $\nu$:

---

**Algorithm 1** Layer-wise training algorithm

---

**Input:** Initialization $\theta^{(0)}$; learning rates $\eta_1, \eta_2$; weight decay $\lambda$; time $T$
  {Stage 1: Train $W$}
  $W^{(1)} \leftarrow W^{(0)} - \eta_1 \nabla_W L_1(\theta^{(0)})$
  $a^{(1)} \leftarrow a^{(0)}$
  $\theta^{(1)} \leftarrow (a^{(1)}, W^{(1)}, b^{(0)}, V^{(0)})$
  {Stage 2: Train $a$}
  **for** $t = 2, \cdots, T$ **do**
    $a^{(t)} \leftarrow a^{(t-1)} - \eta_2 \big[ \nabla_a L_2(\theta^{(t-1)}) + \lambda a^{(t-1)} \big]$
    $\theta^{(t)} \leftarrow (a^{(t)}, W^{(1)}, b^{(0)}, V^{(0)})$
  **end for**
  $\hat{\theta} \leftarrow \theta^{(T)}$
**Output:** $\hat{\theta}$

---

**Definition 1** (Sub-Gaussian Vector). *A mean-zero random vector $X \in \mathbb{R}^d$ is $\gamma$-subGaussian if, for all unit vectors $v \in \mathbb{R}^d$, $\mathbb{E}[\exp(\lambda X \cdot v)] \leq \exp(\gamma^2 \lambda^2)$ for all $\lambda \in \mathbb{R}$.*

**Assumption 1.** $\mathbb{E}_{x \sim \nu}[x] = 0$ *and $\nu$ is $C_\gamma$-subGaussian for some constant $C_\gamma$.*

**Assumption 2.** $f^*$ *has polynomially growing moments, i.e there exist constants $(C_f, \ell)$ such that $\mathbb{E}_{x \sim \nu}[f^*(x)^q]^{1/q} \leq C_f q^\ell$ for all $q \geq 1$.*

We note that Assumption 2 is satisfied by a number of common distributions and functions, and we will verify that Assumption 2 holds for each example in Section 4.

**Three-layer neural network.** Let $m_1, m_2$ be the two hidden layer widths, and $\sigma_1, \sigma_2$ be two activation functions. Our learner is a three-layer neural network parameterized by $\theta = (a, W, b, V)$, where $a \in \mathbb{R}^{m_1}, W \in \mathbb{R}^{m_1 \times m_2}, b \in \mathbb{R}^{m_1}$, and $V \in \mathbb{R}^{m_2 \times d}$. The network $f(x; \theta)$ is defined as:

$$f(x; \theta) := \frac{1}{m_1} a^T \sigma_1(W \sigma_2(Vx) + b) = \frac{1}{m_1} \sum_{i=1}^{m_1} a_i \sigma_1 \Big( \langle w_i, h^{(0)}(x) \rangle + b_i \Big). \tag{1}$$

Here, $w_i \in \mathbb{R}^{m_2}$ is the $i$th row of $W$, and $h^{(0)}(x) := \sigma_2(Vx) \in \mathbb{R}^{m_2}$ is the random feature embedding arising from the innermost layer. The parameter vector $\theta^{(0)} := (a^{(0)}, W^{(0)}, b^{(0)}, V^{(0)})$ is initialized with $a_i^{(0)} \sim_{iid} \text{Unif}(\{\pm 1\})$, $W^{(0)} = 0$, the biases $b_i^{(0)} \sim_{iid} \mathcal{N}(0, 1)$, and the rows $v_i^{(0)}$ of $V^{(0)}$ drawn $v_i \sim_{iid} \tau$, where $\tau$ is the uniform measure on $\mathcal{S}^{d-1}(1)$, the $d$-dimensional unit sphere. We make the following assumption on the activations, and note that the polynomial growth assumption on $\sigma_2$ is satisfied by all activations used in practice.

**Assumption 3.** $\sigma_1$ *is the* ReLU *activation, i.e $\sigma_1(z) = \max(z, 0)$, and $\sigma_2$ has polynomial growth, i.e $|\sigma_2(x)| \leq C_\sigma (1 + |x|)^{\alpha_\sigma}$ for some constants $C_\sigma, \alpha_\sigma > 0$.*

**Training Algorithm.** Let $L_i(\theta)$ denote the empirical loss on dataset $\mathcal{D}_i$; that is for $i = 1, 2$: $L_i(\theta) := \frac{1}{n} \sum_{x \in \mathcal{D}_i} (f(x; \theta) - f^*(x))^2$. Our network is trained via layer-wise gradient descent with sample splitting. Throughout training, the first layer weights $V$ and second layer bias $b$ are held constant. First, the second layer weights $W$ are trained for $t = 1$ timesteps. Next, the outer layer weights $a$ are trained for $t = T - 1$ timesteps. This two stage training process is common in prior works analyzing gradient descent on two-layer networks [18, 8, 1, 10], and as we see in Section 5, is already sufficient to establish a separation between two and three-layer networks. Pseudocode for the training procedure is presented in Algorithm 1.

## 2.2 Technical definitions

The activation $\sigma_2$ admits a random feature kernel $K : \mathcal{X}_d \times \mathcal{X}_d \to \mathbb{R}$ and corresponding integral operator $\mathbb{K} : L^2(\mathcal{X}_d, \nu) \to L^2(\mathcal{X}_d, \nu)$:

**Definition 2** (Kernel objects). $\sigma_2$ *admits the random feature kernel*

$$K(x, x') := \mathbb{E}_{v \sim \tau}[\sigma_2(x \cdot v) \sigma_2(x' \cdot v)] \tag{2}$$

*and corresponding integral operator*

$$(\mathbb{K}f)(x) := \mathbb{E}_{x' \sim \nu}[K(x, x')f(x')]. \tag{3}$$

We make the following assumption on $\mathbb{K}$, which we verify for the examples in Section 4:

**Assumption 4.** $\mathbb{K}f^*$ *has polynomially bounded moments, i.e there exist constants $C_K, \chi$ such that, for all $1 \leq q \leq d$, $\|\mathbb{K}f^*\|_{L^q(\nu)} \leq C_K q^\chi \|\mathbb{K}f^*\|_{L^2(\nu)}$.*

We also require the definition of the Sobolev space:

**Definition 3.** *Let $\mathcal{W}^{2,\infty}([-1,1])$ be the Sobolev space of twice continuously differentiable functions $q : [-1, 1] \to \mathbb{R}$ equipped with the norm $\|q\|_{k,\infty} := \max_{s \leq k} \max_{x \in [-1,1]} |q^{(s)}(x)|$ for $k = 1, 2$.*

### 2.3 Notation

We use big $O$ notation (i.e $O, \Theta, \Omega$) to ignore absolute constants ($C_\sigma, C_f$, etc.) that do not depend on $d, n, m_1, m_2$. We further write $a_d \lesssim b_d$ if $a_d = O(b_d)$, and $a_d = o(b_d)$ if $\lim_{d \to \infty} a_d/b_d = 0$. Additionally, we use $\tilde{O}$ notation to ignore terms that depend logarithmically on $dnm_1m_2$. For $f : \mathcal{X}_d \to \mathbb{R}$, define $\|f\|_{L^p(\mathcal{X}_d,\nu)} = (\mathbb{E}_{x \sim \nu}[f(x)^p])^{1/p}$. To simplify notation we also call this quantity $\|f\|_{L^p(\nu)}$, and $\|g\|_{L^p(\mathcal{X}_d,\nu)}, \|g\|_{L^p(\tau)}$ are defined analogously for functions $g : \mathcal{S}^{d-1}(1) \to \mathbb{R}$. When the domain is clear from context, we write $\|f\|_{L^p}, \|g\|_{L^p}$. We let $L^p(\mathcal{X}_d, \nu)$ be the space of $f$ with finite $\|f\|_{L^p(\mathcal{X}_d,\nu)}$. Finally, we write $\mathbb{E}_x$ and $\mathbb{E}_v$ as shorthand for $\mathbb{E}_{x \sim \nu}$ and $\mathbb{E}_{v \sim \tau}$ respectively.

## 3 Main Result

The following is our main theorem which upper bounds the population loss of Algorithm 1:

**Theorem 1.** *Select $q \in \mathcal{W}^{2,\infty}([-1,1])$. Let $\eta_1 = \frac{m_1}{m_2}\overline{\eta}$, and assume $n, m_1, m_2 = \tilde{\Omega}(\|\mathbb{K}f^*\|_{L^2}^{-2})$ There exist $\overline{\eta}, \lambda, \eta_2$ such that after $T = \mathrm{poly}(n, m_1, m_2, d, \|q\|_{2,\infty})$ timesteps, with high probability over the initialization and datasets the output $\hat{\theta}$ of Algorithm 1 satisfies the population $L^2$ loss bound*

$$\mathbb{E}_x\left[\left(f(x; \hat{\theta}) - f^*(x)\right)^2\right]$$

$$\leq \tilde{O}\left(\underbrace{\|q \circ (\overline{\eta} \cdot \mathbb{K}f^*) - f^*\|_{L^2}^2}_{\substack{\text{accuracy of} \\ \text{feature learning}}} + \underbrace{\frac{\|q\|_{1,\infty}^2 \|\mathbb{K}f^*\|_{L^2}^{-2}}{\min(n, m_1, m_2)}}_{\substack{\text{sample complexity of} \\ \text{feature learning}}} + \underbrace{\frac{\|q\|_{2,\infty}^2}{m_1} + \frac{\|q\|_{2,\infty}^2 + 1}{\sqrt{n}}}_{\text{complexity of } q}\right) \tag{4}$$

The full proof of this theorem is in Appendix D. The population risk upper bound (4) has three terms:

1. The first term quantifies the extent to which feature learning is useful for learning the target $f^*$, and depends on how close $f^*$ is to having hierarchical structure. Concretely, if there exists $q : \mathbb{R} \to \mathbb{R}$ such that the compositional function $q \circ \overline{\eta} \cdot \mathbb{K}f^*$ is close to the target $f^*$, then this first term is small. In Section 4, we show that this is true for certain *hierarchical* functions. In particular, say that $f^*$ satisfies the hierarchical structure $f^* = g^* \circ h^*$. If the quantity $\mathbb{K}f^*$ is nearly proportional to the true feature $h^*$, then this first term is negligible. As such, we refer to the quantity $\mathbb{K}f^*$ as the *learned feature*.

2. The second term is the sample (and width) complexity of learning the feature $\mathbb{K}f^*$. It is useful to compare the $\|\mathbb{K}f^*\|_{L^2}^{-2}$ term to the standard kernel generalization bound, which requires $n \gtrsim \langle f^\star, \mathbb{K}^{-1}f^\star \rangle_{L^2}$. Unlike in the kernel bound, the feature learning term in (4) does not require inverting the kernel $\mathbb{K}$ as it only requires a lower bound on $\|\mathbb{K}f^\star\|_{L^2}$. This difference can be best understood by considering the alignment of $f^\star$ with the eigenfunctions of $\mathbb{K}$. Say that $f^\star$ has nontrivial alignment with eigenfunctions of $\mathbb{K}$ with both small ($\lambda_{min}$) and large ($\lambda_{max}$) eigenvalues. Kernel methods require $\Omega(\lambda_{min}^{-1})$ samples, which blows up when $\lambda_{min}$ is small; the sample complexity of kernel methods depends on the high frequency components of $f^*$. On the other hand, the guarantee in Theorem 1 scales with

$\|\mathbb{K}f^\star\|_{L^2}^{-2} = O(\lambda_{max}^{-2})$, which can be much smaller. In other words, the sample complexity of feature learning scales with the low-frequency components of $f^*$. The feature learning process can thus be viewed as extracting the low-frequency components of the target.

3. The last two terms measure the complexity of learning the univariate function $q$. In the examples in Section 4, the effect of these terms is benign.

Altogether, if $f^*$ satisfies the hierarchical structure that its high-frequency components can be inferred from the low-frequency ones, then a good $q$ for Theorem 1 exists and the dominant term in (4) is the sample complexity of feature learning term, which only depends on the low-frequency components. This is not the case for kernel methods, as small eigenvalues blow up the sample complexity. As we show in Section 4, this ability to ignore the small eigenvalue components of $f^\star$ during the feature learning process is critical for achieving good sample complexity in many problems.

## 3.1 Proof Sketch

At initialization, $f(x; \theta^{(0)}) \approx 0$. The first step of GD on the population loss for a neuron $w_j$ is thus

$$w^{(1)} = -\eta_1 \nabla_{w_j} \mathbb{E}_x \left[ \left( f(x; \theta^{(0)}) - f^*(x) \right)^2 \right] \tag{5}$$

$$= \eta_1 \mathbb{E}_x \left[ f^*(x) \nabla_{w_j} f(x; \theta^{(0)}) \right] \tag{6}$$

$$= \frac{1}{m_2} \overline{\eta} \mathbf{1}_{b_j^{(0)} \geq 0} a_j^{(0)} \mathbb{E}_x \left[ f^*(x) h^{(0)}(x) \right]. \tag{7}$$

Therefore the network $f(x'; \theta^{(1)})$ after the first step of GD is given by

$$f(x'; \theta^{(1)}) = \frac{1}{m_1} \sum_{j=1}^{m_1} a_j \sigma_1 \left( \langle w_j^{(1)}, h^{(0)}(x') \rangle + b_j \right) \tag{8}$$

$$= \frac{1}{m_1} \sum_{j=1}^{m_1} a_j \sigma_1 \left( a_j^{(0)} \cdot \overline{\eta} \frac{1}{m_2} \mathbb{E}_x \left[ f^*(x) h^{(0)}(x)^T h^{(0)}(x') \right] + b_j \right) \mathbf{1}_{b_j^{(0)} \geq 0}. \tag{9}$$

We first notice that this network now implements a 1D function of the quantity

$$\phi(x') := \overline{\eta} \frac{1}{m_2} \mathbb{E}_x \left[ f^*(x) h^{(0)}(x)^T h^{(0)}(x') \right]. \tag{10}$$

Specifically, the network can be rewritten as

$$f(x'; \theta^{(1)}) = \frac{1}{m_1} \sum_{j=1}^{m_1} a_j \sigma_1 \left( a_j^{(0)} \cdot \phi(x') + b_j \right) \cdot \mathbf{1}_{b_j^{(0)} \geq 0}. \tag{11}$$

Since $f$ implements a hierarchical function of the quantity $\phi(x)$, we term $\phi$ the *learned feature*.

The second stage of Algorithm 1 is equivalent to random feature regression. We first use results on ReLU random features to show that any $q \in \mathcal{W}^{2,\infty}([-1,1])$ can be approximated on $[-1,1]$ as $q(z) \approx \frac{1}{m_1} \sum_{j=1}^{m_1} a_j^* \sigma_1(a_j^{(0)} z + b_j)$ for some $\|a^*\| \lesssim \|q\|_{2,\infty}$ (Lemma 3). Next, we use the standard kernel Rademacher bound to show that the excess risk scales with the smoothness of $q$ (Lemma 5). Hence we can efficiently learn functions of the form $q \circ \phi$.

It suffices to compute this learned feature $\phi$. For $m_2$ large, we observe that

$$\phi(x') = \frac{\overline{\eta}}{m_2} \sum_{j=1}^{m_2} \mathbb{E}_x[f^*(x) \sigma(x \cdot v) \sigma(x' \cdot v)]) \approx \overline{\eta} \mathbb{E}_x[f^*(x) K(x, x')] = \overline{\eta}(\mathbb{K}f^*)(x'). \tag{12}$$

The learned feature is thus approximately $\overline{\eta} \cdot \mathbb{K}f^*$. Choosing $\overline{\eta}$ so that $|\phi(x')| \leq 1$, we see that Algorithm 1 learns functions of the form $\phi \circ (\overline{\eta} \cdot \mathbb{K}f^*)$. Finally, we translate the above analysis to the finite sample gradient via standard concentration tools. Since the empirical estimate to $\mathbb{K}f^*$ concentrates at a $1/\sqrt{n}$ rate, $n \gtrsim \|\mathbb{K}f^*\|_{L^2}^{-2}$ samples are needed to obtain a constant factor approximation (Lemma 7).

# 4 Examples

We next instantiate Theorem 1 in two specific statistical learning settings which satisfy the hierarchical prescription detailed in Section 3. As a warmup, we show that three-layer networks efficiently learn single index models. Our second example shows how three-layer networks can obtain a sample complexity improvement over existing guarantees for two-layer networks.

## 4.1 Warmup: single index models

Let $f^* = g^*(w^* \cdot x)$, for unknown direction $w^* \in \mathbb{R}^d$ and unknown link function $g^* : \mathbb{R} \to \mathbb{R}$, and take $\mathcal{X}_d = \mathbb{R}^d$ with $\nu = \mathcal{N}(0, I)$. Prior work [18, 13] shows that two-layer neural networks learn such functions with an improved sample complexity over kernel methods. Let $\sigma_2(z) = z$, so that the network is of the form $f(x; \theta) = \frac{1}{m_1} a^T \sigma_1(WVx)$. We can verify that Assumptions 1 to 4 are satisfied, and thus applying Theorem 1 in this setting yields the following:

**Theorem 2.** *Let $f^*(x) = g^*(w^* \cdot x)$, where $\|w^*\|_2 = 1$. Assume that $g^*, (g^*)'$ and $(g^*)''$ are polynomially bounded and that $\mathbb{E}_{z \sim \mathcal{N}(0,1)}[(g^*)'(z)] \neq 0$. Then with high probability Algorithm 1 satisfies the population loss bound*

$$\mathbb{E}_x\left[\left(f(x; \hat{\theta}) - f^*(x)\right)^2\right] = \tilde{O}\left(\frac{d^2}{\min(n, m_1, m_2)} + \frac{1}{\sqrt{n}}\right). \tag{13}$$

Given widths $m_1, m_2 = \tilde{\Theta}(d^2)$, $n = \tilde{\Theta}(d^2)$ samples suffice to learn $f^*$, which matches existing guarantees for two-layer neural networks [13, 18]. We remark that prior work on learning single-index models under assumptions on the link function such as monotonicity or the condition $\mathbb{E}_{z \sim \mathcal{N}(0,1)}[(g^*)'(z)] \neq 0$ require $d$ samples [49, 37, 12]. However, our sample complexity improves on that of kernel methods, which require $d^p$ samples when $g^*$ is a degree $p$ polynomial.

Theorem 2 is proved in Appendix E.1; a brief sketch is as follows. Since $\sigma_2(z) = z$, the kernel is $K(x, x') = \mathbb{E}_v[(x \cdot v)(x' \cdot v)] = \frac{x \cdot x'}{d}$. By an application of Stein's Lemma, the learned feature is

$$(\mathbb{K}f^*)(x) = \frac{1}{d}\mathbb{E}_{x'}[x \cdot x' f^*(x)] = \frac{1}{d}x^T \mathbb{E}_{x'}[\nabla f^*(x')] \tag{14}$$

Since $f^*(x) = g^*(w^* \cdot x)$, $\nabla f^*(x) = w^*(g^*)'(w^* \cdot x)$, and thus

$$(\mathbb{K}f^*)(x) = \frac{1}{d}\mathbb{E}_{z \sim \mathcal{N}(0,1)}[(g^*)'(z)]w^* \cdot x \propto \frac{1}{d}w^* \cdot x. \tag{15}$$

The learned feature is proportional to the true feature, so an appropriate choice of $\overline{\eta}$ and choosing $q = g^*$ in Theorem 1 implies that $\|\mathbb{K}f^*\|_{L^2}^{-2} = d^2$ samples are needed to learn $f^*$.

## 4.2 Functions of quadratic features

The next example shows how three-layer networks can learn nonlinear features, and thus obtain a sample complexity improvement over two-layer networks.

Let $\mathcal{X}_d = \mathcal{S}^{d-1}(\sqrt{d})$, the sphere of radius $\sqrt{d}$, and $\nu$ the uniform measure on $\mathcal{X}_d$. The integral operator $\mathbb{K}$ has been well studied [39, 27, 38], and its eigenfunctions correspond to the spherical harmonics. Preliminaries on spherical harmonics and this eigendecomposition are given in Appendix F.

Consider the target $f^*(x) = g^*(x^T Ax)$, where $A \in \mathbb{R}^{d \times d}$ is a symmetric matrix and $g^* : \mathbb{R} \to \mathbb{R}$ is an unknown link function. In contrast to a single-index model, the feature $x^T Ax$ we aim to learn is a *quadratic* function of $x$. Since one can write $x^T Ax = x^T\left(A - \text{Tr}(A) \cdot \frac{I}{d}\right)x + \text{Tr}(A)$, we without loss of generality assume $\text{Tr}(A) = 0$. We also select the normalization $\|A\|_F^2 = \frac{d+2}{2d} = \Theta(1)$; this ensures that $\mathbb{E}_{x \sim \nu}[(x^T Ax)^2] = 1$. We first make the following assumptions on the target function.:

**Assumption 5.** $\mathbb{E}_x[f^*(x)] = 0$, $\mathbb{E}_{z \sim \mathcal{N}(0,1)}[(g^*)'(z)] = \Theta(1)$, $g^*$ *is 1-Lipschitz, and $(g^*)''$ has polynomial growth.*

The first assumption can be achieved via a preprocessing step which subtracts the mean of $f^*$, the second is a nondegeneracy condition, and the last two assume the target is sufficiently smooth.

We next require the eigenvalues of $A$ to satisfy an incoherence condition:

**Assumption 6.** *Define* $\kappa := \|A\|_{op} \sqrt{d}$. *Then* $\kappa = o(\sqrt{d}/polylog(d))$.

Note that $\kappa \leq \sqrt{d}$. If $A$ has rank $\Theta(d)$ and condition number $\Theta(1)$, then $\kappa = \Theta(1)$. Furthermore, when the entries of $A$ are sampled i.i.d, $\kappa = \tilde{\Theta}(1)$ with high probability by Wigner's semicircle law.

Finally, we make the following nondegeneracy assumption on the Gegenbauer decomposition of $\sigma_2$ (defined in Appendix F). We show that $\lambda_2^2(\sigma_2) = O(d^{-2})$, and later argue that the following assumption is mild and indeed satisfied by standard activations such as $\sigma_2 = \text{ReLU}$.

**Assumption 7.** *Let* $\lambda_2(\sigma_2)$ *be the 2nd Gegenbauer coefficient of* $\sigma_2$. *Then* $\lambda_2^2(\sigma_2) = \Theta(d^{-2})$.

We can verify Assumptions 1 to 4 hold for this setting, and thus applying Theorem 1 yields the following:

**Theorem 3.** *Under Assumptions 5 to 7, with high probability Algorithm 1 satisfies the population loss bound*

$$\mathbb{E}_x\left[\left(f(x;\hat{\theta}) - f^*(x)\right)^2\right] \lesssim \tilde{O}\left(\frac{d^4}{\min(n, m_1, m_2)} + \frac{1}{\sqrt{n}} + \left(\frac{\kappa}{\sqrt{d}}\right)^{1/3}\right) \tag{16}$$

We thus require sample size $n = \tilde{\Omega}(d^4)$ and widths $m_1, m_2 = \tilde{\Omega}(d^4)$ to obtain $o_d(1)$ test loss.

**Proof Sketch.** The integral operator $\mathbb{K}$ has eigenspaces corresponding to spherical harmonics of degree $k$. In particular, [27] shows that, in $L^2$,

$$\mathbb{K}f^* = \sum_{k \geq 0} c_k P_k f^*, \tag{17}$$

where $P_k$ is the orthogonal projection onto the subspace of degree $k$ spherical harmonics, and the $c_k$ are constants satisfying $c_k = O(d^{-k})$. Since $f^*$ is an even function and $\mathbb{E}_x[f^*(x)] = 0$, truncating this expansion at $k = 2$ yields

$$\mathbb{K}f^* = \Theta(d^{-2}) \cdot P_2 f^* + O(d^{-4}), \tag{18}$$

It thus suffices to compute $P_2 f^*$. To do so, we draw a connection to the universality phenomenon in high dimensional probability. Consider two features $x^T A x$ and $x^T B x$ with $\langle A, B \rangle = 0$. We show that, when $d$ is large, the distribution of $x^T A x$ approaches that of the standard Gaussian, while $x^T B x$ approaches a mixture of $\chi^2$ and Gaussian random variables independent of $x^T A x$. As such, we show

$$\mathbb{E}_x\left[g^*(x^T A x)x^T B x\right] \approx \mathbb{E}_x\left[g^*(x^T A x)\right] \cdot \mathbb{E}_x\left[x^T B x\right] = 0 \tag{19}$$

$$\mathbb{E}_x\left[g^*(x^T A x)x^T A x\right] \approx \mathbb{E}_{z \sim \mathcal{N}(0,1)}\left[g^*(z)z\right] = \mathbb{E}_{z \sim \mathcal{N}(0,1)}\left[(g^*)'(z)\right]. \tag{20}$$

The second expression can be viewed as an approximate version of Stein's lemma, which was applied in Section 4.1 to compute the learned feature. Altogether, our key technical result (stated formally in Lemma 20) is that for $f^* = g^*(x^T A x)$, the projection $P_2 f^*$ satisfies

$$(P_2 f^*)(x) = \mathbb{E}_{z \sim \mathcal{N}(0,1)}[(g^*)'(z)] \cdot x^T A x + o_d(1) \tag{21}$$

The learned feature is thus $\mathbb{K}f^*(x) = \Theta(d^{-2}) \cdot (x^T A x + o_d(1))$; plugging this into Theorem 1 yields the $d^4$ sample complexity.

The full proof of Theorem 3 is deferred to Appendix E.2. In Appendix E.3 we show that when $g^*$ is a degree $p = O(1)$ polynomial, Algorithm 1 learns $f^*$ in $\tilde{O}(d^4)$ samples with an improved error floor.

**Comparison to two-layer networks.** Existing guarantees for two-layer networks cannot efficiently learn functions of the form $f^*(x) = g^*(x^T A x)$ for arbitrary Lipschitz $g^*$. In fact, in Section 5 we provide an explicit lower bound against two-layer networks efficiently learning a subclass of these functions. When $g^*$ is a degree $p$ polynomial, networks in the kernel regime require $d^{2p} \gg d^4$ samples to learn $f^*$ [27]. Improved guarantees for two-layer networks learn degree $p'$ polynomials in $r^{p'}$ samples when the target only depends on a rank $r \ll d$ projection of the input [18]. However, $g^*(x^T A x)$ cannot be written in this form for some $r \ll d$, and thus existing guarantees do not apply. We conjecture that two-layer networks require $d^{\Omega(p)}$ samples when $g^*$ is a degree $p$ polynomial.

Altogether, the ability of three-layer networks to efficiently learn the class of functions $f^*(x) = g^*(x^T A x)$ hinges on their ability to extract the correct *nonlinear* feature. Empirical validation of the above examples is given in Appendix A.

## 5 An Optimization-Based Depth Separation

We complement the learning guarantee in Section 4.2 with a lower bound showing that there exist functions in this class that cannot be approximated by a polynomial size two-layer network.

The class of candidate two-layer networks is as follows. For a parameter vector $\theta = (a, W, b_1, b_2)$, where $a \in \mathbb{R}^m, W \in \mathbb{R}^{m \times d}, b_1 \in \mathbb{R}^m, b_2 \in \mathbb{R}$, define the associated two-layer network as

$$N_\theta(x) := a^T \sigma(Wx + b_1) + b_2 = \sum_{i=1}^{m} a_i \sigma(w_i^T x + b_{1,i}) + b_2. \tag{22}$$

Let $\|\theta\|_\infty := \max(\|a\|_\infty, \|W\|_\infty, \|b_1\|_\infty, \|b_2\|_\infty)$ denote the maximum parameter value. We make the following assumption on $\sigma$, which holds for all commonly-used activations.

**Assumption 8.** *There exist constants $C_\sigma, \alpha_\sigma$ such that $|\sigma(z)| \leq C_\sigma(1 + |z|)^{\alpha_\sigma}$.*

Our main theorem establishing the separation is the following.

**Theorem 4.** *Let $d$ be a suffiently large even integer. Consider the target function $f^*(x) = \mathrm{ReLU}(x^T A x) - c_0$, where $A = \frac{1}{\sqrt{d}} U \begin{pmatrix} 0 & I_{d/2} \\ I_{d/2} & 0 \end{pmatrix} U^T$ for some orthogonal matrix $U$ and $c_0 = \mathbb{E}_{x \sim \nu} \left[ \mathrm{ReLU}(x^T A x) \right]$. Under Assumption 8, there exist constants $C_1, C_2, C_3, c_3$, depending only on $(C_\sigma, \alpha_\sigma)$, such that for any $c_3 \geq \epsilon \geq C_3 d^{-2}$, any two layer neural network $N_\theta(x)$ of width $m$ and population $L^2$ error bound $\|N_\theta - f^*\|_{L^2}^2 \leq \epsilon$ must satisfy $\max(m, \|\theta\|_\infty) \geq C_1 \exp \left( C_2 \epsilon^{-1/2} \log(d\epsilon) \right)$. However, Algorithm 1 outputs a predictor satisfying the population $L^2$ loss bound*

$$\mathbb{E}_x \left[ \left( f(x; \hat{\theta}) - f^*(x) \right)^2 \right] \lesssim O\left( \frac{d^4}{n} + \sqrt{\frac{d}{n}} + d^{-1/6} \right). \tag{23}$$

*after $T = \mathrm{poly}(d, m_1, m_2, n)$ timesteps.*

The lower bound follows from a modification of the argument in [19] along with an explicit decomposition of the ReLU function into spherical harmonics. We remark that the separation applies for any link function $g^*$ whose Gegenbauer coefficients decay sufficiently slow. The upper bound follows from an application of Theorem 1 to a smoothed version of ReLU, since ReLU is not in $\mathcal{W}^{2,\infty}([-1, 1])$. The full proof of the theorem is given in Appendix G.

**Remarks.** In order for a two-layer network to achieve test loss matching the $d^{-1/6}$ error floor, either the width or maximum weight norm of the network must be $\exp(\Omega(d^\delta))$ for some constant $\delta$; this lower bound is superpolynomial in $d$. As a consequence, gradient descent on a poly-width two-layer neural network with stable step size must run for superpolynomially many iterations in order for some weight to grow this large and thus converge to a solution with low test error. Therefore $f^*$ is not learnable via gradient descent in polynomial time. This reduction from a weight norm lower bound to a runtime lower bound is made precise in [45].

We next describe a specific example of such an $f^*$. Let $S$ be a $d/2$-dimensional subspace of $\mathbb{R}^d$, and let $A = d^{-\frac{1}{2}} P_S - d^{-\frac{1}{2}} P_S^\perp$, where $P_S, P_S^\perp$ are projections onto the subspace $S$ and and its orthogonal complement respectively. Then, $f^*(x) = \mathrm{ReLU}\left( 2d^{-\frac{1}{2}} \|P_S x\|^2 - d^{\frac{1}{2}} \right)$. [44] established an optimization-based separation for the target $\mathrm{ReLU}(1 - \|x\|)$, under a different distribution $\nu$. However, their analysis relies heavily on the rotational symmetry of the target, and they posed the question of learning $\mathrm{ReLU}(1 - \|P_S x\|)$ for some subspace $S$. Our separation applies to a similar target function, and crucially does not rely on this rotational invariance.

## 6 Discussion

In this work we showed that three-layer networks can both learn nonlinear features and leverage these features to obtain a provable sample complexity improvement over two-layer networks. There are a number of interesting directions for future work. First, can our framework be used to learn hierarchical functions of a larger class of features beyond quadratic functions? Next, since Theorem 1 is general

purpose and makes minimal distributional assumptions, it would be interesting to understand if it can be applied to standard empirical datasets such as CIFAR-10, and what the learned feature $\mathbb{K}f^*$ and hierarchical learning correspond to in this setting. Finally, our analysis studies the nonlinear feature learning that arises from a neural representation $\sigma_2(Vx)$ where $V$ is fixed at initialization. This alone was enough to establish a separation between two and three-layer networks. A fascinating question is to understand the additional features that can be learned when both $V$ and $W$ are jointly trained. Such an analysis, however, is incredibly challenging in feature-learning regimes.

## Acknowledgements

EN acknowledges support from a National Defense Science & Engineering Graduate Fellowship. AD acknowledges support from a NSF Graduate Research Fellowship. EN, AD, and JDL acknowledge support of the ARO under MURI Award W911NF-11-1-0304, the Sloan Research Fellowship, NSF CCF 2002272, NSF IIS 2107304, NSF CIF 2212262, ONR Young Investigator Award, and NSF CAREER Award 2144994. The authors would like to thank Itay Safran for helpful discussions.

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

## Limitations

One limitation of our work is that the innermost layer weights $V$ are fixed throughout training, and our training procedure is layerwise. We remark that this is already enough to establish an algorithmic separation between two and three-layer networks (Section 5), and that prior works on two-layer networks also rely on layerwise training procedures. It is a very interesting direction of future work to understand if training $V$ can lead to learning a larger class of functions. Another limitation is that our examples in Section 4 rely on the data distribution $\nu$ being either the standard Gaussian or uniform on the sphere. This allows us to carefully characterize the integral operator $\mathbb{K}$, and we remark that other works on two-layer networks make similar distributional assumptions (such as uniform on the hypercube). However, it is a very important direction of future work to extend these lines of work to more general data distributions.

## A    Empirical Validation

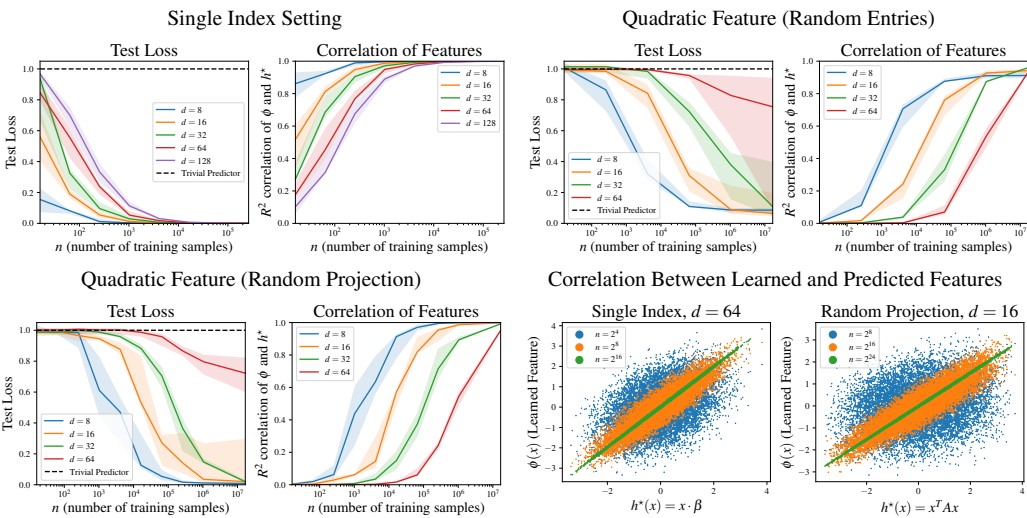

Figure 1: We ran Algorithm 1 on both the single index and quadratic feature settings described in Section 4. Each trial was run with 5 random seeds. The solid lines represent the medians and the shaded areas represent the min and max values. For every trial we recorded both the test loss on a test set of size $2^{15}$ and the linear correlation between the learned feature map $\phi(x)$ and the true intermediate feature $h^\star(x)$ where $h^\star(x) = x \cdot \beta$ for the single index setting and $h^\star(x) = x^T A x$ for the quadratic feature setting. Our results show that the test loss goes to $0$ as the linear correlation between the learned feature map $\phi$ and the true intermediate feature $h^\star$ approaches $1$.

We empirically verify our conclusions in the single index setting of Section 4.1 and the quadratic feature setting of Section 4.2:

**Single Index Setting**    We learn the target function $g^\star(w^\star \cdot x)$ using Algorithm 1 where $w^\star \in S^{d-1}$ is drawn randomly and $g^\star(x) = \text{sigmoid}(x) = \frac{1}{1+e^{-x}}$, which satisfies the condition $\mathbb{E}_{z \sim \mathcal{N}(0,1)}[g'(z)] \neq 0$. As in Theorem 2, we choose the initial activation $\sigma_2(z) = z$. We optimize the hyperparameters $\eta_1, \lambda$ using grid search over a holdout validation set of size $2^{15}$ and report the final error over a test set of size $2^{15}$.

**Quadratic Feature Setting**    We learn the target function $g^\star(x^T A x)$ using Algorithm 1 where $g^\star(x) = \text{ReLU}(x)$. We ran our experiments with two different choices of $A$:

- $A$ is symmetric with random entries, i.e. $A_{ij} \sim N(0,1)$ and $A_{ji} = A_{ij}$ for $i \leq j$.
- $A$ is a random projection, i.e. $A = \Pi_S$ where $S$ is a random $d/2$ dimensional subspace.

Both choices of $A$ were then normalized so that $\operatorname{tr} A = 0$ and $\|A\|_F = 1$ by subtracting the trace and dividing by the Frobenius norm. We chose initial activation $\sigma_2(z) = \operatorname{ReLU}(z)$. We note that in both examples, $\kappa = \Theta(1)$. As above, we optimize the hyperparameters $\eta_1, \lambda$ using grid search over a holdout validation set of size $2^{15}$ and report the final error over a test set of size $2^{15}$.

To focus on the sample complexity and avoid width-related bottlenecks, we directly simulate the infinite width limit ($m_2 \to \infty$) of Algorithm 1 by computing the kernel $\mathbb{K}$ in closed form. Finally, we run each trial with 5 random seeds and report the min, median, and max values in Figure 1.

**Comparison Between Two and Three-Layer Networks**   We also show that the sample complexity separation between two and three layer networks persists in standard training settings. In Figure 2, we train both a two and three-layer neural network on the target $f^*(x) = \operatorname{ReLU}(x^T A x)$, where $A$ is symmetric with random entries, as described above. Both networks are initialized using the $\mu$P parameterization [57] and are trained using SGD with momentum on all layers simultaneously. The input dimension is $d = 32$, and the widths are chosen to be 256 for the three-layer network and 2048 for the two-layer network, so that the parameter counts are approximately equal. Figure 2 plots the average test loss over 5 random seeds against sample size; here, we see that the three-layer network has a better sample complexity than the two-layer network.

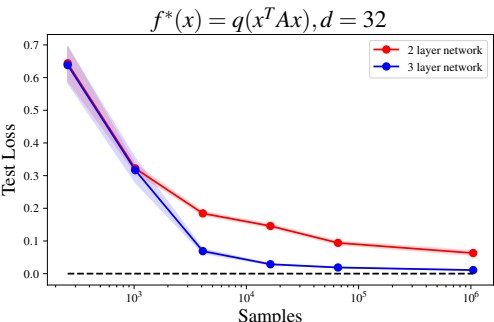

Figure 2: Three-layer neural networks learn the target $f^*(x) = \operatorname{ReLU}(x^T A x)$ with better sample complexity than two-layer networks.

**Experimental Details.**   Our experiments were written in JAX [14], and were run on a single NVIDIA RTX A6000 GPU.

# B   Notation

## B.1   Asymptotic Notation

Throughout the proof we will let $C$ be a fixed but sufficiently large constant.

**Definition 4** (high probability events)**.** *Let* $\iota = C \log(dnm_1 m_2)$*. We say that an event happens* with high probability *if it happens with probability at least* $1 - \operatorname{poly}(d, n, m_1, m_2)e^{-\iota}$.

**Example 5.** *If* $z \sim N(0, 1)$ *then* $|z| \le \sqrt{2\iota}$ *with high probability.*

Note that high probability events are closed under union bounds over sets of size $\operatorname{poly}(d, n, m_1, m_2)$. We will also assume throughout that $\iota \le C^{-1} d$.

## B.2   Tensor Notation

For a $k$-tensor $T \in (\mathbb{R}^d)^{\otimes k}$, let $\operatorname{Sym}$ be the symmetrization of $T$ across all $k$ axes, i.e

$$\operatorname{Sym}(T)_{i_1, \cdots, i_k} = \frac{1}{k!} \sum_{\sigma \in S_k} T_{i_{\sigma(1)}, \cdots, i_{\sigma(k)}}$$

Next, given tensors $A \in (\mathbb{R}^d)^{\otimes a}, B \in (\mathbb{R}^d)^{\otimes b}$, we let $A \otimes B \in (\mathbb{R}^d)^{\otimes a+b}$ be their tensor product.

**Definition 5** (Symmetric tensor product). *Given tensors $A \in (\mathbb{R}^d)^{\otimes a}$, $B \in (\mathbb{R}^d)^{\otimes b}$, we define their symmetric tensor product $A \tilde{\otimes} B \in (\mathbb{R}^d)^{\otimes a+b}$ as*

$$A \tilde{\otimes} B := \mathrm{Sym}(A \otimes B).$$

*We note that $\tilde{\otimes}$ satisfies associativity.*

**Definition 6** (Tensor contraction). *Given a symmetric $k$-tensor $T \in (\mathbb{R}^d)^{\otimes k}$ and an $a$-tensor $A \in (\mathbb{R}^d)^{\otimes a}$, where $k \geq a$, we define the tensor contraction $T(A)$ to be the $k - a$ tensor given by*

$$T(A)_{i_1,\ldots,i_{k-a}} = \sum_{(j_1,\ldots,j_a) \in [d]^a} T_{j_1,\ldots,j_a,i_1,\ldots,i_{k-a}} A_{j_1,\ldots,j_a}.$$

*When $A$ is also a $k$-tensor, then $\langle T, A \rangle := T(A) \in \mathbb{R}$ denotes their Euclidean inner product. We further define $\|T\|_F^2 := \langle T, T \rangle$.*

## C  Univariate Approximation

Throughout this section, let $\mu(b) := \frac{\exp(-b^2/2)}{\sqrt{2\pi}}$ denote the PDF of a standard Gaussian.

**Lemma 1.** *Let $a \sim \mathrm{Unif}(\{-1, 1\})$ and let $b \sim N(0, 1)$. Then there exists $v(a, b)$ supported on $\{-1, 1\} \times [0, 2]$ such that for any $|x| \leq 1$,*

$$\mathbb{E}_{a,b}[v(a,b)\sigma(ax+b)] = 1 \quad and \quad \sup_{a,b}|v(a,b)| \lesssim 1.$$

*Proof.* Let $v(a, b) = c\mathbf{1}_{b \in [1,2]}$ where $c = \frac{1}{\mu(1)-\mu(2)}$. Then for $|x| \leq 1$,

$$\mathbb{E}_{a,b}[v(a,b)\sigma(ax+b)] = c \int_1^2 \frac{1}{2}[\sigma(x+b) + \sigma(-x+b)]\mu(b)db$$

$$= c \int_1^2 b\mu(b)$$

$$= 1.$$

$\square$

**Lemma 2.** *Let $a \sim \mathrm{Unif}(\{-1, 1\})$ and let $b \sim N(0, 1)$. Then there exists $v(a, b)$ supported on $\{-1, 1\} \times [0, 2]$ such that for any $|x| \leq 1$,*

$$\mathbb{E}_{a,b}[v(a,b)\sigma(ax+b)] = x \quad and \quad \sup_{a,b}|v(a,b)| \lesssim 1.$$

*Proof.* Let $v(a, b) = ca\mathbf{1}_{b \in [1,2]}$ where $c = \frac{1}{\int_0^1 \mu(b)db}$. Then for $|x| \leq 1$,

$$\mathbb{E}_{a,b}[v(a,b)\sigma(ax+b)] = c \int_1^2 \frac{1}{2}[\sigma(x+b) - \sigma(-x+b)]\mu(b)db$$

$$= cx \int_1^2 \mu(b)db$$

$$= x.$$

$\square$

**Lemma 3.** *Let $f : [-1, 1] \to \mathbb{R}$ be any twice differentiable function. Then there exists $v(a, b)$ supported on $\{-1, 1\} \times [0, 2]$ such that for any $|x| \leq 1$,*

$$\mathbb{E}_{a,b}[v(a,b)\sigma(ax+b)] = f(x) \quad and \quad \sup_{a,b}|v(a,b)| \lesssim \sup_{\substack{x \in [-1,1] \\ k \in \{0,1,2\}}} |f^{(k)}(x)|.$$

*Proof.* First consider $v(a,b) = \frac{\mathbf{1}_{b\in[0,1]}}{\mu(b)} 2f''(-ab)$. Then when $x \geq 0$ we have by integration by parts:

$$\mathbb{E}_{ab}[v(a,b)\sigma(ax+b)]$$

$$= \int_0^1 [f''(-b)\sigma(x+b) + f''(b)\sigma(-x+b)]db$$

$$= f'(0)x - f'(-1)(x+1) + f'(1)(-x+1) + \int_0^1 f'(-b)db - \int_x^1 f'(b)db$$

$$= f(x) + c_1 + c_2 x$$

where $c_1 = f(0) - f(1) - f(-1) + f'(1) - f'(-1)$ and $c_2 = f'(0) - f'(1) - f'(-1)$. In addition when $x < 0$,

$$\mathbb{E}_{ab}[v(a,b)\sigma(ax+b)]$$

$$= \int_0^1 [f''(-b)\sigma(x+b) + f''(b)\sigma(-x+b)]db$$

$$= f'(0)x - f'(-1)(x+1) + f'(1)(-x+1) + \int_{-x}^1 f'(-b)db - \int_0^1 f'(b)db$$

$$= f(x) + c_1 + c_2 x$$

so this equality is true for all $x$. By Lemmas 1 and 2 we can subtract out the constant and linear terms so there exists $v(a,b)$ such that

$$\mathbb{E}_{a,b}[v(a,b)\sigma(ax+b)] = f(x).$$

In addition, using that $\mu(b) \gtrsim 1$ for $b \in [0,1]$ gives that this $v(a,b)$ satisfies

$$\sup_{a,b} |v(a,b)| \lesssim |c_1| + |c_2| + \max_{x\in[-1,1]} |f''(x)| \lesssim \sup_{\substack{x\in[-1,1]\\k\in\{0,1,2\}}} |f^{(k)}(x)|.$$

$\square$

# D    Proofs for Section 3

The following is a formal restatement of Theorem 1.

**Theorem 6.** *Select $q \in \mathcal{W}^{2,\infty}([-1,1])$. Let $\eta_1 = \frac{m_1}{m_2}\overline{\eta}$, and $n \gtrsim \|\mathbb{K}f^*\|_{L^2}^{-2}\iota^{2\ell+2\alpha_\sigma+1}, m_1 \gtrsim \|\mathbb{K}f^*\|_{L^2}^{-2}\iota$, $m_2 \gtrsim \|\mathbb{K}f^*\|_{L^2}^{-2}\iota^{2\alpha_\sigma+1}$. There exists a choice of $\overline{\eta} = \Theta(\iota^{-\chi}\|\mathbb{K}f^*\|_{L^2}^{-1})$, $T = poly(d,n,m_1,m_2,\|q\|_{2,\infty})$, and $\lambda, \eta_2$ such that with high probability the output $\hat{\theta}$ of Algorithm 1 satisfies the population $L^2$ loss bound*

$$\mathbb{E}_x\left[\left(f(x;\hat{\theta}) - f^*(x)\right)^2\right] \lesssim \|q\|_{1,\infty}^2 \|\mathbb{K}f^*\|_{L^2}^{-2} \cdot \left(\frac{\iota^{2\ell+2\alpha_\sigma+2\chi+1}}{n} + \frac{\iota^{2\alpha_\sigma+2\chi+1}}{m_2} + \frac{\iota}{m_1}\right)$$

$$+ \|q \circ (\overline{\eta} \cdot \mathbb{K}f^*) - f^*\|_{L^2}^2 + \frac{\|q\|_{2,\infty}^2 \iota}{m_1} + \sqrt{\frac{\|q\|_{2,\infty}^4 \iota + \iota^{4\ell+1}}{n}}$$

*Proof of Theorem 6.* The gradient with respect to $w_j$ is

$$\nabla_{w_j} L_1(\theta^{(0)}) = \frac{1}{n}\sum_i^n \left(f(x_i;\theta^{(0)}) - f^*(x_i)\right)\nabla_{w_j} f(x_i;\theta^{(0)})$$

$$= \sigma_1'(b_j^{(0)}) \cdot \frac{1}{n}\sum_{i=1}^n \left(f(x_i;\theta^{(0)}) - f^*(x_i)\right)\frac{a_j^{(0)}}{m_1}h^{(0)}(x_i)$$

$$= \mathbf{1}_{b_j^{(0)}>0} \cdot \frac{1}{n}\sum_{i=1}^n \left(f(x_i;\theta^{(0)}) - f^*(x_i)\right)\frac{a_j^{(0)}}{m_1}h^{(0)}(x_i)$$

Therefore

$$w_j^{(1)} = -\eta_1 \nabla_{w_j} L_1(\theta^{(0)})$$

$$= \mathbf{1}_{b_j^{(0)}>0} \cdot \frac{\overline{\eta}}{m_2} a_j^{(0)} \frac{1}{n} \sum_{i=1}^n \Big( f^*(x_i) - f(x_i; \theta^{(0)}) \Big) h^{(0)}(x_i).$$

One then has

$$f(x; (a, W^{(1)}, b^{(1)}, V^{(0)}))$$

$$= \sum_{j=1}^{m_1} \frac{a_j}{m_1} \sigma_1 \left( \mathbf{1}_{b_j^{(0)}>0} \cdot a_j^{(0)} \cdot \frac{\overline{\eta}}{m_2} \frac{1}{n} \sum_{i=1}^n \Big( f^*(x_i) - f(x_i; \theta^{(0)}) \Big) \langle h^{(0)}(x_i), h^{(0)}(x) \rangle + b_j \right)$$

$$= \sum_{j=1}^{m_1} \frac{a_j}{m_1} \sigma_1 (a_j^{(0)} \cdot \overline{\eta} \phi(x) + b_j) \cdot \mathbf{1}_{b_j^{(0)}>0},$$

where $\phi(x) := \frac{1}{m_2 n} \sum_{i=1}^n \big( f^*(x_i) - (x_i; \theta^{(0)}) \big) \langle h^{(0)}(x_i), h^{(0)}(x) \rangle$.

The second stage of Algorithm 1 is equivalent to random feature regression. The next lemma shows that there exists $a^*$ with small norm that acheives low empirical loss on the dataset $\mathcal{D}_2$.

**Lemma 4.** *There exists $a^*$ with $\|a^*\|_2 \lesssim \|q\|_{2,\infty} \sqrt{m_1}$ such that $\theta^* = \big(a^*, W^{(1)}, b^{(0)}, V^{(0)}\big)$ satisfies*

$$L_2(\theta^*) \lesssim \|q\|_{1,\infty}^2 \|\mathbb{K}f^*\|_{L^2}^{-2} \cdot \left( \frac{\iota^{2\ell+2\alpha_\sigma+2\chi+1}}{n} + \frac{\iota^{2\alpha_\sigma+2\chi+1}}{m_2} + \frac{\iota}{m_1} \right)$$

$$+ \frac{\iota^{2\ell+1}}{n} + \frac{\|q\|_{2,\infty}^2 \iota}{m_1} + \|q \circ (\overline{\eta} \cdot \mathbb{K}f^*) - f^*\|_{L^2}^2$$

The proof of Lemma 4 is deferred to Appendix D.1. We first show that $\phi$ is approximately proportional to $\mathbb{K}f^*$, and then invoke the ReLU random feature expressivity results from Appendix C.

Next, set

$$\lambda = \|a^*\|_2^{-2} \left( \|q\|_{1,\infty}^2 \|\mathbb{K}f^*\|_{L^2}^{-2} \cdot \left( \frac{\iota^{2\ell+2\alpha_\sigma+2\chi+1}}{n} + \frac{\iota^{2\alpha_\sigma+2\chi+1}}{m_2} + \frac{\iota}{m_1} \right) \right.$$

$$\left. + \frac{\iota^{2\ell+1}}{n} + \frac{\|q\|_{2,\infty}^2 \iota}{m_1} + \|q \circ (\overline{\eta}(\mathbb{K}f^*)) - f^*\|_{L^2}^2 \right),$$

so that $L_2(\theta^*) \lesssim \|a^*\|_2^2 \lambda$.

Define the regularized loss to be $\hat{L}(a) := L_2\big((a, W^{(1)}, b^{(0)}, V^{(0)})\big) + \frac{\lambda}{2}\|a\|_2^2$. Let $a^{(\infty)} = \arg\min_a \hat{L}(a)$, and $a^{(t)}$ be the predictor from running gradient descent for $t$ steps initialized at $a^{(0)}$. We first note that

$$\hat{L}(a^{(\infty)}) \le \hat{L}(a^*) \lesssim \|a^*\|_2^2 \lambda.$$

Next, we remark that $\hat{L}(a)$ is $\lambda$-strongly convex. Additionally, we can write $f(x; (a, W^{(1)}, b^{(0)}, V^{(0)})) = a^T \psi(x)$ where $\psi(x) = \text{Vec}\left(\frac{1}{m_1} \sigma_1(a_j^{(0)} \overline{\eta} \phi(x) + b_j^{(0)}) \cdot \mathbf{1}_{b_j^{(0)}>0}\right)$. In Appendix D.2 we show $\sup_{x \in \mathcal{D}_2} \|\psi(x)\| \lesssim \frac{1}{m_1}$. Therefore

$$\lambda_{max}\left(\nabla_a^2 \hat{L}\right) \le \frac{2}{n} \sum_{x \in \mathcal{D}_2} \|\psi(x)\|^2 \lesssim \frac{1}{m_1},$$

so $\hat{L}$ is $O(\frac{1}{m_1}) + \lambda$ smooth. Choosing a learning rate $\eta_2 = \Theta(m_1)$, after $T = \tilde{O}\left(\frac{1}{\lambda m_1}\right) = \text{poly}(d, n, m_1, m_2, \|q\|_{2,\infty})$ steps we reach an iterate $\hat{a} = a^{(T)}, \hat{\theta} = (\hat{a}, W^{(1)}, b^{(0)}, V^{(0)})$ satisfying

$$L_2(\hat{\theta}) \lesssim L_2(\theta^*) \quad \text{and} \quad \|\hat{a}\|_2 \lesssim \|a^*\|_2.$$

For $\tau > 0$, define the truncated loss $\ell_\tau$ by $\ell_\tau(z) = \min(z^2, \tau^2)$. We have that $\ell_\tau(z) \leq z^2$, and thus

$$\frac{1}{n} \sum_{x \in \mathcal{D}_2} \ell_\tau\Big(f(x; \hat{\theta}) - f^*(x)\Big) \leq L_2(\hat{\theta}).$$

Consider the function class

$$\mathcal{F}(B_a) := \{f(\cdot; (a, W^{(1)}, b^{(0)}, V^{(0)}) : \|a\|_2 \leq B_a)\}$$

The following lemma bounds the empirical Rademacher complexity of this function class

**Lemma 5.** *Given a dataset $\mathcal{D}$, recall that the empirical Rademacher complexity of $\mathcal{F}$ is defined as*

$$\mathcal{R}_\mathcal{D}(\mathcal{F}) := \mathbb{E}_{\sigma \in \{\pm 1\}^n} \left[ \sup_{f \in \mathcal{F}} \frac{1}{n} \sum_{i=1}^n \sigma_i f(x_i) \right].$$

*Then with high probability*

$$\mathcal{R}_{\mathcal{D}_2}(\mathcal{F}(B_a)) \lesssim \sqrt{\frac{B_a^2}{n m_1}}.$$

Since $\ell_\tau$ loss is $2\tau$-Lipschitz, the above lemma with $B_a = O(\|a^*\|_2) = O\Big(\|q\|_{2,\infty} \sqrt{m_1}\Big)$ along with the standard empirical Rademacher complexity bound yields

$$\mathbb{E}_x \ell_\tau\Big(f(x; \hat{\theta}) - f^*(x)\Big) \lesssim \frac{1}{n} \sum_{x \in \mathcal{D}_2} \ell_\tau\Big(f(x; \hat{\theta}) - f^*(x)\Big) + \tau \cdot \mathcal{R}_{\mathcal{D}_2}(\mathcal{F}) + \tau^2 \sqrt{\frac{\iota}{n}}$$

$$\lesssim \|q\|_{1,\infty}^2 \|\mathbb{K}f^*\|_{L^2}^{-2} \cdot \left( \frac{\iota^{2\ell + 2\alpha_\sigma + 2\chi + 1}}{n} + \frac{\iota^{2\alpha_\sigma + 2\chi + 1}}{m_2} + \frac{\iota}{m_1} \right)$$

$$+ \|q \circ (\bar{\eta} \cdot \mathbb{K}f^*) - f^*\|_{L^2}^2 + \frac{\iota^{2\ell+1}}{n} + \frac{\|q\|_{2,\infty}^2 \iota}{m_1}$$

$$+ \tau \sqrt{\frac{\|q\|_{2,\infty}^2}{n}} + \tau^2 \sqrt{\frac{\iota}{n}}.$$

Finally, we relate the $\ell_\tau$ population loss to the $\ell^2$ population loss.

**Lemma 6.** *Let $\tau = \Theta(\max(\iota^\ell, \|q\|_{2,\infty}))$. Then with high probability over $\hat{\theta}$,*

$$\mathbb{E}_x \left[ \Big(f(x; \hat{\theta}) - f^*(x)\Big)^2 \right] \leq \mathbb{E}_x \ell_\tau\Big(f(x; \hat{\theta}) - f^*(x)\Big) + \frac{\|q\|_{2,\infty}^2}{m_1}.$$

Plugging in $\tau$ and applying Lemma 6 yields

$$\mathbb{E}_x \left[ \Big(f(x; \hat{\theta}) - f^*(x)\Big)^2 \right] \lesssim \|q\|_{1,\infty}^2 \|\mathbb{K}f^*\|_{L^2}^{-2} \cdot \left( \frac{\iota^{2\ell + 2\alpha_\sigma + 2\chi + 1}}{n} + \frac{\iota^{2\alpha_\sigma + 2\chi + 1}}{m_2} + \frac{\iota}{m_1} \right)$$

$$+ \|q \circ (\bar{\eta} \cdot \mathbb{K}f^*) - f^*\|_{L^2}^2 + \frac{\|q\|_{2,\infty}^2 \iota}{m_1} + \sqrt{\frac{\|q\|_{2,\infty}^4 \iota + \iota^{4\ell+1}}{n}}$$

as desired. $\qquad\square$

### D.1 Proof of Lemma 4

We require three auxiliary lemmas, all of whose proofs are deferred to Appendix D.4. The first lemma bounds the error between the population learned feature $\mathbb{K}f^*$ and the finite sample learned feature $\phi$ over the dataset $\mathcal{D}_2$.

**Lemma 7.** *With high probability*

$$\sup_{x \in \mathcal{D}_2} |\phi(x) - (\mathbb{K}f^*)(x)| \lesssim \sqrt{\frac{\iota^{2\ell + 2\alpha_\sigma + 1}}{n}} + \sqrt{\frac{\iota^{2\alpha_\sigma + 1}}{m_2}} + \sqrt{\frac{\iota}{m_1}}.$$

The second lemma shows that with appropriate choice of $\eta$, the quantity $\eta\phi(x)$ is small.

**Lemma 8.** *Let* $n \gtrsim \|\mathbb{K}f^*\|_{L^2}^{-2} \iota^{2\ell + 2\alpha_\sigma + 1}, m_1 \gtrsim \|\mathbb{K}f^*\|_{L^2}^{-2}\iota$, *and* $m_2 \gtrsim \|\mathbb{K}f^*\|_{L^2}^{-2}\iota^{2\alpha_\sigma + 1}$. *There exists* $\overline{\eta} = \tilde{\Theta}(\|\mathbb{K}f^*\|_{L^2}^{-2})$ *such that with high probability,* $\sup_{x \in \mathcal{D}_2} |\overline{\eta}\phi(x)| \leq 1$ *and* $\sup_{x \in \mathcal{D}_2} |\overline{\eta} \cdot \mathbb{K}f^*(x)| \leq 1$

The third lemma expresses the compositional function $q \circ \overline{\eta}\phi$ via an infinite width network.

**Lemma 9.** *Assume that* $n, m_1, m_2 = \tilde{\Omega}(\|\mathbb{K}f^*\|_{L^2}^{-2})$. *There exists* $v : \{\pm 1\} \times \mathbb{R} \to \mathbb{R}$ *such that* $\sup_{a,b} |v(a,b)| \leq \|q\|_{2,\infty}$ *and, with high probability over* $\mathcal{D}_2$, *the infinite width network*

$$f_v^\infty(x) := \mathbb{E}_{a,b}[v(a,b)\sigma_1(a\overline{\eta}\phi(x) + b)\mathbf{1}_{b>0}]$$

*satisfies*

$$f_v^\infty(x) = q(\overline{\eta}\phi(x))$$

*for all* $x \in \mathcal{D}_2$.

*Proof of Lemma 4.* Let $v$ be the infinite width construction defined in Lemma 9. Define $a^* \in \mathbb{R}^{m_1}$ to be the vector with $a_j^* = v(a_j^{(0)}, b_j^{(1)})$. We can decompose

$$
\begin{aligned}
L_2(\theta^*) &= \frac{1}{n} \sum_{x \in \mathcal{D}_2} (f(x; \theta^*) - f^*(x))^2 \\
&\lesssim \frac{1}{n} \sum_{x \in \mathcal{D}_2} (f(x; \theta^*) - f_v^\infty(x))^2 + \frac{1}{n} \sum_{x \in \mathcal{D}_2} (f_v^\infty(x) - q(\overline{\eta}\phi(x)))^2 \\
&\quad + \frac{1}{n} \sum_{x \in \mathcal{D}_2} (q(\overline{\eta}\phi(x)) - q(\overline{\eta}(\mathbb{K}f^*)(x)))^2 + \frac{1}{n} \sum_{x \in \mathcal{D}_2} (q(\overline{\eta}(\mathbb{K}f^*)(x)) - f^*)^2
\end{aligned}
$$

Take $m_1 \gtrsim \iota$. The first term is the error between the infinite width network $f_v^\infty(x)$ and the finite width network $f(x; \theta^*)$. This error can be controlled via standard concentration arguments: by Corollary 2 we have that with high probability $\|a^*\| \lesssim \frac{\|q\|_{2,\infty}}{\sqrt{m_1}}$, and by Lemma 17 we have

$$\frac{1}{n} \sum_{x \in \mathcal{D}_2} (f(x; \theta^*) - f_v^\infty(x))^2 \lesssim \frac{\|q\|_{2,\infty}^2 \iota}{m_1}.$$

Next, by Lemma 9 we get that the second term is zero with high probability. We next turn to the third term. Since $q$ is $\|q\|_{1,\infty}$-Lipschitz on $[-1, 1]$, and $\sup_{x \in \mathcal{D}_2} |\overline{\eta}\phi(x)| \leq 1, \sup_{x \in \mathcal{D}_2} |(\overline{\eta} \cdot \mathbb{K}f^*)(x)| \leq 1$ by Lemma 8, we can apply Lemma 7 to get

$$
\begin{aligned}
\sup_{x \in \mathcal{D}_2} |q(\overline{\eta}\phi(x)) - q(\overline{\eta}(\mathbb{K}f^*)(x))| &\leq \|q\|_{1,\infty} \sup_{x \in \mathcal{D}_2} |\overline{\eta}\phi(x) - \overline{\eta}(\mathbb{K}f^*)(x)| \\
&\lesssim \|q\|_{1,\infty} \overline{\eta} \left( \sqrt{\frac{\iota^{2\ell + 2\alpha_\sigma + 1}}{n}} + \sqrt{\frac{\iota^{2\alpha_\sigma + 1}}{m_2}} + \sqrt{\frac{\iota}{m_1}} \right) \\
&\lesssim \|q\|_{1,\infty} \|\mathbb{K}f^*\|_{L^2}^{-1} \left( \sqrt{\frac{\iota^{2\ell + 2\alpha_\sigma + 1}}{n}} + \sqrt{\frac{\iota^{2\alpha_\sigma + 1}}{m_2}} + \sqrt{\frac{\iota}{m_1}} \right)
\end{aligned}
$$

and thus

$$\frac{1}{n} \sum_{x \in \mathcal{D}_2} (q(\overline{\eta}\phi(x)) - q(\overline{\eta}(\mathbb{K}f^*)(x)))^2 \lesssim \|q\|_{1,\infty}^2 \|\mathbb{K}f^*\|_{L^2}^{-2} \cdot \left( \frac{\iota^{2\ell + 2\alpha_\sigma + 2\chi + 1}}{n} + \frac{\iota^{2\alpha_\sigma + 2\chi + 1}}{m_2} + \frac{\iota}{m_1} \right).$$

Finally, we must relate the empirical error between $q \circ \overline{\eta} \cdot \mathbb{K}f^*$ and $f^*$ to the population error. This can be done via standard concentration arguments: in Lemma 19, we show

$$\frac{1}{n} \sum_{x \in \mathcal{D}_2} (q(\overline{\eta}(\mathbb{K}f^*)(x)) - f^*(x))^2 \lesssim \|q \circ (\overline{\eta}(\mathbb{K}f^*)) - f^*\|_{L^2}^2 + \frac{\|q\|_\infty^2 \iota + \iota^{2\ell+1}}{n}.$$

Altogether,

$$L_2(\theta^*) \lesssim \|q\|_{1,\infty}^2 \|\mathbb{K}f^*\|_{L^2}^{-2} \cdot \left( \frac{\iota^{2\ell+2\alpha_\sigma+2\chi+1}}{n} + \frac{\iota^{2\alpha_\sigma+2\chi+1}}{m_2} + \frac{\iota}{m_1} \right) + \frac{\|q\|_\infty^2 \iota + \iota^{2\ell+1}}{n} + \frac{\|q\|_{2,\infty}^2 \iota}{m_1}$$
$$\qquad + \|q \circ (\overline{\eta}(\mathbb{K}f^*)) - f^*\|_{L^2}^2$$
$$\lesssim \|q\|_{1,\infty}^2 \|\mathbb{K}f^*\|_{L^2}^{-2} \cdot \left( \frac{\iota^{2\ell+2\alpha_\sigma+2\chi+1}}{n} + \frac{\iota^{2\alpha_\sigma+2\chi+1}}{m_2} + \frac{\iota}{m_1} \right) + \frac{\iota^{2\ell+1}}{n} + \frac{\|q\|_{2,\infty}^2 \iota}{m_1}$$
$$\qquad + \|q \circ (\overline{\eta}(\mathbb{K}f^*)) - f^*\|_{L^2}^2$$

since $\|\mathbb{K}\|_{op} \lesssim 1$ and thus $\|\mathbb{K}f^*\|_{L^2} \gtrsim 1$. $\qquad\qquad\square$

## D.2 Proof of Lemma 5

*Proof.* The standard linear Rademacher bound states that for functions of the form $x \mapsto w^T \psi(x)$ with $\|w\|_2 \lesssim B_w$, the empirical Rademacher complexity is upper bounded by

$$\frac{B_w}{n} \sqrt{\sum_{x \in \mathcal{D}} \|\psi(x)\|^2}.$$

We have that $f(x; \theta) = a^T \psi(x)$, where

$$\psi(x) = \text{Vec} \left( \frac{1}{m_1} \sigma_1(a_j^{(0)} \overline{\eta} \phi(x) + b_j^{(0)}) \cdot \mathbf{1}_{b_j^{(0)} > 0} \right).$$

By Lemma 8, with high probability, $\sup_{x \in \mathcal{D}_2} |\overline{\eta} \phi(x)| \leq 1$. Thus for $x \in \mathcal{D}_2$

$$\|\psi(x)\|^2 \leq \frac{1}{m_1^2} (1 + |b_j|)^2 \lesssim \frac{1}{m_1}$$

with high probability by Lemma 10. Altogether, we have

$$\mathcal{R}_{\mathcal{D}_2}(\mathcal{F}(B_a)) \lesssim \sqrt{\frac{B_a \cdot n \cdot 1/m_1}{n}} = \sqrt{\frac{B_a^2}{nm_1}}.$$

$\qquad\qquad\square$

## D.3 Proof of Lemma 6

*Proof.* We can bound

$$\mathbb{E}_x \left[ \left( f(x; \hat{\theta}) - f^*(x) \right)^2 \right] - \mathbb{E}_x \left[ \ell_\tau \left( f(x; \hat{\theta}) - f^*(x) \right) \right]$$
$$= \mathbb{E}_x \left[ \left( \left( f(x; \hat{\theta}) - f^*(x) \right)^2 - \tau^2 \right) \cdot \mathbf{1}_{|f(x;\hat{\theta}) - f^*(x)| \geq \tau} \right]$$
$$\leq \mathbb{E}_x \left[ \left( f(x; \hat{\theta}) - f^*(x) \right)^2 \cdot \mathbf{1}_{|f(x;\hat{\theta}) - f^*(x)| \geq \tau} \right]$$
$$\leq \mathbb{E}_x \left[ \left( f(x; \hat{\theta}) - f^*(x) \right)^4 \right]^{1/2} \cdot \mathbb{P}\left( \left| f(x; \hat{\theta}) - f^*(x) \right| \geq \tau \right)^{1/2}$$
$$\lesssim \left[ \mathbb{E}_x \left[ f(x; \hat{\theta})^4 \right]^{1/2} + \mathbb{E}_x \left[ f^*(x)^4 \right]^{1/2} \right] \cdot \left[ \mathbb{P}\left( \left| f(x; \hat{\theta}) \right| \geq \tau \right) + \mathbb{P}(|f^*(x)| \geq \tau) \right]$$

Next, we bound $f(x; \hat{\theta})$:

$$\left| f(x; \hat{\theta}) \right| = \left| \frac{1}{m_1} \sum_{j=1}^{m_1} \hat{a}_j \sigma_1 \left( a_j^{(0)} \cdot \overline{\eta}\phi(x) + b_j \right) \mathbf{1}_{b_j^{(0)} > 0} \right|$$

$$\leq \frac{1}{m_1} \sum_{j=1}^{m_1} |\hat{a}_j| (|\overline{\eta}\phi(x)| + |b_j|)$$

$$\leq \frac{\|\hat{a}\|_2}{\sqrt{m_1}} |\overline{\eta}\phi(x)| + \frac{1}{m_1} \|\hat{a}\|_2 \|b\|_2$$

By Lemma 10, with high probability over the initialization we have $\|b\|_2 \lesssim \sqrt{m_1}$. Next, by Lemma 4, with high probability over the initialization and dataset we have $\|\hat{a}\|_2 \lesssim \|a^*\|_2 \lesssim \|q\|_{2,\infty} \sqrt{m_1}$. Thus with high probability we have

$$\left| f(x; \hat{\theta}) \right| \lesssim \|q\|_{2,\infty} (|\overline{\eta}\phi(x)| + 1)$$

uniformly over $x$.

We naively bound

$$|\phi(x)| \leq \frac{1}{m_2} \left\| h^{(0)}(x) \right\| \cdot \frac{1}{n} \sum_{i=1}^{n} \left\| h^{(0)}(x_i) \right\| \left| f(x_i; \theta^{(0)}) - f^*(x_i) \right|.$$

By Lemma 11 and Lemma 12, with high probability we have $\left| f(x_i; \theta^{(0)}) - f^*(x_i) \right| \lesssim \iota^\ell$ for all $i$. With high probability, we also have $\left\| h^{(0)}(x_i) \right\| \lesssim \sqrt{m_2}$. Additionally, we have

$$\left\| h^{(0)}(x) \right\|_2^4 = \left( \sum_{j=1}^{m_2} \sigma_2 (x \cdot v_j)^2 \right)^2$$

$$\leq m_2 \sum_{j=1}^{m_2} \sigma_2 (x \cdot v_j)^4$$

$$\lesssim m_2 \sum_{j=1}^{m_2} (1 + |x \cdot v_j|^{4\alpha_\sigma})$$

Since $x$ is $C_\gamma$ subGaussian and $\|v\| = 1$, we have

$$\mathbb{E}_x \left\| h^{(0)}(x) \right\|_2^4 \lesssim m_2^2 \mathbb{E} |x \cdot v_j|^{4\alpha\sigma}$$

$$\lesssim m_2^2.$$

Altogether,

$$\mathbb{E}_x |\phi(x)|^4 \lesssim \iota^{4\ell},$$

and thus

$$\mathbb{E}_x \left[ \left| f(x; \hat{\theta}) \right|^4 \right]^{1/2} \lesssim \|q\|_{2,\infty}^2 \overline{\eta}^2 \iota^{2\ell} \leq \|q\|_{2,\infty}^2 m_1^2 \|\mathbb{K}f^*\|_{L^2}^{-2} \iota^{2\ell}$$

By Assumption 2, $\mathbb{E}_x \left[ (f^*(x))^4 \right] \lesssim 1$.

We choose $\tau = \Theta(\max(\iota^\ell, \|q\|_{2,\infty}))$. By Lemma 7 and Lemma 8 $\{ \left| f(x, \hat{\theta}) \right| > \tau \}$ and $\{ |f^*(x)| > \tau \}$ are both high probability events. Therefore by choosing $C$ sufficiently large, we have the bound

$$\left[ \mathbb{P} \left( \left| f(x; \hat{\theta}) \right| \geq \tau \right) + \mathbb{P}(|f^*(x)| \geq \tau) \right] \leq m_1^{-4} \iota^{-2\ell}$$

Altogether, this gives

$$\mathbb{E}_x \left[ \left( f(x; \hat{\theta}) - f^*(x) \right)^2 \right] - \mathbb{E}_x \left[ \ell_\tau \left( f(x; \hat{\theta}) - f^*(x) \right) \right] \lesssim \frac{\|q\|_{2,\infty}^2 \|\mathbb{K}f^*\|_{L^2}^{-2}}{m_1^2} + \frac{1}{m_1^4} \leq \frac{\|q\|_{2,\infty}^2}{m_1}.$$

$\square$

## D.4 Auxiliary Lemmas

*Proof of Lemma 7.* We can decompose

$$|\phi(x) - (\mathbb{K}f^*)(x)| = \left| \frac{1}{m_2} \cdot \frac{1}{n} \sum_{i=1}^{n} \left( f(x_i; \theta^{(0)}) - f^*(x_i) \right) \langle h^{(0)}(x_i), h^{(0)}(x) \rangle - (\mathbb{K}f^*)(x) \right|$$

$$\leq \frac{1}{m_2 n} \sum_{i=1}^{n} \left| f(x_i; \theta^{(0)}) \langle h^{(0)}(x_i), h^{(0)}(x) \rangle \right|$$

$$+ \frac{1}{m_2} \left| \frac{1}{n} \sum_{i=1}^{n} f^*(x_i) \langle h^{(0)}(x_i), h^{(0)}(x) \rangle - \left\langle \mathbb{E}_x \left[ f^*(x) h^{(0)}(x) \right], h^{(0)}(x) \right\rangle \right|$$

$$+ \left| \frac{1}{m_2} \left\langle \mathbb{E}_x \left[ f^*(x) h^{(0)}(x) \right], h^{(0)}(x) \right\rangle - (\mathbb{K}f^*)(x) \right|.$$

By Lemma 11 and Lemma 13 we can upper bound the first term by $\sqrt{\frac{\iota}{m_1}}$.

For the second term, by Lemma 13 and Lemma 15 we have for all $x \in \mathcal{D}_2$

$$\frac{1}{m_2} \left\| \frac{1}{n} \sum_{i=1}^{n} f^*(x_i) h^{(0)}(x_i) - \mathbb{E}_x \left[ f^*(x) h^{(0)}(x) \right] \right\|_2 \left\| h^{(0)}(x) \right\|_2 \lesssim \frac{1}{m_2} \sqrt{m_2} \cdot \sqrt{\frac{m_2 \iota^{2\ell + 2\alpha_\sigma + 1}}{n}} \lesssim \sqrt{\frac{\iota^{2\ell + 2\alpha_\sigma + 1}}{n}}.$$

The third term can be bounded as

$$\left| \frac{1}{m_2} \sum_{i=1}^{m_2} \mathbb{E}_{x'}[f^*(x') \sigma_2(x' \cdot v_j) \sigma_2(x \cdot v_j)] - \mathbb{E}_{x', v}[f^*(x) \sigma_2(x' \cdot v) \sigma_2(x \cdot v)] \right| \lesssim \sqrt{\frac{\iota^{2\alpha_\sigma + 1}}{m_2}}$$

$\square$

*Proof of Lemma 8.* Conditioning on the event where Corollary 1 holds, and with the choice $\overline{\eta} = \frac{1}{2} C_K^{-1} e^{-1} \|\mathbb{K}f^*\|_{L^2}^{-1} \iota^{-\chi}$, we get that with high probability

$$\sup_{x \in \mathcal{D}_2} |(\overline{\eta} \cdot \mathbb{K}f^*)(x)| \leq \frac{1}{2}.$$

Therefore

$$\sup_{x \in \mathcal{D}_2} |\overline{\eta} \phi(x)| = C_K^{-1} e^{-1} \|\mathbb{K}f^*\|_{L^2}^{-1} \iota^{-\chi} \cdot \sup_x |\phi(x)|$$

$$\leq C_K^{-1} e^{-1} \|\mathbb{K}f^*\|_{L^2}^{-1} \iota^{-\chi} \left[ \sup_x |(\mathbb{K}f^*)(x)| + O\left( \sqrt{\frac{\iota^{2\ell + 2\alpha_\sigma + 1}}{n}} + \sqrt{\frac{\iota^{2\alpha_\sigma + 1}}{m_2}} + \sqrt{\frac{\iota}{m_1}} \right) \right]$$

$$\leq \frac{1}{2} + O\left( \|\mathbb{K}f^*\|_{L^2}^{-1} \iota^{-\chi} \cdot \left( \sqrt{\frac{\iota^{2\ell + 2\alpha_\sigma + 1}}{n}} + \sqrt{\frac{\iota^{2\alpha_\sigma + 1}}{m_2}} + \sqrt{\frac{\iota}{m_1}} \right) \right)$$

$$\leq 1.$$

$\square$

*Proof of Lemma 9.* Conditioning on the event where Lemma 8 holds and applying Lemma 3, we get that there exists $v$ such that

$$\mathbb{E}_{a,b}[v(a,b) \sigma_1(a\overline{\eta}\phi(x) + b)] = q(\overline{\eta}\phi(x))$$

for all $x \in \mathcal{D}_2$. Since $v(a,b) = 0$ for $b \leq 0$, we have that $v(a,b) = v(a,b)\mathbf{1}_{b>0}$. Thus the desired claim is true. $\square$

### D.5 Concentration

**Lemma 10.** *Let $m_1 \gtrsim \iota$. With high probability,*

$$\frac{1}{m_1} \sum_{i=1}^{m_1} \left(b_i^{(0)}\right)^2 \lesssim 1.$$

*Proof.* By Bernstein, we have

$$\left| \frac{1}{m_1} \sum_{i=1}^{m_1} \left(b_i^{(0)}\right)^2 - 1 \right| \lesssim \sqrt{\frac{\iota}{m_1}} \leq 1.$$

$\square$

**Lemma 11.** *With high probability, $\left| f(x; \theta^{(0)}) \right| \lesssim \sqrt{\frac{\iota}{m_1}}$. for all $x$.*

*Proof.* We have

$$f(x; \theta^{(0)}) = \frac{1}{m_1} \sum_{i=1}^{m_1} a_i \sigma_1(b_i).$$

Since $a_i \sim \text{Unif}(\{\pm 1\})$ and $b_i \sim \mathcal{N}(0, 1)$, the quantities $a_i \sigma_1(b_i)$ are 1-subGaussian, and thus by Hoeffding with high probability we have

$$\left| f(x; \theta^{(0)}) \right| \lesssim \sqrt{\frac{\iota}{m_1}}.$$

$\square$

**Lemma 12.** *With high probability*

$$\sup_{x \in \mathcal{D}_2} \mathbb{E}_v \left[ \sigma_2(x \cdot v)^4 \right] \lesssim 1 \quad and \quad \sup_{j \in [m_2]} \mathbb{E}_x \left[ \sigma_2(x \cdot v_j)^4 \right] \lesssim 1.$$

*Proof.* First, we have that $\sigma_2(x \cdot v)^4 \lesssim 1 + (x \cdot v)^{4\alpha_\sigma}$. Therefore

$$\mathbb{E}_v \left[ \sigma_2(x \cdot v)^4 \right] \lesssim 1 + \mathbb{E}_v \left[ (x \cdot v)^{4\alpha_\sigma} \right] \lesssim 1 + \|x\|^{4\alpha_\sigma} d^{-2\alpha_\sigma}.$$

Next, with high probability we have $\sup_{x \in \mathcal{D}_2} \left| \|x\|^2 - \mathbb{E}\|x\|^2 \right| \leq \iota C_\gamma^2 \lesssim \iota$. Since $\mathbb{E}\|x\|^2 \lesssim d\gamma_x^2 \lesssim d$, we can thus bound

$$\sup_{x \in \mathcal{D}_2} \mathbb{E}_v \left[ \sigma_2(x \cdot v)^4 \right] \lesssim 1 + \gamma_v^{4\alpha_\sigma} \sup_{x \in \mathcal{D}_2} \|x\|^2$$
$$\lesssim 1 + d^{-2\alpha_\sigma} (d + \iota)^{2\alpha_\sigma}$$
$$\lesssim 1.$$

The proof for the other inequality is identical. $\square$

**Lemma 13.** *Let $m_2 \gtrsim \iota^{2\alpha_\sigma + 1}$. With high probability, $\sup_{x \in \mathcal{D}_1 \cup \mathcal{D}_2} \left\| h^{(0)}(x) \right\| \lesssim \sqrt{m_2}$*

*Proof.* Since $x$ is $C_\gamma$ subGaussian and $v$ is $1/\sqrt{d}$ subGaussian, with high probability we have

$$x \cdot v \leq O(C_\gamma \iota) = O(\iota).$$

Union bounding over $x \in \mathcal{D}_2, j \in [m_2]$, with high probability we have $\sup_{x \in \mathcal{D}_1 \cup \mathcal{D}_2, j \in [m_2]} |x \cdot v_j| \leq O(\iota)$. Therefore

$$\sup_{x, v_j} |\sigma_2(x \cdot v_j)| \leq O(\iota^{\alpha_\sigma}).$$

Next, see that

$$\frac{1}{m_2}\left\|h^{(0)}(x)\right\|^2 = \frac{1}{m_2}\sum_{j=1}^{m_2}\sigma_2(x\cdot v_j)^2.$$

Pick truncation radius $R = O(\iota^{\alpha_\sigma})$. By Hoeffding's inequality and a union bound over $x \in \mathcal{D}_1 \cup \mathcal{D}_2$, we have that with high probability

$$\sup_{x\in\mathcal{D}_1\cup\mathcal{D}_2}\left|\frac{1}{m_2}\sum_{j=1}^{m_2}\mathbf{1}_{\sigma_2(x\cdot v_j)^2\leq R}\sigma_2(x\cdot v_j)^2 - \mathbb{E}_v\left[\mathbf{1}_{\sigma_2(x\cdot v_j)^2\leq R}\sigma_2(x\cdot v)^2\right]\right| \lesssim \sqrt{\frac{\iota^{2\alpha_\sigma+1}}{m_2}}.$$

Observe that

$$\left|\mathbb{E}_v\left[\mathbf{1}_{\sigma_2(x\cdot v)^2\leq R}\sigma_2(x\cdot v)^2\right] - \mathbb{E}_v\left[\sigma(x\cdot v)^2\right]\right| = \mathbb{E}_v\left[\mathbf{1}_{\sigma_2(x\cdot v)^2>R}\sigma_2(x\cdot v)^2\right]$$
$$\leq \mathbb{P}(\sigma_2(x\cdot v)^2 > R)\mathbb{E}\left[\sigma_2(x\cdot v)^4\right]^{1/2}$$
$$\leq \frac{1}{d}.$$

Therefore with high probability

$$\left|\frac{1}{m_2}\left\|h^{(0)}(x)\right\|^2 - \mathbb{E}_v\left[\sigma(x\cdot v)^2\right]\right| \lesssim \sqrt{\frac{\iota^{2\alpha_\sigma+1}}{m_2}} + \frac{1}{d}.$$

by Lemma 12. Lemma 12 also tells us that $\mathbb{E}_v\left[\sigma_2(x\cdot v)^2\right] = O(1)$. Altogether, for $m_2 \gtrsim \iota^{2\alpha_\sigma+1}$ we have

$$\frac{1}{m_2}\left\|h^{(0)}(x)\right\|^2 \leq 2$$

and hence $\left\|h^{(0)}(x)\right\| \lesssim \sqrt{m_2}$, as desired. $\square$

**Lemma 14.** $\mathbb{P}_{x\sim\nu}\left(|f^*(x)| \geq C_f e\iota^\ell\right) \leq e^{-\iota}$ and $\mathbb{P}_{x\sim\nu}\left(|(\mathbb{K}f^*)(x)| \geq C_K e\iota^\chi \cdot \|\mathbb{K}f^*\|_{L^2}\right) \leq e^{-\iota}$

*Proof.* By Markov's inequality, we have

$$\mathbb{P}(|f^*(x)| > t) = \mathbb{P}(|f^*(x)|^q > t^q) \leq \frac{\|f^*\|_q^q}{t^q} \leq \frac{C_f^q q^{q\ell}}{t^q}.$$

Choose $t = C_f e\iota^\ell$. We select $q = \frac{\iota}{e^{1-1/\ell}}$, which is at least 1 for $C$ in the definition of $\iota$ sufficiently large. Plugging in, we get

$$\mathbb{P}\left(|f^*(x)| > C_f e\iota^\ell\right) \leq \frac{C_f^q \iota^{q\ell}}{C_f^q e^{q\ell}\iota^{q\ell}} = e^{-q\ell} = e^{-\iota\ell e^{1/\ell-1}} \leq e^{-\iota},$$

since $\ell e^{1/\ell-1} \geq 1$.

An analogous derivation for the function $\frac{\mathbb{K}f^*}{\|\mathbb{K}f^*\|_{L^2}}$ yields the second bound $\square$

**Corollary 1.** *With high probability,* $\sup_{x\in\mathcal{D}_2}|f^*(x)| \leq C_f e \cdot \iota^\ell$ *and* $\sup_{x\in\mathcal{D}_2}|(\mathbb{K}f^*)(x)| \lesssim C_K e \cdot \iota^\chi\|\mathbb{K}f^*\|_{L^2}$

*Proof.* Union bounding the previous lemma over $x \in \mathcal{D}_2$ yields the desired result. $\square$

**Lemma 15.** *With high probability,*

$$\left\|\frac{1}{n}\sum_{i=1}^n f^*(x_i)h^{(0)}(x_i) - \mathbb{E}_x\left[f^*(x)h^{(0)}(x)\right]\right\|_2 \lesssim \sqrt{\frac{m_2\iota^{2\ell+2\alpha_\sigma+1}}{n}}$$

*Proof.* Consider the quantity

$$\left| \frac{1}{n} \sum_{i=1}^{n} f^*(x_i) \sigma_2(x_i \cdot v_j) - \mathbb{E}_x \left[ f^*(x) h^{(0)}(x_i) \right] \right|.$$

With high probability, $\sup_{x \in \mathcal{D}_2, j \in [m_2]} |f^*(x_i) \sigma_2(x_i \cdot v_j)| \lesssim \iota^{\ell + \alpha_\sigma}$. Pick truncation radius $R = O(\iota^{\ell + \alpha_\sigma})$. By Hoeffding, we have with high probability.

$$\left| \frac{1}{n} \sum_{i=1}^{n} f^*(x_i) \sigma_2(x_i \cdot v_j) \mathbf{1}_{f^*(x_i)\sigma_2(x_i \cdot v_j) \leq R} - \mathbb{E}_x \left[ f^*(x) \sigma_2(x_i \cdot v_j) \mathbf{1}_{f^*(x_i)\sigma_2(x_i \cdot v_j) \leq R} \right] \right| \lesssim \sqrt{\frac{\iota^{2\ell + 2\alpha_\sigma + 1}}{n}}.$$

Furthermore, note that

$$\left| \mathbb{E}_x \left[ f^*(x) \sigma_2(x \cdot v_j) \mathbf{1}_{f^*(x_i)\sigma_2(x \cdot v_j) \leq R} \right] - \mathbb{E}_x [f^*(x) \sigma_2(x_i \cdot v_j)] \right|$$
$$= \left| \mathbb{E} \left[ \mathbf{1}_{f^*(x)\sigma_2(x \cdot v_j) > R} f^*(x) \sigma_2(x_i \cdot v_j) \right] \right|$$
$$\leq \mathbb{P}(f^*(x) \sigma_2(x \cdot v_j) > R) \mathbb{E}_x \left[ \sigma_2(x \cdot v_j)^4 \right]$$
$$\lesssim \frac{1}{n}.$$

Thus

$$\left| \frac{1}{n} \sum_{i=1}^{n} f^*(x_i) \sigma_2(x_i \cdot v_j) - \mathbb{E}_x \left[ f^*(x) h^{(0)}(x_i) \right] \right| \lesssim \sqrt{\frac{\iota^{2\ell + 2\alpha_\sigma + 1}}{n}}$$

By a union bound, the above holds with high probability for all $j \in [m_2]$, and thus

$$\left\| \frac{1}{n} \sum_{i=1}^{n} f^*(x_i) h^{(0)}(x_i) - \mathbb{E}_x \left[ f^*(x) h^{(0)}(x) \right] \right\|_2 \lesssim \sqrt{\frac{m_2 \iota^{2\ell + 2\alpha_\sigma + 1}}{n}}$$

$\square$

**Lemma 16.** *With high probability,*

$$\sup_{x \in \mathcal{X}_d} \left| \frac{1}{m_2} \sum_{i=1}^{m_2} \mathbb{E}_{x'}[f^*(x')\sigma_2(x' \cdot v_j)\sigma_2(x \cdot v_j)] - \mathbb{E}_{x',v}[f^*(x)\sigma_2(x' \cdot v)\sigma_2(x \cdot v)] \right| \lesssim \sqrt{\frac{\iota^{2\alpha_\sigma + 1}}{m_2}}$$

*Proof.* Fix $x \in \mathcal{D}_2$. Consider the random variables $Z(v) = \mathbb{E}_{x'}[f^*(x')\sigma_2(x' \cdot v)\sigma_2(x \cdot v)]$. With high probability we have $|\sigma_2(x \cdot v_j)| \lesssim \iota^{\alpha_\sigma}$ and thus

$$|Z(v_j)| = |\mathbb{E}_{x'}[f^*(x')\sigma_2(x' \cdot v_j)\sigma_2(x \cdot v_j)]| \lesssim \iota^{\alpha_\sigma} \cdot \mathbb{E}_{x'} \left[ f^*(x')^2 \right]^{1/2} \mathbb{E} \left[ \sigma_2(x' \cdot v_j)^2 \right]^2 \lesssim \iota^{\alpha_\sigma}.$$

For all $j \in [m_2]$. Choosing truncation radius $R = \iota^{\alpha_\sigma}$, with Hoeffding we have with high probability that

$$\left| \frac{1}{m_2} \sum_{i=1}^{m_2} \mathbb{E}_{x'}[f^*(x')\sigma_2(x' \cdot v_j)]\sigma_2(x \cdot v_j)\mathbf{1}_{|Z(v_j)| \leq R} - \mathbb{E}_{x',v} \left[ f^*(x')\sigma_2(x' \cdot v)\sigma_2(x \cdot v)\mathbf{1}_{|Z(v)| \leq R} \right] \right| \lesssim \sqrt{\frac{\iota^{2\alpha_\sigma + 1}}{m_2}}$$

Next, we have that

$$\left| \mathbb{E}_{x',v} \left[ f^*(x')\sigma_2(x' \cdot v)\sigma_2(x \cdot v)\mathbf{1}_{|Z(v)| \leq R} \right] - \mathbb{E}_{x',v}[f^*(x')\sigma_2(x' \cdot v)\sigma_2(x \cdot v)] \right|$$
$$= \left| \mathbb{E}_{x',v} \left[ f^*(x')\sigma_2(x' \cdot v)\sigma_2(x \cdot v)\mathbf{1}_{|Z(v)| > R} \right] \right|$$
$$\leq \mathbb{P}(Z(v) > R) \cdot \mathbb{E}_{x',v} \left[ f^*(x')^2 \sigma_2(x' \cdot v)^2 \sigma_2(x \cdot v)^2 \right]^{1/2}$$
$$\lesssim \frac{1}{m_2}.$$

Conditioning on the high probability event that $\sup_{j \in [m_2]} |Z(v_j)| \leq R$, we have with high probability that

$$\left| \frac{1}{m_2} \sum_{i=1}^{m_2} \mathbb{E}_{x'}[f^*(x')\sigma_2(x' \cdot v_j)]\sigma_2(x \cdot v_j) - \mathbb{E}_{x',v}[f^*(x')\sigma_2(x' \cdot v)\sigma_2(x \cdot v)] \right| \lesssim \sqrt{\frac{\iota^{2\alpha_\sigma + 1}}{m_2}}.$$

A union bound over $x \in \mathcal{D}_2$ yields the desired result. $\square$

**Lemma 17.** *With high probability,*

$$\sup_{x \in \mathcal{D}_2} \left| \frac{1}{m_1} \sum_{i=1}^{m_1} v(a_i, b_i) \sigma_1(a_i \bar{\eta} \phi(x) + b_i) \mathbf{1}_{b_i > 0} - f_v^\infty(x) \right| \lesssim \sqrt{\frac{\|v\|_\infty^2 \iota}{m_1}}$$

*Proof.* Condition on the high probability event $\sup_{x \in \mathcal{D}_2} |\bar{\eta} \phi(x)| \leq 1$. Next, note that whenever $b > 2$ that $v(a, b) = 0$. Therefore we can bound

$$|v(a, b) \sigma_1(a \bar{\eta} \phi(x) + b_i) \mathbf{1}_{b_i > 0}| \leq 2\|v\|_\infty$$

Therefore by Hoeffding's inequality we have that

$$\left| \frac{1}{m_1} \sum_{i=1}^{m_1} v(a_i, b_i) \sigma_1(a_i \bar{\eta} \phi(x) + b_i) \mathbf{1}_{b_i > 0} - f_v^\infty(x) \right| \lesssim \sqrt{\frac{\|v\|_\infty^2 \iota}{m_1}} +$$

The desired result follows via a Union bound over $x \in \mathcal{D}_2$. $\square$

**Lemma 18.** *With high probability,*

$$\left| \frac{1}{m_1} \sum_{i=1}^{m_1} v(a_i, b_i)^2 - \|v\|_{L^2}^2 \right| \lesssim \sqrt{\frac{\|v\|_\infty^4 \iota}{m_1}}.$$

*Proof.* Note that

$$v(a_i, b_i)^2 \leq \|v\|_\infty^2$$

Thus by Hoeffding's inequality we have that

$$\left| \frac{1}{m_1} \sum_{i=1}^{m_1} v(a_i, b_i)^2 - \|v\|_{L^2}^2 \right| \lesssim \sqrt{\frac{\|v\|_\infty^4 \iota}{m_1}}.$$

$\square$

**Corollary 2.** *Let $m_1 \gtrsim \iota$. Then with high probability*

$$\sum_{i=1}^{m_i} v(a_i, b_i)^2 \lesssim \|v\|_\infty^2 m_1.$$

*Proof.* By the previous lemma, we have that

$$\left| \frac{1}{m_1} \sum_{i=1}^{m_1} v(a_i, b_i)^2 - \|v\|_{L^2}^2 \right| \lesssim \sqrt{\frac{\|v\|_\infty^4 \iota}{m_1}} \lesssim \|v\|_\infty^2.$$

Thus

$$\sum_{i=1}^{m_i} v(a_i, b_i)^2 \lesssim \|v\|_{L^2}^2 m_1 + \|v\|_\infty^2 m_1 \lesssim \|v\|_\infty^2 m_1.$$

$\square$

**Lemma 19.** *With high probability,*

$$\frac{1}{n} \sum_{x \in \mathcal{D}_2} (q(\bar{\eta}(\mathbb{K}f^*)(x)) - f^*(x))^2 \lesssim \|q \circ (\bar{\eta}(\mathbb{K}f^*)) - f^*\|_{L^2}^2 + \frac{\|q\|_\infty^2 \iota + \iota^{2\ell+1}}{n}.$$

*Proof.* Let $S$ be the set of $x$ so that $|\overline{\eta}(\mathbb{K}f^*)(x)| \leq 1$ and $|f^*(x)| \lesssim \iota^\ell$. Consider the random variables $(q(\overline{\eta}(\mathbb{K}f^*)(x)) - f^*(x))^2 \cdot \mathbf{1}_{x \in S}$. We have that

$$\left| (q(\overline{\eta}(\mathbb{K}f^*)(x)) - f^*(x))^2 \cdot \mathbf{1}_{x \in S} \right| \lesssim \sup_{z \in [-1,1]} |q(z)|^2 + \iota^{2\ell}.$$

and

$$\mathbb{E}\left[ (q(\overline{\eta}(\mathbb{K}f^*)(x)) - f^*(x))^2 \cdot \mathbf{1}_{x \in S} \right]^2 \lesssim \left( \sup_{z \in [-1,1]} |q(z)|^2 + \iota^{2\ell} \right) \cdot \|q \circ (\overline{\eta}(\mathbb{K}f^*)) - f^*\|_{L^2}^2.$$

Therefore by Berstein's inequality we have that

$$\left| \frac{1}{n} \sum_{x \in \mathcal{D}_2} (q(\overline{\eta}(\mathbb{K}f^*)(x)) - f^*(x))^2 \mathbf{1}_{x \in S} - \|(q \circ (\overline{\eta}(\mathbb{K}f^*)) - f^*)\mathbf{1}_{x \in S}\|_{L^2}^2 \right|$$

$$\lesssim \sqrt{\frac{\left( \sup_{z \in [-1,1]} |q(z)|^2 + \iota^{2\ell} \right) \cdot \|(q \circ (\overline{\eta}(\mathbb{K}f^*)) - f^*)\mathbf{1}_{x \in S}\|_{L^2}^2 \iota}{n}} + \frac{\sup_{z \in [-1,1]} |q(z)|^2 + \iota^{2\ell}}{n} \iota$$

$$\lesssim \|(q \circ (\overline{\eta}(\mathbb{K}f^*)) - f^*)\mathbf{1}_{x \in S}\|_{L^2}^2 + \frac{\sup_{z \in [-1,1]} |q(z)|^2 + \iota^{2\ell}}{n} \iota.$$

Conditioning on the high probability event that $x \in S$ for all $x \in \mathcal{D}_2$, we get that

$$\frac{1}{n} \sum_{x \in \mathcal{D}_2} (q(\overline{\eta}(\mathbb{K}f^*)(x)) - f^*(x))^2 \lesssim \|(q \circ (\overline{\eta}(\mathbb{K}f^*)) - f^*)\mathbf{1}_{x \in S}\|_{L^2}^2 + \frac{\sup_{z \in [-1,1]} |q(z)|^2 + \iota^{2\ell}}{n} \iota$$

$$\leq \|q \circ (\overline{\eta}(\mathbb{K}f^*)) - f^*\|_{L^2}^2 + \frac{\sup_{z \in [-1,1]} |q(z)|^2 \iota + \iota^{2\ell+1}}{n}.$$

$\square$

# E    Proofs for Section 4

## E.1    Single Index Model

*Proof of Theorem 2.* It is easy to see that Assumptions 1 and 3 are satisfied. By assumption $g^*$ is polynomially bounded, i.e there exist constants $C_g, \alpha_g$ such that

$$g^*(z) \leq C_g(1 + |z|)^{\alpha_g} \leq C_g 2^{\alpha_g - 1}(1 + |z|^{\alpha_g}).$$

Therefore

$$\|f^*\|_q = \|g^*\|_{L^q(\mathbb{R}, \mathcal{N}(0,1)} \leq C_g 2^{\alpha_g - 1}(1 + \mathbb{E}_{z \sim \mathcal{N}(0,1)} [|z|^{\alpha_g q}]^{1/q}) \leq C_g 2^{\alpha_g} \alpha_g^{\alpha_g/2} q^{\alpha_g/2}.$$

Thus Assumption 2 is satisfied with $C_f = C_g 2^{\alpha_g} \alpha_g^{\alpha_g/2}$ and $\ell = \alpha_g/2$.

Next, we see that

$$K(x, x') = \mathbb{E}_v[(x \cdot v)(x' \cdot v)] = \frac{x \cdot x'}{d}$$

Therefore

$$(\mathbb{K}f)(x) = \mathbb{E}_{x'}[f^*(x')x' \cdot x/d] = \frac{1}{d} \mathbb{E}_{x'}[\nabla f^*(x')] \cdot x.$$

Furthermore, we have

$$\mathbb{E}_{x'}[\nabla f^*(x')] = \mathbb{E}_{x'}[w^* g'(w^* \cdot x')] = w^* \mathbb{E}_{z \sim \mathcal{N}(0,1)}[g'(z)].$$

Altogether, letting $c_1 = \mathbb{E}_{z \sim \mathcal{N}(0,1)}[g'(z)] = \Omega(1)$, we have $(\mathbb{K}f^*)(x) = \frac{1}{d} c_1 x^T w^*$. Assumption 4 is thus satisfied with $\chi = 1/2$.

Next, see that $\|\mathbb{K}f^*\|_{L^2} = c_1/d$. We select the test function $q$ to be $q(z) = g(\overline{\eta}^{-1}d/c_1 \cdot z)$, so that

$$q(\overline{\eta}(\mathbb{K}f^*)(x)) = g(x^* \cdot x) = f^*(x).$$

Since $\overline{\eta} = \Theta(d\iota^{-\chi})$, we see that

$$\sup_{z \in [-1,1]} |q(z)| = \sup_{z \in [-\Theta(\iota^\chi), \Theta(\iota^\chi)]} |g(z)| = \mathrm{poly}(\iota)$$

$$\sup_{z \in [-1,1]} |q'(z)| = \overline{\eta}^{-1}d/c_1 \sup_{z \in [-\Theta(\iota^\chi), \Theta(\iota^\chi)]} |g'(z)| = \mathrm{poly}(\iota)$$

$$\sup_{z \in [-1,1]} |q''(z)| = \overline{\eta}^{-2}d^2/c_1^2 \sup_{z \in [-\Theta(\iota^\chi), \Theta(\iota^\chi)]} |g''(z)| = \mathrm{poly}(\iota)$$

Therefore we can bound the population loss as

$$\mathbb{E}_x\left[ \left( f(x; \hat{\theta}) - f^*(x) \right)^2 \right] \lesssim \frac{d^2 \, \mathrm{poly}(\iota)}{\min(n, m_1, m_2)} + \frac{1}{\sqrt{n}}.$$

$\square$

## E.2 Quadratic Feature

Throughout this section, we call $x^T A x$ a degree 2 *spherical harmonic* if $A$ is symmetric, $\mathbb{E}[x^T A x] = 0$, and $\mathbb{E}\left[(x^T A x)^2\right] = 1$. Then, we have that $\mathrm{Tr}(A) = 0$, and also

$$1 = \mathbb{E}[x^{\otimes 4}](A^{\otimes 2}) = 3\chi_2 I^{\tilde{\otimes} 2}(A^{\otimes 2}) = 2\chi_2 \|A\|_F^2 \implies \|A\|_F = \sqrt{\frac{1}{2\chi_2}} = \sqrt{\frac{d+2}{2d}} = \Theta(1).$$

See Appendix F for technical background on spherical harmonics.

Our goal is to prove the following key lemma, which states that the projection of $f^*$ onto degree 2 spherical harmonics is approximately $x^T A x$.

**Lemma 20.** *Let $q$ be a $L$-Lipschitz function with $|q(0)| \leq L$, and let the target $f^*$ be of the form $f^*(x) = q(x^T A x)$, where $x^T A x$ is a spherical harmonic. Let $c_1 = \mathbb{E}_{z \sim \mathcal{N}(0,1)}[q'(z)]$. Then*

$$\left\| P_2 f^* - c_1 x^T A x \right\|_{L^2} \lesssim L\kappa^{1/6}d^{-1/12}\log d.$$

We defer the proof of this Lemma to Appendix E.2.1.

As a consequence, the learned feature $\mathbb{K}f^*$ is approximately proportional to $x^T A x$.

**Lemma 21.** *Recall $c_1 = \mathbb{E}_{z \sim \mathcal{N}(0,1)}[q'(z)]$. Then*

$$\left\| \mathbb{K}f^* - \lambda_2^2(\sigma)c_1 x^T A x \right\|_{L^2} \lesssim L\kappa^{1/6}d^{-2-1/12}\log d$$

*Proof.* Since $\mathbb{E}[f^*(x)] = 0$, $P_0 f^* = 0$. Next, since $f^*$ is an even function, $P_k f^* = 0$ for $k$ odd. Thus

$$\left\| \mathbb{K}f^* - \lambda_2^2(\sigma)P_2 f^* \right\|_{L^2} \lesssim d^{-4}.$$

Additionally, by Lemma 20 we have that

$$\left\| P_2 f^* - c_1 x^T A x \right\|_{L^2} \lesssim \|T_2 - c_1 \cdot A\|_F \lesssim L\kappa^{1/6}d^{-1/12}\log d.$$

Since $\lambda_2^2(\sigma) = \Theta(d^{-2})$, we have

$$\left\| \mathbb{K}f^* - \lambda_2^2(\sigma)c_1 x^T A x \right\|_{L^2} \lesssim L\kappa^{1/6}d^{-2-1/12}\log d.$$

$\square$

**Corollary 3.** *Assume $\kappa = o(\sqrt{d})$. Then*

$$\left\| x^T A x - \|\mathbb{K}f^*\|_{L^2}^{-1}\mathbb{K}f^* \right\|_{L^2} \lesssim L\kappa^{1/6}d^{-1/12}\log d$$

*Proof.*

$$\left\| x^T A x - \| \mathbb{K} f^* \|_{L^2}^{-1} \mathbb{K} f^* \right\|_{L^2}$$

$$= \| \mathbb{K} f^* \|_{L^2}^{-1} \left\| x^T A x \| \mathbb{K} f^* \|_{L^2} - \mathbb{K} f^* \right\|_{L^2}$$

$$\leq \| \mathbb{K} f^* \|_{L^2}^{-1} \left\| \mathbb{K} f^* - \lambda_2^2(\sigma) c_1 x^T A x \right\|_{L^2} + \| \mathbb{K} f^* \|_{L^2}^{-1} \left| \| \mathbb{K} f^* \| - \lambda_2^2(\sigma) |c_1| \right|$$

$$\lesssim L \kappa^{1/6} d^{-2-1/12} \log d \| \mathbb{K} f^* \|_{L^2}^{-1}$$

$$\lesssim L \kappa^{1/6} d^{-1/12} \log d.$$

$$\square$$

*Proof of Theorem 3.* By our choice of $\nu$, we see that Assumption 1 is satisfied. We next verify Assumption 2. Since $f^*$ is 1-Lipschitz, we can bound $|g^*(z)| \leq |g^*(0)| + |z|$, and thus

$$\mathbb{E}_x \left[ g^*(x^T A x)^q \right]^{1/q} \leq |g^*(0)| + \mathbb{E}_x \left[ \left| x^T A x \right|^q \right]^{1/q}$$

$$\leq |g^*(0)| + q$$

$$\leq (1 + g^*(0)) q,$$

where we used Lemma 35. Thus Assumption 2 holds with $\ell = 1$.

Finally, we have

$$\mathbb{K} f^* = \sum_{k \geq 2} \lambda_k^2(\sigma_2) P_k f^*.$$

By Lemma 21 we have $\| \mathbb{K} f^* \|_{L^2} \geq \frac{1}{2} \lambda_2^2(\sigma) c_1$ for $d$ larger than some absolute constant. Next, by Lemma 35 we have for any $q \leq \frac{1}{4} d^2$

$$\| \mathbb{K} f^* \|_q \leq \sum_{k \geq 2} \lambda_k^2(\sigma_2) \| P_k f^* \|_q$$

$$\lesssim \sum_{k \geq 2} d^{-k} q^{k/2} \| P_k f^* \|_{L^2}$$

$$\lesssim \sum_{k \geq 2} (\sqrt{q}/d)^k$$

$$= \frac{q}{d^2} \cdot \frac{1}{1 - \sqrt{q} d}$$

$$\leq 2 q d^{-2}.$$

Therefore

$$\| \mathbb{K} f^* \|_q \lesssim \frac{4}{d^2 \lambda_2^2(\sigma_2) c_1} q \| \mathbb{K} f^* \|_{L^2} \lesssim q \| \mathbb{K} f^* \|_{L^2},$$

since $\lambda_2^2(\sigma_2) = \Omega(d^{-2})$. Thus Assumption 4 holds with $\ell = 1$.

Next, observe that $\| \mathbb{K} f^* \|_{L^2} \lesssim d^{-2}$. We select the test function $q$ to be $q(z) = g^*(\overline{\eta}^{-1} \| \mathbb{K} f^* \|_{L^2}^{-1} \cdot z)$. We see that

$$q(\overline{\eta}(\mathbb{K} f^*)(x)) = g^* \left( \| \mathbb{K} f^* \|_{L^2}^{-1} (\mathbb{K} f^*)(x) \right),$$

and thus

$$\| f^* - q(\overline{\eta}(\mathbb{K} f^*)(x)) \|_{L^2} = \left\| g^*(x^T A x) - g^* \left( \| \mathbb{K} f^* \|_{L^2}^{-1} (\mathbb{K} f^*)(x) \right) \right\|_{L^2}$$

$$\lesssim \left\| x^T A x - \| \mathbb{K} f^* \|_{L^2}^{-1} \mathbb{K} f^* \right\|_{L^2}$$

$$\lesssim \kappa^{1/6} d^{-1/12} \log d,$$

where the first inequality follows from Lipschitzness of $g^*$, and the second inequality is Corollary 3. Furthermore since $\overline{\eta} = \Theta(\|\mathbb{K}f^*\|_{L^2}^{-1}\iota^{-\chi})$, we get that $\overline{\eta}^{-1}\|\mathbb{K}f^*\|_{L^2}^{-1} = \Theta(\iota^{\chi})$, and thus

$$\sup_{z\in[-1,1]} |q(z)| = \sup_{z\in[-\Theta(\iota^{\chi}),\Theta(\iota^{\chi})]} |g^*(z)| = \text{poly}(\iota)$$

$$\sup_{z\in[-1,1]} |q'(z)| = \overline{\eta}^{-1}\|\mathbb{K}f^*\|_{L^2}^{-1} \sup_{z\in[-\Theta(\iota^{\chi}),\Theta(\iota^{\chi})]} |(g^*)'(z)| = \text{poly}(\iota)$$

$$\sup_{z\in[-1,1]} |q''(z)| = \left(\overline{\eta}^{-1}\|\mathbb{K}f^*\|_{L^2}^{-1}\right)^2 \sup_{z\in[-\Theta(\iota^{\chi}),\Theta(\iota^{\chi})]} |(g^*)''(z)| = \text{poly}(\iota)$$

Therefore by Theorem 6 we can bound the population loss as

$$\mathbb{E}_x\left|f(x;\hat{\theta}) - f^*(x)\right| \lesssim \left(\frac{d^4\,\text{poly}(\iota)}{\min(n, m_1, m_2)} + \frac{1}{\sqrt{n}} + \kappa^{1/3}d^{-1/6}\iota\right).$$

$\square$

### E.2.1 Proof of Lemma 20

The high level sketch of the proof of Lemma 20 is as follows. Consider a second spherical harmonic $x^T Bx$ satisfying $\mathbb{E}[(x^T Ax)(x^T Bx)] = 0$ (a simple computation shows that this is equivalent to $\text{Tr}(AB) = 0$). We appeal to a key result in universality to show that in the large $d$ limit, the distribution of $x^T Ax$ converges to a standard Gaussian; additionally, $x^T Bx$ converges to an independent mean-zero random variable. As a consequence, we show that

$$\mathbb{E}[q(x^T Ax)x^T Ax] \approx \mathbb{E}_{z\sim\mathcal{N}(0,1)}[q(z)z] = \mathbb{E}_{z\sim\mathcal{N}(0,1)}[q'(z)] = c_1.$$

and

$$\mathbb{E}[q(x^T Ax)x^T Bx] \approx \mathbb{E}[q(x^T Ax)] \cdot \mathbb{E}[x^T Bx] = 0.$$

From this, it immediately follows that $P_2 f^* \approx c_1 x^T Ax$.

The key universality theorem is the following.

**Definition 7.** *For two probability measures $\mu, \nu$, the* Wasserstein 1-distance *between $\mu$ and $\nu$ is defined as*

$$W_1(\mu,\nu) := \sup_{\|f\|_{Lip}\leq 1} |\mathbb{E}_{z\sim\mu}[f(z)] - \mathbb{E}_{z\sim\nu}[f(z)]|,$$

*where $\|f\|_{Lip} := \sup_{x\neq y}\frac{|f(x)-f(y)|}{\|x-y\|_2}$ is the Lipschitz norm of $f$.*

**Lemma 22** ([52][Theorem 9.20]). *] Let $z \sim \mathcal{N}(0, I_d)$ be a standard Gaussian vector, and let $f : \mathbb{R}^d \to \mathbb{R}$ satisfy $\mathbb{E}[f(z)] = 0, \mathbb{E}[(f(z))^2] = 1$. Then*

$$W_1(\text{Law}(f(z)), \mathcal{N}(0,1)) \lesssim \mathbb{E}\left[\|\nabla f(z)\|^4\right]^{1/4} \mathbb{E}\left[\|\nabla^2 f(z)\|_{op}^4\right]^{1/4},$$

*where $W_1$ is the Wasserstein 1-distance.*

We next apply this lemma to show that the quantities $x^T Ax + x^T Bx$ and $x^T Ax + x \cdot u$ are approximately Gaussian, given appropriate operator norm bounds on $A, B$.

**Lemma 23.** *Let $x^T Ax$ and $x^T Bx$ be orthogonal spherical harmonics. Then, for constants $c_1, c_2$ with $c_1^2 + c_2^2 = 1$, we have that the random variable $Y = c_1 x^T Ax + c_2 x^T Bx$ satisfies*

$$W_1(Y, \mathcal{N}(0,1)) \lesssim \|A\|_{op} + \|B\|_{op}.$$

*Proof.* Define the function $f(z) = c_1 d\frac{z^T Az}{\|z\|^2} + c_2 d\frac{z^T Bz}{\|z\|^2}$, and let $x = \frac{z\sqrt{d}}{\|z\|}$. Observe that when $z \sim \mathcal{N}(0, I)$, we have $x \sim \text{Unif}(\mathcal{S}^{d-1}(\sqrt{d}))$. Therefore $f(z)$ is equal in distribution to $Y$. Define $f_1(z) = d\frac{z^T Az}{\|z\|^2}$. We compute

$$\nabla f_1(z) = 2d\left(\frac{Az}{\|z\|^2} - \frac{z^T Az \cdot z}{\|z\|^4}\right)$$

and

$$\nabla^2 f_1(z) = 2d\left(\frac{A}{\|z\|^2} - \frac{2Azz^T}{\|z\|^4} - \frac{2zz^T A}{\|z\|^4} - 2\frac{z^T Az}{\|z\|^4}I + 4\frac{z^T Azzz^T}{\|z\|^6}\right).$$

Thus

$$\|\nabla f_1(z)\| \le 2d\left(\frac{\|Az\|}{\|z\|^2} + \frac{|z^T Az|}{\|z\|^3}\right) \le \frac{\sqrt{d}}{\|z\|} \cdot \|Ax\| + \frac{|x^T Ax|}{\|z\|}.$$

and

$$\left\|\nabla^2 f_1(z)\right\|_{op} \lesssim \frac{d}{\|z\|^2}\|A\|_{op}.$$

$\|z\|^2$ is distributed as a chi-squared random variable with $d$ degrees of freedom, and thus

$$\mathbb{E}\left[\|z\|^{-2k}\right] = \frac{1}{\prod_{j=1}^{k}(d-2j)}$$

Therefore

$$\mathbb{E}\left[\left\|\nabla^2 f_1(z)\right\|_{op}^4\right]^{1/4} \lesssim d\|A\|_{op}\mathbb{E}\left[\|z\|^{-8}\right]^{1/4} \lesssim \|A\|_{op}.$$

and, using the fact that $x$ and $\|z\|$ are independent,

$$\mathbb{E}\left[\|\nabla f_1(z)\|^4\right]^{1/4} \lesssim \sqrt{d}\mathbb{E}\left[\|z\|^{-4}\right]^{1/4}\mathbb{E}\left[\|Ax\|^4\right]^{1/4} + \mathbb{E}\left[\|z\|^{-4}\right]^{1/4}\mathbb{E}\left[(x^T Ax)^4\right]^{1/4} \lesssim 1.$$

As a consequence, we have

$$\mathbb{E}\left[\|\nabla f(z)\|^4\right]^{1/4} \lesssim 1 \quad \text{and} \quad \mathbb{E}\left[\left\|\nabla^2 f(z)\right\|_{op}^4\right]^{1/4} \lesssim \|A\|_{op} + \|B\|_{op}.$$

Thus by Lemma 22 we have

$$W_1(Y, \mathcal{N}(0,1)) = W_1(f(z), \mathcal{N}(0,1)) \lesssim \|A\|_{op} + \|B\|_{op}.$$

$\square$

**Lemma 24.** *Let $x^T Ax$ be a spherical harmonic, and $\|u\| = 1$. Then, for constants $c_1, c_2$ with $c_1^2 + c_2^2 = 1$, we have that the random variable $Y = c_1 x^T Ax + c_2 x^T u$ satisfies*

$$W_1(Y, \mathcal{N}(0,1)) \lesssim \|A\|_{op}.$$

*where $W_1$ is the 1-Wasserstein distance.*

*Proof.* Define $f_2(z) = \frac{\sqrt{d}z^T u}{\|z\|}$. We have

$$\nabla f_2(z) = \sqrt{d}\left(\frac{u}{\|z\|} - \frac{zz^T u}{\|z\|^3}\right).$$

and

$$\nabla^2 f_2(z) = \sqrt{d}\left(-\frac{uz^T + zu^T}{\|z\|^3} - \frac{z^T u}{\|z\|^3}I + 3\frac{z^T uzz^T}{\|z\|^5}\right).$$

Thus

$$\|\nabla f_2(z)\| \lesssim \frac{\sqrt{d}}{\|z\|} \quad \text{and} \quad \left\|\nabla^2 f_2(z)\right\|_{op} \lesssim \frac{\sqrt{d}}{\|z\|^2},$$

so

$$\mathbb{E}\left[\|\nabla f_1(z)\|^4\right]^{1/4} \lesssim 1 \quad \text{and} \quad \mathbb{E}\left[\|\nabla f_1(z)\|^4\right]^{1/4} \lesssim \frac{1}{\sqrt{d}}.$$

We finish using the same argument as above.

$\square$

Lemma 23 implies that, when $\|A\|_{op}, \|B\|_{op}$ are small, $(x^T A x, x^T B x)$ is close in distribution to the standard Gaussian in 2-dimensions. As a consequence, $\mathbb{E}[q(x^T A x) x^T B x] \approx \mathbb{E}[q(z_1)] \mathbb{E}[z_2] = 0$, where $z_1, z_2$ are i.i.d Gaussians. This intuition is made formal in the following lemma.

**Lemma 25.** *Let $x^T A x, x^T B x$ be two orthogonal spherical harmonics. Then*

$$\left| \mathbb{E}\big[ q(x^T A x) x^T B x \big] \right| \leq L\Big( \sqrt{\|A\|_{op} \log d} + \|B\|_{op} \log d \Big)$$

*Proof.* Define the function $F(t) = \int_0^t q(s) ds$. Then $F'(t) = q(t)$, so by a Taylor expansion we get that

$$|F(x + \epsilon y) - F(x) - \epsilon y q(x)| \leq \epsilon^2 y^2 L.$$

Therefore

$$\left| \mathbb{E}\big[ q(x^T A x) x^T B x \big] \right| \leq \epsilon L \mathbb{E}\big[ (x^T B x)^2 \big] + \epsilon^{-1} \left| \mathbb{E}\big[ F(x^T A x + \epsilon x^T B x) \big] - \mathbb{E}\big[ F(x^T A x) \big] \right|.$$

Pick truncation radius $R$, and define the function $\overline{F}(z) = F(\max(-R, \min(R, z)))$. $\overline{F}$ has Lipschitz constant $\sup_{z \in [-R,R]} |q(z)|$, and thus since $W_1\big( x^T A x, \mathcal{N}(0, 1) \big) \lesssim \|A\|_{op}$, we have

$$\left| \mathbb{E}\overline{F}(x^T A x) - \mathbb{E}_{z \sim \mathcal{N}(0,1)} \overline{F}(z) \right| \lesssim \sup_{z \in [-R,R]} |q(z)| \cdot \|A\|_{op}.$$

Next, we have

$$\left| \mathbb{E}\big[ \overline{F}(x^T A x) - F(x^T A x) \big] \right| \leq \mathbb{E}\big[ \mathbf{1}_{|x^T A x| > R} \big| F(x^T A x) \big| \big] \leq \mathbb{P}(|x^T A x| > R) \cdot \mathbb{E}\big[ F(x^T A x)^2 \big]^{1/2}.$$

Likewise,

$$\left| \mathbb{E}_z \big[ \overline{F}(z) - F(z) \big] \right| \leq \mathbb{P}(|z| > R) \cdot \mathbb{E}\big[ F(z)^2 \big]^{1/2}.$$

Since $q$ is $L$-Lipschitz, we can bound $|F(z)| \leq |z||q(0)| + \frac{1}{2} L|z|^2$, and thus

$$\mathbb{E}\big[ F(z)^2 \big] \lesssim L^2 \quad \text{and} \quad \mathbb{E}\big[ F(x^T A x)^2 \big] \lesssim L^2.$$

The standard Gaussian tail bound yields $\mathbb{P}(|z| > R) \lesssim \exp\big(-C_1 R^2\big)$ for appropriate constant $C_1$, and polynomial concentration yields $\mathbb{P}\big(\big|x^T A x\big| > R\big) \lesssim \exp(-C_2 R)$ for appropriate constant $C_2$. Thus choosing $R = C_3 \log d$ for appropriate constant $C_3$, we get that

$$\left| \mathbb{E}\big[ \overline{F}(x^T A x) - F(x^T A x) \big] \right| + \left| \mathbb{E}_z \big[ \overline{F}(z) - F(z) \big] \right| \lesssim \frac{L}{d}.$$

Altogether, since $|q(z)| \leq |q(0)| + L|z|$, we get that

$$\left| \mathbb{E} F(x^T A x) - \mathbb{E}_{z \sim \mathcal{N}(0,1)} F(z) \right| \lesssim \|A\|_{op} L \log d.$$

By an identical calculation, we have that for $\epsilon < 1$,

$$\left| \mathbb{E} F(x^T A x + x^T B x) - \mathbb{E}_{z \sim \mathcal{N}(0,1)} F(z\sqrt{1 + \epsilon^2}) \right| \lesssim \Big( \|A\|_{op} + \epsilon \|B\|_{op} \Big) \cdot L \log d$$

Altogether, we get that

$$\left| \mathbb{E}\big[ F(x^T A x + \epsilon x^T B x) \big] - \mathbb{E}\big[ F(x^T A x) \big] \right|$$
$$\leq \Big( \|A\|_{op} + \epsilon \|B\|_{op} \Big) \cdot L \log d + \left| \mathbb{E}_{z \sim \mathcal{N}(0,1)} F(z) - \mathbb{E}_{z \sim \mathcal{N}(0,1)} F(z\sqrt{1 + \epsilon^2}) \right|.$$

Via a simple calculation, one sees that

$$\left| F(z\sqrt{1 + \epsilon^2}) - F(z) \right| \leq |q(0)||z|\Big( \sqrt{1 + \epsilon^2} - 1 \Big) + \frac{L}{2} z^2 \epsilon^2 \lesssim L|z|\epsilon^2 + Lz^2 \epsilon^2.$$

Therefore

$$\left| \mathbb{E}\big[ F(x^T A x + \epsilon x^T B x) \big] - \mathbb{E}\big[ F(x^T A x) \big] \right| \lesssim L\Big( \Big( \|A\|_{op} + \epsilon \|B\|_{op} \Big) \log d + \epsilon^2 \Big),$$

so

$$\left| \mathbb{E}\big[ q(x^T A x) x^T B x \big] \right| \leq L\Big( \epsilon^{-1} \|A\|_{op} \log d + \epsilon + \|B\|_{op} \log d \Big).$$

Setting $\epsilon = \sqrt{\|A\|_{op} \log d}$ yields

$$\left| \mathbb{E}\big[ q(x^T A x) x^T B x \big] \right| \lesssim L\Big( \sqrt{\|A\|_{op} \log d} + \|B\|_{op} \log d \Big),$$

as desired. $\qquad\square$

Similarly, we use the consequence of Lemma 24 that $(x^T A x, u^T x)$ is close in distribution to a 2d standard Gaussian, and show that $\mathbb{E}[q(x^T A x)((u^T x)^2 - 1)] \approx \mathbb{E}[q(z_1)(z_2^2 - 1)] = 0$.

**Lemma 26.**
$$\left| \mathbb{E}\left[q(x^T A x)(u^T x)^2\right] - \mathbb{E}_z[q(z)] \right| \lesssim L \|A\|_{op}^{1/3} \log^{2/3} d.$$

*Proof.* Let $G(t) = \int_0^t F(s)ds$. Then $G'(t) = F(t), G''(t) = q(t)$, so a Taylor expansion yields
$$G(x + \epsilon y) = G(x) + \epsilon y F(x) + \epsilon^2 y^2 q(x) + O(\epsilon^3 |y|^3 L)$$
$$G(x - \epsilon y) = G(x) - \epsilon y F(x) + \epsilon^2 y^2 q(x) + O(\epsilon^3 |y|^3 L).$$

Thus
$$\epsilon^2 y^2 q(x) = \frac{1}{2}(G(x + \epsilon y) + G(x - \epsilon y) - 2G(x)) + O(\epsilon^3 |y|^3 L).$$

Therefore
$$\left| \mathbb{E}\left[q(x^T A x)(u^T x)^2\right] \right| \lesssim \epsilon L + \epsilon^{-2} \left| \mathbb{E}\left[G(x^T A x + \epsilon u^T x) + G(x^T A x - \epsilon u^T x) - 2G(x^T A x)\right] \right|.$$

For truncation radius $R$, define $\overline{G}(z) = G(\max(-R, \min(R, z)))$. We get that $G$ has Lipschitz constant $\sup_{|z| \leq R} |F(z)| \lesssim LR^2$. Therefore
$$\left| \mathbb{E}\overline{G}(x^T A x) - \mathbb{E}_{z \sim \mathcal{N}(0,1)}\overline{G}(z) \right| \lesssim LR^2 |A|_{op},$$

and by a similar argument in the previous lemma, setting $R = C_3 \log d$ yields
$$\left| \mathbb{E}\left[\overline{G}(x^T A x) - G(x^T A x)\right] \right| + \left| \mathbb{E}_z \overline{G}(z) - G(z) \right| \lesssim \frac{L}{d}.$$

Altogether,
$$\left| \mathbb{E}G(x^T A x) - \mathbb{E}_z G(z) \right| \lesssim \|A\|_{op} L \log^2 d.$$

By an identical calculation,
$$\left| \mathbb{E}G(x^T A x \pm \epsilon u^T x) - \mathbb{E}_z G(z\sqrt{1 + \epsilon^2}) \right| \lesssim \|A\|_{op} \cdot L \log^2 d.$$

Additionally, letting $z, w$ be independent standard Gaussians,
$$\epsilon^{-2} \mathbb{E}_z\left[2G(z\sqrt{1 + \epsilon^2} - 2G(z)\right] = \epsilon^{-2} \mathbb{E}_{z,w}[G(z + \epsilon w) + G(z - \epsilon w) - 2G(z)]$$
$$= \mathbb{E}_{z,w}\left[q(z)w^2\right] + O(\epsilon L)$$
$$= \mathbb{E}_z[q(z)] + O(\epsilon L).$$

Altogether,
$$\left| \mathbb{E}\left[q(x^T A x)(u^T x)^2\right] - \mathbb{E}_z[q(z)] \right| \lesssim \epsilon L + \epsilon^{-2} \|A\|_{op} \cdot L \log^2 d$$
$$\lesssim L \|A\|_{op}^{1/3} \log^{2/3} d,$$

where we set $\epsilon = \|A\|_{op}^{1/3} \log^{2/3} d$. $\qquad \square$

Lemma 25 shows that when $\|B\|_{op} \ll 1$, $\mathbb{E}[q(x^T A x)x^T B x] \approx 0$. However, we need to show this is true for all spherical harmonics, even those with $\|B\|_{op} = \Theta(1)$. To accomplish this, we decompose $B$ into the sum of a low rank component and small operator norm component. We use Lemma 25 to bound the small operator norm component, and Lemma 26 to bound the low rank component. Optimizing over the rank threshold yields the following desired result:

**Lemma 27.** *Let $A, B$ be orthogonal spherical harmonics. Then*
$$\left| \mathbb{E}\left[q(x^T A x)x^T B x\right] \right| \leq L \|A\|_{op}^{1/6} \log d.$$

*Proof.* Let $\tau > \|A\|_{op}$ be a threshold to be determined later. Decompose $B$ as follows:

$$B = \sum_{i=1}^{d} \lambda_i u_i u_i^T = \sum_{|\lambda_i|>\tau} \lambda_i \left( u_i u_i^T - \frac{1}{d} I \right) - \frac{1}{\|A\|_F^2} \sum_{|\lambda_i|>\tau} u_i^T A u_i \cdot A + \tilde{B},$$

where

$$\tilde{B} = \sum_{|\lambda_i|\leq\tau} \lambda_i u_i u_i^T + I \cdot \frac{1}{d} \sum_{|\lambda_i|>\tau} \lambda_i + \frac{1}{\|A\|_F^2} \sum_{|\lambda_i|>\tau} \lambda_i u_i^T A u_i \cdot A.$$

By construction, we have,

$$\mathrm{Tr}\left(\tilde{B}\right) = \sum_{|\lambda_i|\leq\tau} \lambda_i + \sum_{|\lambda_i|>\tau} \lambda_i = \sum_{i\in[d]} \lambda_i = 0$$

and

$$\langle \tilde{B}, A \rangle = \sum_{|\lambda_i|\leq\tau} \lambda_i u_i^T A u_i + \sum_{|\lambda_i|>\tau} \lambda_i u_i^T A u_i = \langle A, B \rangle = 0.$$

Therefore by Lemma 25,

$$\left| \mathbb{E}\left[ q(x^T A x) x^T \tilde{B} x \right] \right| \lesssim L \sqrt{\|A\|_{op} \log d} \left\| \tilde{B} \right\|_F + L \left\| \tilde{B} \right\|_{op} \log d.$$

There are at most $O(\tau^{-2})$ indices $i$ satisfying $|\lambda_i| > \tau$, and thus

$$\sum_{|\lambda_i|>\tau} |\lambda_i| \lesssim \sqrt{\tau^{-2} \cdot \sum_{|\lambda_i|>\tau} |\lambda_i|^2} \leq \tau^{-1}.$$

We thus compute that

$$\left\| \tilde{B} \right\|_F^2 \lesssim \sum_{|\lambda_i|\leq\tau} \lambda_i^2 + \frac{1}{d} \left( \sum_{\lambda_i>\tau} \lambda_i \right)^2 + \left| \sum_{|\lambda_i|>\tau} \lambda_i u_i^T A u_i \right|^2$$

$$\lesssim \sum_{|\lambda_i|\leq\tau} \lambda_i^2 + \|A\|_{op}^2 \left| \sum_{|\lambda_i|>\tau} \lambda_i \right|^2$$

$$\lesssim 1 + \tau^{-2} \|A\|_{op}^2$$

$$\lesssim 1.$$

and

$$\left\| \tilde{B} \right\|_{op} \leq \tau + \left( \frac{1}{d} + \|A\|_{op} \right) \left| \sum_{|\lambda_i|>\tau} \lambda_i u_i^T A u_i \right|$$

$$\lesssim \tau + \|A\|_{op}^2 \tau^{-1}.$$

Next, since $\left| \mathbb{E}\left[ q(x^T A x) \right] - \mathbb{E}_z[q(x)] \right| \lesssim L\|A\|_{op}$, Lemma 26 yields

$$\left| \mathbb{E}\left[ q(x^T A x) \cdot \sum_{|\lambda_i|>\tau} \lambda_i \left( u_i u_i^T - \frac{1}{d} I \right) \right] \right| \leq \sum_{|\lambda_i|>\tau} |\lambda_i| \left| \mathbb{E}\left[ q(x^T A x)(u_i^T x)^2 \right] - \mathbb{E}\left[ q(x^T A x) \right] \right|$$

$$\lesssim \sum_{|\lambda_i|>\tau} |\lambda_i| \cdot L\|A\|_{op}^{1/3} \log^{2/3} d$$

$$\lesssim L\tau^{-1} \|A\|_{op}^{1/3} \log^{2/3} d.$$

Finally,

$$\left| \mathbb{E}\left[ q(x^T A x) \cdot \frac{1}{\|A\|_F^2} \sum_{|\lambda_i| > \tau} \lambda_i u_i^T A u_i \cdot x^T A x \right] \right| \lesssim L \left| \sum_{|\lambda_i| > \tau} \lambda_i u_i^T A u_i \right| \lesssim L\|A\|_{op} \tau^{-1}.$$

Altogether,

$$\left| \mathbb{E}\left[ q(x^T A x) x^T B x \right] \right| \lesssim L \log d \left( \|A\|_{op}^{1/2} + \tau + \|A\|_{op}^2 \tau^{-1} + \tau^{-1} \|A\|_{op}^{1/3} + \|A\|_{op} \tau^{-1} \right) \lesssim L\|A\|_{op}^{1/6} \log d.$$

where we set $\tau = \|A\|_{op}^{1/6}$. $\qquad\square$

Finally, we use the fact that $x^T A x$ is approximately Gaussian to show that $\mathbb{E}[q(x^T A x)x^T A x] \approx c_1$.

**Lemma 28.** *Let $x^T A x$ be a spherical harmonic. Then*

$$\left| \mathbb{E}[q(x^T A x)x^T A x] - c_1 \right| \lesssim L\|A\|_{op} \log d$$

*Proof.* Define $H(z) = q(z)z$. For truncation radius $R$, define $\overline{H}(z) = H(\max(-R, \min(R, z)))$. For $x, y \in [-R, R]$ we can bound

$$\begin{aligned}
|H(x) - H(y)| &= |q(x)x - q(y)y| \\
&\leq |x||q(x) - q(y)| + |q(y)|\|x - y\| \\
&\lesssim RL\|x - y\|.
\end{aligned}$$

Thus $\overline{H}$ has Lipschitz constant $O(RL)$. Since $W_1(x^T A x, \mathcal{N}(0,1)) \lesssim \|A\|_{op}$, we have

$$\left| \mathbb{E}\,\overline{H}(x^T A x) - \mathbb{E}_{z \sim \mathcal{N}(0,1)}\,\overline{H}(z) \right| \lesssim RL\|A\|_{op}.$$

Furthermore, choosing $R = C \log d$ for appropriate constant $C$, we have that

$$\left| \mathbb{E}_x[\overline{H}(x^T A x) - H(x^T A x)] \right| \leq \mathbb{P}\big(|x^T A x| > R\big) \cdot \mathbb{E}[H(x^T A x)^2]^{1/2} \lesssim \frac{L}{d}$$

$$\left| \mathbb{E}_z[\overline{H}(z) - H(z)] \right| \leq \mathbb{P}(|z| > R) \cdot \mathbb{E}[H(z)^2]^{1/2} \lesssim \frac{L}{d}.$$

Altogether,

$$\left| \mathbb{E}_x[H(x^T A x)] - \mathbb{E}_{z \sim \mathcal{N}(0,1)}[H(z)] \right| \lesssim L\|A\|_{op} \log d.$$

Substituting $H(x^T A x) = q(x^T A x)x^T A x$ and $\mathbb{E}_{z \sim \mathcal{N}(0,1)}[H(z)] = \mathbb{E}_z[q(z)z] = \mathbb{E}_z[q'(z)] = c_1$ yields the desired bound. $\qquad\square$

We are now set to prove Lemma 20.

*Proof of Lemma 20.* Let $P_2 f^* = x^T T_2 x$. Then $\left\| P_2 f^* - c_1 x^T A x \right\|_{L^2} \lesssim \|T_2 - c_1 A\|_F$.

Write $T_2 = \alpha A + A^\perp$, where $\langle A^\perp, A \rangle = 0$. We first have that

$$\mathbb{E}\big[f^*(x)x^T A^\perp x\big] = \mathbb{E}\big[x^T T_2 x \cdot x^T A^\perp x\big] = 2\chi_2 \langle T_2, A^\perp \rangle = 2\chi_2 \big\|A^\perp\big\|_F^2.$$

Also, by Lemma 27, we have

$$\mathbb{E}\big[f^*(x)x^T A^\perp x\big] = \mathbb{E}\big[q(x^T A x)x^T A^\perp x\big] \lesssim \big\|A^\perp\big\|_F \cdot L\|A\|_{op}^{1/6} \log d.$$

Therefore $\big\|A^\perp\big\|_F \lesssim L\|A\|_{op}^{1/6} \log d$. Next, see that

$$\mathbb{E}\big[f^*(x)x^T A x\big] = 2\alpha\chi_2\|A\|_F^2 = \alpha,$$

so by Lemma 28 we have $|c_1 - \alpha| \lesssim L\|A\|_{op} \log d$. Altogether,

$$\|T_2 - c_1 A\|_F \leq |\alpha - c_1|\|A\|_F + \big\|A^\perp\big\|_F \lesssim L\|A\|_{op}^{1/6} \log d = L\kappa^{1/6} d^{-1/12} \log d.$$

$\qquad\square$

## E.3 Improved Error Floor for Polynomials

When $q$ is a polynomial of degree $p = O(1)$, we can improve the exponent of $d$ in the error floor.

**Theorem 7.** *Assume that $q$ is a degree $p$ polynomial, where $p = O(1)$. Under Assumption 5, Assumption 6, and Assumption 7, with high probability Algorithm 1 satisfies the population loss bound*

$$\mathbb{E}_x\left[\left(f(x;\hat{\theta}) - f^*(x)\right)^2\right] \lesssim \tilde{O}\left(\frac{d^4}{\min(n, m_1, m_2)} + \frac{1}{\sqrt{n}} + \frac{\kappa^2}{d}\right)$$

The high level strategy to prove Theorem 7 is similar to that for Theorem 3, as we aim to show $\mathbb{K}f^*$ is approximately proportional to $x^T A x$. Rather to passing to universality as in Lemma 20, however, we use an algebraic argument to estimate $P_2 f^*$.

The key algebraic lemma is the following:

**Lemma 29.** *Let $\text{Tr}(A) = 0$. Then*

$$A^{\tilde{\otimes}k}(I^{\otimes k-1}) = \sum_{s=1}^{k} d_{k,s} A^s \cdot A^{\tilde{\otimes}k-s}(I^{\otimes(k-s)}),$$

*where the constants $d_{k,s}$ are defined by*

$$d_{k,s} := 2^{s-1}\frac{(2k - 2s - 1)!!(k-1)!}{(2k-1)!!(k-s)!}$$

*and we denote $(-1)!! = 1$.*

*Proof.* The proof proceeds via a counting argument. We first have that

$$A^{\tilde{\otimes}k}(I^{\otimes k-1}) = \sum_{(\alpha_1,\ldots,\alpha_{k-1})\in[d]^{k-1}} \left(A^{\tilde{\otimes}k}\right)_{\alpha_1,\alpha_1,\ldots,\alpha_{k-1},\alpha_{k-1},i,j}.$$

Consider any permutation $\sigma \in S_{2k}$. We can map this permutation to the graph $G(\sigma)$ on $k$ vertices and $k-1$ edges as follows: for $m \in [k-1]$, if $\sigma^{-1}(2m-1) \in \{2a-1, 2a\}$ and $\sigma^{-1}(2m) \in \{2b-1, 2b\}$, then we draw an edge $e(m)$ between $a$ and $b$. In the resulting graph $G(\sigma)$ each node has degree at most 2, and hence there are either two vertices with degree 1 or one vertex with degree 0. For a vertex $v$, let $e_1(v), e_2(v) \in [k-1]$ be the two edges $v$ is incident to if $v$ has degree 2, and otherwise $e_1(v)$ be the only edge $v$ is incident to. For shorthand, let $(i_1, \ldots, i_{2k}) = (\alpha_1, \alpha_1, \ldots, \alpha_{k-1}, \alpha_{k-1}, i, j)$.

If there are two vertices $(u_1, u_2)$ with degree 1, we have that

$$\left(A^{\otimes k}\right)_{i_{\sigma(1)}, i_{\sigma(2)}, \cdots i_{\sigma(2k)}} = A_{i,\alpha_{e_1(u_1)}} A_{j,\alpha_{e_2(u_2)}} \prod_{v \neq u_1, u_2} A_{\alpha_{e_1(v)}, \alpha_{e_2(v)}}$$

Let $u_1, u_2$ be connected to eachother via a path of $s$ total vertices, and let $\mathcal{P}$ be the ordered set of vertices in this path. Via the matrix multiplication formula, one sees that

$$\sum_{(\alpha_1,\ldots,\alpha_{k-1})\in[d]^{k-1}} \left(A^{\otimes k}\right)_{i_{\sigma(1)}, i_{\sigma(2)}, \cdots i_{\sigma(2k)}} = (A^s)_{i,j} \sum \prod_{v \notin \mathcal{P}} A_{\alpha_{e_1(v)}, \alpha_{e_2(v)}},$$

where the sum is over the $k - s$ $\alpha$'s that are still remaining in $\{e_i(v) : v \notin \mathcal{P}\}$

Likewise, if there is one vertex $u_1$ with degree 0, we have

$$\left(A^{\otimes k}\right)_{i_{\sigma(1)}, i_{\sigma(2)}, \cdots i_{\sigma(2k)}} = A_{i,j} \prod_{v \neq u_1} A_{\alpha_{e_1(v)}, \alpha_{e_2(v)}}.$$

and thus, since $\mathcal{P} = \{u_1\}$

$$\sum_{(\alpha_1,\ldots,\alpha_{k-1})\in[d]^{k-1}} \left(A^{\otimes k}\right)_{i_{\sigma(1)}, i_{\sigma(2)}, \cdots i_{\sigma(2k)}} = A_{i,j} \sum_{(\alpha_1,\ldots,\alpha_{k-1})} \prod_{v \notin \mathcal{P}} A_{\alpha_{e_1(v)}, \alpha_{e_2(v)}}.$$

Altogether, we have that

$$A^{\tilde{\otimes} k}(I^{\otimes k-1}) = \frac{1}{(2k)!} \sum_{\sigma \in S_{2k}} A^s \sum \prod_{v \notin \mathcal{P}} A_{\alpha_{e_1(v)}, \alpha_{e_2(v)}}$$

where $s, \mathcal{P}$ are defined based on the graph $\mathcal{G}(\sigma)$. Consider a graph with fixed path $\mathcal{P}$, and let $S_{\mathcal{P}}$ be the set of permutations which give rise to the path $\mathcal{P}$. We have that

$$A^{\tilde{\otimes} k}(I^{\otimes k-1}) = \frac{1}{(2k)!} \sum_{\mathcal{P}} A^s \sum_{\sigma \in S_{\mathcal{P}}} \sum \prod_{v \notin \mathcal{P}} A_{\alpha_{e_1(v)}, \alpha_{e_2(v)}}.$$

There are $(k-1) \cdots (k-s+1)$ choices for the $k$ edges to use in the path, and at each vertex $v$ there are two choices for which edge should correspond to $2v$ or $2v+1$. Additionally, there are 2 ways to orient each edge. Furthermore, there are $\frac{k!}{(k-s)!}$ ways to choose the ordering of the path. Altogether, there are $2^{2s-1} \frac{(k-1)!}{(k-s)!} \frac{k!}{(k-s)!}$ ways to construct a path of length $s$. We can thus write

$$A^{\tilde{\otimes} k}(I^{\otimes k-1}) = \frac{1}{(2k)!} \sum_s A^s 2^{2s-1} \frac{(k-1)!}{(k-s)!} \frac{k!}{(k-s)!} \sum \prod_{v \notin \mathcal{P}} A_{\alpha_{e_1(v)}, \alpha_{e_2(v)}},$$

where this latter sum is over all permutations where the mapping corresponding to vertices not on the path have not been decided, along with the sum over the unused $\alpha$'s. Reindexing, this latter sum is (letting $(i_1, \ldots, i_{2k-2s}) = (\alpha_1, \alpha_1, \ldots, \alpha_{k-s}, \alpha_{k-s})$)

$$\sum_{\sigma \in S_{2k-2s}} \sum_{(\alpha_1, \ldots, \alpha_{k-s}) \in [d]^{k-s}} \prod A_{i_{\sigma(2j-1)}, i_{\sigma(2j)}} = (2k-2s)! A^{\tilde{\otimes} k-s}(I^{\otimes k-s})$$

Altogether, we obtain

$$\begin{aligned}
A^{\tilde{\otimes} k}(I^{\otimes k-1}) &= \sum_{s \geq 1} \frac{(2k-2s)!}{(2k)!} \frac{(k-1)!}{(k-s)!} \frac{k!}{(k-s)!} 2^{2s-1} \cdot A^s \cdot A^{\tilde{\otimes} k-s}(I^{\otimes k-s}) \\
&= \sum_{s \geq 1} \frac{k!}{(2k)!} \frac{(2k-2s)!}{(k-s)!} \frac{(k-1)!}{(k-s)!} \cdot A^s \cdot A^{\tilde{\otimes} k-s}(I^{\otimes k-s}) \\
&= \sum_{s \geq 1} \frac{(2k-2s-1)!!}{(2k-1)!! 2^s} \frac{(k-1)!}{(k-s)!} 2^{2s-1} \cdot A^s \cdot A^{\tilde{\otimes} k-s}(I^{\otimes k-s}) \\
&= \sum_{s \geq 1} d_{k,s} \cdot A^s \cdot A^{\tilde{\otimes} k-s}(I^{\otimes k-s}),
\end{aligned}$$

as desired. $\qquad \square$

**Definition 8.** *Define the operator* $\mathcal{T} : \mathbb{R}^{d \times d} \to \mathbb{R}^{d \times d}$ *by* $\mathcal{T}(M) = M - \mathrm{Tr}(M) \cdot \frac{I}{d}$.

**Lemma 30.** *Let* $P_2 f^*(x) = x^T T_2 x$. *Then* $\left\| T_2 - \mathbb{E}_x \left[ (g^*)'(x^T A x) \right] \cdot A \right\|_F \lesssim \frac{\kappa}{\sqrt{d}}$.

*Proof.* Throughout, we treat $p = O(1)$ and thus functions of $p$ independent of $d$ as $O(1)$ quantities. Let $q$ be of the form $g^*(z) = \sum_{k=0}^p \alpha_k z^k$. We then have

$$f^*(x) = \sum_{k=0}^p \alpha_k A^{\tilde{\otimes} k}(x^{\otimes 2k})$$

Therefore $P_2 f^*(x) = x^T T_2 x$, where

$$T_2 := \sum_{k=0}^p \alpha_k (2k-1)!! k \frac{\chi_{k+1}}{\chi_2} \mathcal{T} \left( A^{\tilde{\otimes} k}(I^{\otimes k-1}) \right).$$

Applying Lemma 29, one has

$$T_2 = \sum_{s=0}^p \mathcal{T}(A^s) \cdot \sum_{k=s}^p \alpha_k (2k-1)!! k \frac{\chi_{k+1}}{\chi_2} d_{k,s} A^{\tilde{\otimes} k-s}(I^{\otimes (k-s)}).$$

Define
$$\beta_s = \sum_{k=s}^{p} \alpha_k (2k-1)!! k \frac{\chi_{k+1}}{\chi_2} d_{k,s} A^{\tilde{\otimes}k-s}(I^{\otimes(k-s)})$$

We first see that
$$\beta_1 = \sum_{k=1}^{p} (2k-3)!! \cdot k\alpha_k \frac{\chi_{k+1}}{\chi_2} A^{\tilde{\otimes}k-1}(I^{\otimes(k-1)})$$

Next, see that
$$\frac{\chi_{k+1}}{\chi_2} = \frac{d(d+2)}{(d+2k)(d+2k-2)}\chi_{k-1} = \chi_{k-1} + O(1/d).$$

Thus
$$\beta_1 = \sum_{k=1}^{p} (2k-3)!! \cdot k\alpha_k \chi_{k-1} A^{\tilde{\otimes}k-1}(I^{\otimes(k-1)}) + O(1/d) \cdot \sum_{k=1}^{p} (2k-3)!! \cdot k|\alpha_k| A^{\tilde{\otimes}k-1}(I^{\otimes(k-1)})$$
$$= \sum_{k=1}^{p} k\alpha_k A^{\otimes k-1}\big(\mathbb{E}[x^{\otimes 2k-2}]\big) + O(1/d)$$
$$= \mathbb{E}_x\big[(g^*)'(x^T Ax)\big] + O(1/d).$$

since
$$\left| A^{\tilde{\otimes}k}(I^{\otimes k}) \right| \lesssim \mathbb{E}_x\big[(x^T Ax)^k\big] \lesssim \mathbb{E}_x\big[(x^T Ax)^2\big]^{k/2} = O(1)$$

where $\mathrm{Tr}(A) = 0$ implies $\mathbb{E}_x\big[(x^T Ax)^2\big] = O(1)$ and we invoke spherical hypercontractivity (Lemma 35). Similarly, $|\beta_s| = O(1)$, and thus
$$\left\| T_2 - \mathbb{E}_x\big[(g^*)'(x^T Ax)\big] \cdot A \right\|_F \lesssim \frac{1}{d} + \sum_{s=2}^{p} \|\mathcal{T}(A^s)\|_F \lesssim \frac{\kappa}{\sqrt{d}},$$

where we use the inequality
$$\|\mathcal{T}(X)\|_F \le \|X\|_F + |\mathrm{Tr}(X)| \cdot \frac{1}{\sqrt{d}} \le 2\|X\|_F,$$

along with
$$\|A^s\|_F \le \|A^2\|_F \le \frac{\kappa}{\sqrt{d}}$$

for $s \ge 2$. $\qquad\square$

**Lemma 31.** *Let $c_1 = \mathbb{E}_x\big[(g^*)'(x^T Ax)\big]$. Then* $\left\| \mathbb{K}f^* - \lambda_2^2(\sigma)c_1 x^T Ax \right\|_{L^2} \lesssim \kappa d^{-5/2}$

*Proof.* Since $\mathbb{E}[f^*(x)] = 0$, $P_0 f^* = 0$. Next, since $f^*$ is an even function, $P_k f^* = 0$ for $k$ odd. Thus
$$\left\| \mathbb{K}f^* - \lambda_2^2(\sigma)P_2 f^* \right\|_{L^2} \lesssim d^{-4}.$$

Additionally, by Lemma 30 we have that
$$\left\| P_2 f^* - c_1 x^T Ax \right\|_{L^2} \lesssim \|T_2 - c_1 \cdot A\|_F \lesssim \frac{\kappa}{\sqrt{d}}.$$

Since $\lambda_2^2(\sigma) = \Theta(d^{-2})$, we have
$$\left\| \mathbb{K}f^* - \lambda_2^2(\sigma)c_1 x^T Ax \right\|_{L^2} \lesssim \kappa d^{-5/2}.$$

$\qquad\square$

**Corollary 4.** *Assume $\kappa = o(\sqrt{d})$. Then*
$$\left\| x^T Ax - \|\mathbb{K}f^*\|_{L^2}^{-1} \mathbb{K}f^* \right\|_{L^2} \lesssim \kappa/\sqrt{d}$$

*Proof.*

$$\left\|x^T A x - \|\mathbb{K}f^*\|_{L^2}^{-1}\mathbb{K}f^*\right\|_{L^2}$$

$$= \|\mathbb{K}f^*\|_{L^2}^{-1}\left\|x^T A x \|\mathbb{K}f^*\|_{L^2} - \mathbb{K}f^*\right\|_{L^2}$$

$$\le \|\mathbb{K}f^*\|_{L^2}^{-1}\left\|\mathbb{K}f^* - \lambda_2^2(\sigma)c_1 x^T A x\right\|_{L^2} + \|\mathbb{K}f^*\|_{L^2}^{-1}\left|\|\mathbb{K}f^*\| - \lambda_2^2(\sigma)|c_1|\right|$$

$$\lesssim \kappa d^{-5/2}\|\mathbb{K}f^*\|_{L^2}^{-1}$$

$$\lesssim \kappa/\sqrt{d}.$$

$\square$

The proof of Theorem 7 follows directly from Corollary 4 in an identical manner to the proof of Theorem 3.

# F    Preliminaries on Spherical Harmonics

In this section we restrict to $\mathcal{X}_d = \mathcal{S}^{d-1}(\sqrt{d})$, the sphere of radius $\sqrt{d}$, and $\nu$ the uniform distribution on $\mathcal{X}_d$.

The moments of $\nu$ are given by the following [18]:

**Lemma 32.** *Let $x \sim \nu$. Then*

$$\mathbb{E}_x\left[x^{\otimes 2k}\right] = \chi_k \cdot (2k-1)!!I^{\tilde{\otimes}k}$$

*where*

$$\chi_k := \prod_{j=0}^{k-1}\left(\frac{d}{d+2j}\right) = \Theta(1).$$

For integer $\ell \ge 0$, let $V_{d,\ell}$ be the space of homogeneous harmonic polynomials on $\mathbb{R}^d$ of degree $\ell$ restricted to $\mathcal{X}_d$. One has that $V_{d,\ell}$ form an orthogonal decomposition of $L^2(\nu)$ [27], i.e

$$L^2(\nu) = \bigoplus_{\ell=0}^{\infty} V_{d,\ell}$$

Homogeneous polynomials of degree $\ell$ can be written as $T(x^{\otimes \ell})$ for an $\ell$-tensor $T \in (\mathbb{R}^d)^{\otimes \ell}$. The following lemma characterizes $V_{d,\ell}$:

**Lemma 33.** $T(x^{\otimes \ell}) \in V_{d,\ell}$ *if and only if $T(I) = 0$.*

*Proof.* By definition, a degree $l$ homogeneous polynomial $p(x) \in V_{d,l}$ if and only if $\Delta p(x) = 0$ for all $x \in S^{d-1}$. Note that $\nabla^2 p(x) = l(l-1)T(x^{\otimes(l-2)})$ so this is satisfied if and only if

$$0 = \operatorname{tr} T(x^{\otimes(l-2)}) = T(x^{\otimes(l-2)} \otimes I) = \langle T(I), x^{\otimes(l-2)}\rangle.$$

As this must hold for all $x$, this holds if and only if $T(I) = 0$. $\square$

From the above characterization, we see that $\dim(V_{d,k}) = B(d,k)$, where

$$B(d,k) = \frac{2k+d-2}{k}\binom{k+d-3}{k-1} = \frac{(k+d-3)!(2k+d-2)}{k!(d-2)!} = (1+o_d(1))\frac{d^k}{k!}.$$

Define $P_\ell : L^2(\nu) \to L^2(\nu)$ to be the orthogonal projection onto $V_{d,\ell}$. The action of $P_0, P_1, P_2$ on a homogeneous polynomial is given by the following lemma:

**Lemma 34.** *Let $T \in (\mathbb{R}^d)^{\otimes 2k}$ be a symmetric $2k$ tensor, and let $p(x) = T(x^{\otimes 2k})$ be a polynomial. Then:*

$$P_0 p = \chi_k (2k-1)!! T(I^{\otimes k})$$
$$P_1 p = 0$$
$$P_2 p = \left\langle \frac{k(2k-1)!! \chi_{k+1}}{\chi_2} \left( T(I^{\otimes k-1}) - T(I^{\otimes k}) \cdot \frac{I}{d} \right), xx^T \right\rangle$$

*Proof.* First, we see

$$P_0 p = \mathbb{E}\big[ T(x^{\otimes 2k}) \big] = \chi_k (2k-1)!! T(I^{\otimes k}).$$

Next, since $p$ is even, $P_1 p = 0$. Next, let $P_2 p = x^T T_2 x$. For symmetric $B$ so that $\mathrm{Tr}(B) = 0$, we have that

$$\mathbb{E}\big[ T(x^{\otimes 2k}) x^T B x \big] = \mathbb{E}\big[ x^T T_2 x x^T B x \big].$$

The LHS is

$$\mathbb{E}\big[ T(x^{\otimes 2k}) x^T B x \big] = (2k+1)!! \chi_{k+1} (T \tilde\otimes B) I^{\otimes k+1}$$
$$= (2k+1)!! \chi_{k+1} \frac{2k}{2k+1} \langle T(I^{\otimes k-1}), B \rangle$$
$$= 2k \cdot (2k-1)! \chi_{k+1} \langle T(I^{\otimes k-1}) - T(I^{\otimes k}) \cdot \frac{I}{d}, B \rangle,$$

where the last step is true since $\mathrm{Tr}(B) = 0$. The RHS is

$$\mathbb{E}\big[ x^T T_2 x x^T B x \big] = 3!! \chi_2 (T_2 \tilde\otimes B)(I^{\otimes 2})$$
$$= 2\chi_2 \langle T_2, B \rangle.$$

Since these two quantities must be equal for all $B$ with $\mathrm{Tr}(B) = 0$, and $\mathrm{Tr}(T_2) = 0$, we see that

$$T_2 = \frac{k(2k-1)!! \chi_{k+1}}{\chi_2} \left( T(I^{\otimes k-1}) - T(I^{\otimes k}) \cdot \frac{I}{d} \right),$$

as desired.

$\square$

Polynomials over the sphere verify hypercontractivity:

**Lemma 35** (Spherical hypercontractivity [11, 38]). *Let $f$ be a degree $p$ polynomial. Then for $q \geq 2$*

$$\|f\|_{L^q(\nu)} \leq (q-1)^{p/2} \|f\|_{L^2(\nu)}.$$

### F.1  Gegenbauer Polynomials

For an integer $d > 1$, let $\mu_d$ be the density of $x \cdot e_1$, where $x \sim \mathrm{Unif}(\mathcal{S}^{d-1}(1))$ and $e_1$ is a fixed unit vector. One can verify that $\mu_d$ is supported on $[-1, 1]$ and given by

$$d\mu_d(x) = \frac{\Gamma(d/2)}{\sqrt{\pi}\Gamma(\frac{d-1}{2})} (1 - x^2)^{\frac{d-3}{2}} dx$$

where $\Gamma(n)$ is the Gamma function. For convenience, we let $Z_d := \frac{\Gamma(d/2)}{\sqrt{\pi}\Gamma(\frac{d-1}{2})} = \frac{1}{\beta(\frac{1}{2}, \frac{d-1}{2})}$ denote the normalizing constant.

The Gegenbauer polynomials $\left( G_k^{(d)} \right)_{k \in \mathbb{Z}^{\geq 0}}$ are a sequence of orthogonal polynomials with respect to the density $\mu_d$, defined as $G_0^{(d)}(x) = 1, G_1^{(d)}(x) = x$, and

$$G_k^{(d)}(x) = \frac{d+2k-4}{d+k-3} x G_{k-1}^{(d)}(x) - \frac{k-1}{d+k-3} G_{k-2}^{(d)}(x). \tag{24}$$

By construction, $G_k^{(d)}$ is a polynomial of degree $k$. The $G_k^{(d)}$ are orthogonal in that

$$\mathbb{E}_{x \sim \mu_d}\left[G_k^{(d)}(x)G_j^{(d)}(x)\right] = \delta_{j=k}B(d,k)^{-1}$$

For a function $f \in L^2(\mu_d)$, we can write its Gegenbauer decomposition as

$$f(x) = \sum_{k=0}^{\infty} B(d,k)\langle f, G_k^{(d)}\rangle_{L^2(\mu_d)}G_k^{(d)}(x),$$

where convergence is in $L^2(\mu_d)$. For an integer $k$, we define the operator $P_k^{(d)} : L^2(\mu_d) \to L^2(\mu_d)$ to be the projection onto the degree $k$ Gegenbauer polynomial, i.e

$$P_k^{(d)}f = B(d,k)\langle f, G_k^{(d)}\rangle_{L^2(\mu_d)}G_k^{(d)}.$$

We also define the operators $P_{\leq k}^{(d)} = \sum_{\ell=0}^{k} P_\ell^{(d)}$ and $P_{\geq k}^{(d)} = \sum_{\ell=k}^{\infty} P_\ell^{(d)}$.

Recall that $\nu = \mathrm{Unif}(\mathcal{S}^{d-1}(\sqrt{d}))$. Let $\tilde{\mu}_d$ be the density of $x \cdot e_1$, where $x \sim \nu$. For a function $\sigma \in L^2(\tilde{\mu}_d)$, we define its Gegenbauer coefficients as

$$\lambda_k(\sigma) := \mathbb{E}_{x \sim \nu}[\sigma(x \cdot e_1)G_k^{(d)}(d^{-1/2}x \cdot e_1)] = \langle \sigma(d^{1/2}e_1 \cdot), G_k^{(d)}\rangle_{L^2(\mu_d)}.$$

By Cauchy, we get that $|\lambda_k(\sigma)| \leq \|\sigma\|_{L^2(\tilde{\mu}_d)}B(d,k)^{-1/2} = O(\|\sigma\|_{L^2(\tilde{\mu}_d)}d^{-k/2})$.

A key property of the Gegebauer coefficients is that they allow us to express the kernel operator $\mathbb{K}$ in closed form [27, 39]

**Lemma 36.** *For a function $g \in L^2(\nu)$, the operator $\mathbb{K}$ acts as*

$$\mathbb{K}g = \sum_{k \geq 0} \lambda_k^2(\sigma)P_k g,$$

One key fact about Gegenbauer polynomials is the following derivative formula:

**Lemma 37** (Derivative Formula).

$$\frac{d}{dx}G_k^{(d)} = \frac{k(k+d-2)}{d-1}G_{k-1}^{(d+2)}(x).$$

Furthermore, the following is a corollary of eq. (24):

**Corollary 5.**

$$G_{2k}^{(d)}(0) = (-1)^k \frac{(2k-1)!!}{\prod_{j=0}^{k-1}(d+2j-1)}.$$

# G  Proofs for Section 5

The proof of Theorem 4 relies on the following lemma, which gives the Gegenbauer decomposition of the ReLU function:

**Lemma 38** (ReLU Gegenbauer). *Let $\mathrm{ReLU}(x) = \max(x, 0)$. Then*

$$\mathrm{ReLU}(x) = \frac{1}{\beta(\frac{1}{2}, \frac{d-1}{2})(d-1)}G_0^{(d)}(x) + \frac{1}{2d}G_1^{(d)}(x)$$

$$+ \sum_{k \geq 1}(-1)^{k+1}\frac{(2k-3)!!}{\beta(\frac{1}{2}, \frac{d-1}{2})\prod_{j=0}^{k}(d+2j-1)}B(d,2k)G_{2k}^{(d)}(x).$$

*As a consequence,*

$$\left\|P_{\geq 2m}^{(d)}\mathrm{ReLU}(x)\right\|_{L^2(\mu_d)}^2 = \sum_{k \geq m}\frac{(2k-3)!!^2B(d,2k)}{\beta(\frac{1}{2}, \frac{d-1}{2})^2\prod_{j=0}^{k}(d+2j-1)^2}$$

The proof of this lemma is deferred to Appendix G.1.

We also require a key result from [19], which lower bounds the approximation error of an inner product function.

**Definition 9.** $f : \mathcal{S}^{d-1}(1) \times \mathcal{S}^{d-1}(1) \to \mathbb{R}$ *is an* inner product function *if* $f(x, x') = \phi(\langle x, x' \rangle)$ *for some* $\phi : [-1, 1] \to \mathbb{R}$.

**Definition 10.** $g : \mathcal{S}^{d-1}(1) \times \mathcal{S}^{d-1}(1) \to \mathbb{R}$ *is a* separable function *if* $g(x, x') = \psi(\langle v, x \rangle, \langle v', x' \rangle)$ *for some* $v, v' \in \mathcal{S}^{d-1}(1)$ *and* $\psi : [-1, 1]^2 \to \mathbb{R}$.

Let $\tilde{\nu}_d$ be the uniform distribution over $\mathcal{S}^{d-1}(1) \times \mathcal{S}^{d-1}(1)$. We note that if $(x, x') \sim \tilde{\nu}_d$, then $\langle x, x' \rangle \sim \mu_d$. For an inner product function $f$, we thus have $\|f\|_{L^2(\tilde{\nu}_d)} = \|\phi\|_{L^2(\mu_d)}$. We overload notation and let $\left\|P_k^{(d)} f\right\|_{L^2(\mu_d)} = \left\|P_k^{(d)} \phi\right\|_{L^2(\mu_d)}$.

**Lemma 39.** *[19, Theorem 3] Let $f$ be an inner product function and $g_1, \ldots, g_r$ be separable functions. Then, for any $k \geq 1$,*

$$\left\| f - \sum_{i=1}^{r} g_i \right\|_{L^2(\tilde{\nu}_d)}^2 \geq \|P_k f\|_{L^2(\mu_d)} \cdot \left( \|P_k f\|_{L^2(\mu_d)} - \frac{2 \sum_{i=1}^{r} \|g_r\|_{L^2(\tilde{\nu}_d)}}{B(d, k)^{1/2}} \right).$$

We now can prove Theorem 4

*Proof of Theorem 4.* We begin with the lower bound. Let $x = U \begin{pmatrix} x_1 \\ x_2 \end{pmatrix}$, where $x_1, x_2 \in \mathbb{R}^{d/2}$. Assume that there exists some $\theta$ such that $\|f^* - N_\theta\|_{L^2(\mathcal{X})} \leq \epsilon$. Then

$$\epsilon^2 \geq \mathbb{E}_x \left[ (f^*(x) - N_\theta(x))^2 \right]$$
$$= \mathbb{E}_{r \sim \mu} \left[ \mathbb{E}_{x_1 \sim \mathcal{S}^{d-1}(\sqrt{r}), x_2 \sim \mathcal{S}^{d-1}(\sqrt{d-r})} (f^*(x) - N_\theta(x))^2 \right],$$

where $r$ is the random variable defined as $r = \|x_1\|^2$ and $\mu$ is the associated measure. The equality comes from the fact that conditioned on $r$, $x_1$ and $x_2$ are independent and distributed uniformly on the spheres of radii $\sqrt{r}$ and $\sqrt{d-r}$, respectively. We see that $\mathbb{E}r = d/2$, and thus

$$\mathbb{P}(|r - d/2| > d/4) \leq \exp(-\Omega(d))$$

Let $\delta = \inf_{r \in [d/4, 3d/4]} \mathbb{E}_{x_1 \sim \mathcal{S}^{d-1}(\sqrt{r}), x_2 \sim \mathcal{S}^{d-1}(\sqrt{d-r})} (f^*(x) - N_\theta(x))^2$. We get the bound

$$\epsilon^2 \geq \delta \cdot \mathbb{P}(r \in [d/4, 3d/4]),$$

and thus $\delta \leq \epsilon^2 (1 - \exp(-\Omega(d)))$. Therefore there exists an $r \in [d/4, 3d/4]$ such that

$$\mathbb{E}_{x_1 \sim \mathcal{S}^{d-1}(\sqrt{r}), x_2 \sim \mathcal{S}^{d-1}(\sqrt{d-r})} (f^*(x) - N_\theta(x))^2 \leq \epsilon^2/2.$$

Next, see that when $\|x_1\|^2 = r, \|x_2\|^2 = d - r$, we have that

$$x^T A x = \frac{2}{\sqrt{d}} \langle x_1, x_2 \rangle = 2\sqrt{\frac{r(d-r)}{d}} \langle \overline{x}_1, \overline{x}_2 \rangle,$$

where now $\overline{x}_1, \overline{x}_2 \sim \mathcal{S}^{d/2-1}(1)$ i.i.d. Defining $q(z) = \text{ReLU}\left( 2\sqrt{\frac{r(d-r)}{d}} z \right) - c_0$, we thus have

$$\overline{q}(\langle \overline{x}_1, \overline{x}_2 \rangle) = \text{ReLU}(x^T A x) - c_0 = f^*(x).$$

Furthermore, defining $\overline{x} = \begin{pmatrix} \overline{x}_1 \\ \overline{x}_2 \end{pmatrix}$, choosing the parameter vector $\overline{\theta} = (a, \overline{W}, b_1, b_2)$, where $\overline{W} = WU \begin{pmatrix} \sqrt{r} \cdot I & 0 \\ 0 & \sqrt{d-r} \cdot I \end{pmatrix}$ yields a network so that $N_{\overline{\theta}}(\overline{x}) = N_\theta(x)$. Therefore we get that the new network $N_{\overline{\theta}}$ satisfies

$$\mathbb{E}_{\overline{x}_1, \overline{x}_2} \left[ \left( \overline{q}(\langle \overline{x}_1, \overline{x}_2 \rangle) - N_{\overline{\theta}}(\overline{x}) \right)^2 \right] \leq \epsilon^2/2,$$

where $\overline{x}_1, \overline{x}_2$ are drawn i.i.d over $\mathrm{Unif}(\mathcal{S}^{d/2-1}(1))$.

We aim to invoke Lemma 39. We note that $(\overline{x}_1, \overline{x}_2) \sim \tilde{\nu}_{d/2}$, and that $\overline{q}$ is an inner product function. Define $g_i(\overline{x}) = a_i \sigma(\overline{w}_i^T \overline{x} + b_{1,i})$. We see that $g_i$ is a separable function, and also that

$$N_{\overline{\theta}}(\overline{x}) = \sum_{i=1}^{m} g_i(\overline{x}) + b_2.$$

Hence $N_{\overline{\theta}})$ is the sum of $m + 1$ separable functions. We can bound the a single function as

$$
\begin{aligned}
|g_i(\overline{x})| &\le |a_i| C_\sigma \left(1 + |\overline{w}_i^T \overline{x} + b_{1,i}|\right)^{\alpha_\sigma} \\
&\le C_\sigma B (1 + \sqrt{d}\|\overline{w}_i\|_\infty + B)^{\alpha_\sigma} \\
&\le C_\sigma B (1 + Bd^{3/2}/2 + B)^{\alpha_\sigma} \\
&\le (Bd^{3/2})^{\alpha_\sigma + 1},
\end{aligned}
$$

since $\|\overline{W}\|_\infty \le \max(\sqrt{r}, \sqrt{d-r})\|WU\|_\infty \le Bd/2$. Therefore by Lemma 39

$$\mathbb{E}_{\overline{x}_1, \overline{x}_2}\left[\left(\overline{q}(\langle \overline{x}_1, \overline{x}_2\rangle) - N_{\overline{\theta}}(\overline{x})\right)^2\right] \ge \|P_{\ge k}\overline{q}\|_{L^2} \cdot \left(\|P_{\ge k}\overline{q}\|_{L^2} - \frac{2(m+1)(Bd^{3/2})^{\alpha_\sigma + 1}}{\sqrt{B_{d/2,k}}}\right).$$

By Lemma 38, we have that

$$\|P_{\ge 2m}\mathrm{ReLU}(x)\|_{L^2(\mu_{d/2})}^2 = \sum_{k \ge m} \frac{(2k-3)!!^2 B(d/2, 2k)}{\beta(\frac{1}{2}, \frac{d/2-1}{2})^2 \prod_{j=0}^{k}(d/2 + 2j - 1)^2}$$

Simplifying, we have that

$$
\begin{aligned}
\frac{(2k-3)!!^2 B(d/2, 2k)}{\prod_{j=0}^{k}(d/2 + 2j - 1)^2} &= \frac{(2k-3)!!^2}{(2k)!} \cdot \frac{(d/2 + 2k - 3)!(d/2 + 4k - 2)}{(d/2-2)! \prod_{j=0}^{k}(d/2 + 2j - 1)^2} \\
&= \frac{(2k-3)!!^2}{(2k)!} \cdot \frac{(d/2 + 4k - 2)}{(d/2 + 2k - 1)^2(d/2 + 2k - 3)} \prod_{j=0}^{k-2} \frac{d/2 + 2j}{d/2 + 2j - 1} \\
&\ge \frac{(2k-3)!!^2}{(2k)!} \cdot \frac{(d/2 + 4k - 2)}{(d/2 + 2k - 1)^2(d/2 + 2k - 3)} \\
&\ge \frac{1}{2k(2k-1)(2k-2)} \cdot \frac{(d/2 + 4k - 2)}{(d/2 + 2k - 1)^2(d/2 + 2k - 3)}
\end{aligned}
$$

By Gautschi's inequality, we can bound

$$\beta\left(\frac{1}{2}, \frac{d-2}{4}\right) = \frac{\sqrt{\pi}\Gamma(\frac{d-2}{4})}{\Gamma(\frac{d}{4})} \le \sqrt{\pi}\left(\frac{d}{4} - 1\right)^{-1/2} \le 4d^{-1/2}.$$

Therefore

$$
\begin{aligned}
\|P_{2k}\mathrm{ReLU}(x)\|_{L^2(\mu_{d/2})}^2 &\ge \frac{1}{2k(2k-1)(2k-2)16} \cdot \frac{(d/2 + 4k - 2)d}{(d/2 + 2k - 1)^2(d/2 + 2k - 3)} \\
&\ge \frac{1}{128k^3} \cdot \frac{(d/2 + 4k - 2)d}{(d/2 + 2k - 1)^2(d/2 + 2k - 3)} \\
&\ge \frac{1}{128k^3 d}
\end{aligned}
$$

for $k \le d/4$. Altogether,

$$\|P_{\ge 2m}\mathrm{ReLU}(x)\|_{L^2(\mu_{d/2})}^2 \ge \frac{1}{128d} \sum_{k=m}^{d/4} \frac{1}{k^3} \ge \frac{1}{512m^2 d}$$

for $m \le d/8$. Since $\overline{q}(z) = \mathrm{ReLU}(2\sqrt{\frac{r(d-r)}{d}}z) - c_0 = 2\sqrt{\frac{r(d-r)}{d}}\mathrm{ReLU}(z) - c_0$, we have that

$$\|P_{\ge 2m}\overline{q}\|_{L^2(\mu)} \ge 4\frac{r(d-r)}{d}\|P_{\ge 2m}\mathrm{ReLU}(x)\|$$

$$\ge \sqrt{3d}\cdot\sqrt{\frac{1}{512m^2d}}$$

$$\ge \frac{1}{16m}.$$

We thus have, for any integer $k < d/8$,

$$\epsilon^2/2 \ge \mathbb{E}_{\overline{x}_1,\overline{x}_2}\left[\left(\overline{q}(\langle\overline{x}_1,\overline{x}_2\rangle) - N_{\overline{\theta}}(\overline{x})\right)^2\right]$$

$$\ge \|P_{\ge 2k}\overline{q}\|_{L^2}\cdot\left(\|P_{\ge 2k}\overline{q}\|_{L^2} - \frac{2(m+1)(Bd^{3/2})^{\alpha_\sigma+1}}{\sqrt{B_{d/2,2k}}}\right)$$

Choose $\epsilon \le \frac{1}{512k^2}$; we then must have

$$\frac{2(m+1)(Bd^{3/2})^{\alpha_\sigma+1}}{\sqrt{B_{d/2,2k}}} \ge \frac{1}{32k},$$

or

$$(m+1)(Bd^{3/2})^{\alpha_\sigma+1} \ge \frac{1}{64k}B(d/2,2k)^{1/2} \ge d^k 2^{-k}\cdot\frac{1}{64k\sqrt{(2k)!}}$$

$$= C_1\exp\left(k\log d - \log k - \frac{1}{2}\log(2k)! - k\log 2\right)$$

$$\ge C_1\exp\left(k\log d - \log k - k\log(2k) - k\log 2\right)$$

$$\ge C_1\exp\left(k\log\frac{d}{k} - \log k - 2k\log 2\right)$$

$$\ge C_1\exp\left(C_2 k\log\frac{d}{k}\right)$$

for any $k \le C_3 d$. Selecting $k = \lfloor\sqrt{\frac{1}{512\epsilon}}\rfloor$ yields

$$\max(m,B) \ge C_1\exp\left(C_2\epsilon^{-1/2}\log(d\epsilon)\right)\cdot d^{-3/2} \ge C_1\exp\left(C_2\epsilon^{-1/2}\log(d\epsilon)\right)$$

for $\epsilon$ less than a universal constant $c_3$.

We next show the upper bound,

It is easy to see that Assumptions 1 and 3 are satisfied. Next, since the verification of Assumptions 2 and 4 only required Lipschitzness, those assumptions are satisfied as well with $\ell,\chi = 1$. Finally, we have

$$\mathbb{E}_x\left[f^*(x)^2\right] \le \mathbb{E}_x\left[\mathrm{ReLU}^2(x^T Ax)\right] = \frac{1}{2}\mathbb{E}_x\left[(x^T Ax)^2\right] = \frac{d}{d+2} < 1.$$

Next, observe that $\|\mathbb{K}f^*\|_{L^2} \lesssim d^{-2}$. Define $\overline{A} = \sqrt{\frac{d+2}{2d}}A$. This scaling ensures $\|x^T\overline{A}x\|_{L^2} = 1$. Then, we can write $f^*(x) = g^*(x^T\overline{A}x)$ for $g^*(z) = \sqrt{\frac{2d}{d+2}}\mathrm{ReLU}(z) - c_0$. For $\epsilon > 0$, define the smoothed ReLU $\mathrm{ReLU}_\epsilon(z)$ as

$$\mathrm{ReLU}_\epsilon(z) = \begin{cases} 0 & z \le -\epsilon \\ \frac{1}{4\epsilon}(x+\epsilon)^2 & -\epsilon \le 0 \le \epsilon \\ x & x \ge \epsilon \end{cases}.$$

One sees that $\mathrm{ReLU}_\epsilon$ is twice differentiable with $\|\mathrm{ReLU}_\epsilon\|_{1,\infty} \le 1$ and $\|\mathrm{ReLU}_\epsilon\|_{2,\infty} = \frac{1}{2\epsilon}$

We select the test function $q$ to be $q(z) = \sqrt{\frac{2d}{d+2}}\mathrm{ReLU}_\epsilon(\overline{\eta}^{-1}\|\mathbb{K}f^*\|_{L^2}^{-1} \cdot z) - c_0$. We see that

$$q\big(\overline{\eta}(\mathbb{K}f^*)(x)\big) = \mathrm{ReLU}_\epsilon\Big(\|\mathbb{K}f^*\|_{L^2}^{-1}(\mathbb{K}f^*)(x)\Big),$$

and thus

$\|f^* - q(\overline{\eta}(\mathbb{K}f^*)(x))\|_{L^2}$

$= \sqrt{\dfrac{2d}{d+2}}\Big\|\mathrm{ReLU}(x^T\overline{A}x) - \mathrm{ReLU}_\epsilon\Big(\|\mathbb{K}f^*\|_{L^2}^{-1}(\mathbb{K}f^*)(x)\Big)\Big\|_{L^2}$

$\le \Big\|\mathrm{ReLU}(x^T\overline{A}x) - \mathrm{ReLU}_\epsilon(x^T\overline{A}x)\Big\|_{L^2} + \Big\|\mathrm{ReLU}_\epsilon(x^T\overline{A}x) - \mathrm{ReLU}_\epsilon\Big(\|\mathbb{K}f^*\|_{L^2}^{-1}(\mathbb{K}f^*)(x)\Big)\Big\|_{L^2}$

$\lesssim \Big\|\mathrm{ReLU}(x^T\overline{A}x) - \mathrm{ReLU}_\epsilon(x^T\overline{A}x)\Big\|_{L^2} + \Big\|x^T\overline{A}x - \|Kf^*\|_{L^2}^{-1}\mathbb{K}f^*\Big\|_{L^2}$

$\lesssim \Big\|\mathrm{ReLU}(x^T\overline{A}x) - \mathrm{ReLU}_\epsilon(x^T\overline{A}x)\Big\|_{L^2} + Ld^{-1/12}\log d,$

where the first inequality follows from Lipschitzness and the second inequality is Corollary 3, using $\kappa = 1$.

There exists a constant upper bound for the density of $x^T\overline{A}x$, and thus we can upper bound

$$\Big\|\mathrm{ReLU}(x^T\overline{A}x) - \mathrm{ReLU}_\epsilon(x^T\overline{A}x)\Big\|_{L^2}^2 \lesssim \int_0^\epsilon \frac{1}{\epsilon^2}z^4 dz \lesssim \epsilon^3.$$

Furthermore since $\overline{\eta} = \Theta(\|\mathbb{K}f^*\|_{L^2}^{-1}\iota^{-\chi})$, we get that $\overline{\eta}^{-1}\|\mathbb{K}f^*\|_{L^2}^{-1} = \Theta(\iota^\chi)$, and thus

$$\sup_{z\in[-1,1]}|q(z)| = \sup_{z\in[-\Theta(\iota^\chi),\Theta(\iota^\chi)]}|\mathrm{ReLU}_\epsilon(z)| = \mathrm{poly}(\iota)$$

$$\sup_{z\in[-1,1]}|q'(z)| = \overline{\eta}^{-1}\|\mathbb{K}f^*\|_{L^2}^{-1}\sup_{z\in[-\Theta(\iota^\chi),\Theta(\iota^\chi)]}|(\mathrm{ReLU}_\epsilon)'(z)| = \mathrm{poly}(\iota)$$

$$\sup_{z\in[-1,1]}|q''(z)| = \Big(\overline{\eta}^{-1}\|\mathbb{K}f^*\|_{L^2}^{-1}\Big)^2 \sup_{z\in[-\Theta(\iota^\chi),\Theta(\iota^\chi)]}|(\mathrm{ReLU}_\epsilon)''(z)| = \mathrm{poly}(\iota)\epsilon^{-1}$$

Therefore by Theorem 6 we can bound the population loss as

$$\mathbb{E}_x\Big[\big(f(x;\hat{\theta}) - f^*(x)\big)^2\Big] \lesssim \tilde{O}\Big(\frac{d^4}{\min(n,m_1,m_2)} + \frac{\epsilon^{-2}}{m_1} + \sqrt{\frac{\epsilon^{-4}}{n}} + \epsilon^3 + d^{-1/6}\Big).$$

Choosing $\epsilon = d^{-1/4}$ yields the desired result. As for the sample complexity, we have $\|q\|_{2,\infty} = \tilde{O}(\epsilon^{-1}) = \tilde{O}(d^{1/4})$, and so the runtime is $\mathrm{poly}(d, m_1, m_2, n)$. $\qquad\square$

### G.1   Proof of Lemma 38

*Proof of Lemma 38.* For any integer $k$, we define the quantities $A_{2k}^{(d)}, B_{2k+1}^{(d)}$ as

$$A_{2k}^{(d)} := \int_0^1 xG_{2k}^{(d)}(x)d\mu_d(x)$$

$$B_{2k+1}^{(d)} = \int_0^1 G_{2k+1}^{(d)}(x)d\mu_d(x).$$

We also let $Z_d = \frac{\Gamma(d/2)}{\sqrt{\pi}\Gamma(\frac{d-1}{2})}$ to be the normalization constant.

Integration by parts yields

$$A_{2k}^{(d)} = Z_d \int_0^1 G_{2k}^{(d)}(x)x(1-x^2)^{\frac{d-3}{2}}dx$$

$$= -Z_d \cdot G_{2k}^{(d)}(x) \cdot \frac{1}{d-1}(1-x^2)^{\frac{d-1}{2}}\Big|_0^1 + \frac{Z_d}{d-1} \cdot \frac{2k(2k+d-2)}{d-1}\int_0^1 G_{2k-1}^{(d+2)}(x)(1-x^2)^{\frac{d-1}{2}}dx$$

$$= \frac{Z_d}{d-1}G_{2k}^{(d)}(0) + \frac{Z_d}{Z_{d+2}} \cdot \frac{2k(2k+d-2)}{(d-1)^2} \cdot B_{2k-1}^{(d+2)}$$

From Corollary 5 we have

$$G_{2k}^{(d)}(0) = \frac{(2k-1)!!}{\Pi_{j=0}^{k-1}(d+2j-1)}(-1)^k.$$

Thus

$$A_{2k}^{(d)} = Z_d \cdot \frac{(2k-1)!!}{(d-1)\Pi_{j=0}^{k-1}(d+2j-1)}(-1)^k + \frac{2k(2k+d-2)}{(d-1)^2} \cdot \frac{Z_d}{Z_{d+2}}B_{2k-1}^{(d+2)}. \qquad (25)$$

The recurrence formula yields

$$B_{2k+1}^{(d)} = \int_0^1 G_{2k+1}^{(d)}(x)d\mu_d(x) \qquad (26)$$

$$= \int_0^1 \left[\frac{4k+d-2}{2k+d-2}xG_{2k}^{(d)}(x) - \frac{2k}{2k+d-2}G_{2k-1}^{(d)}(x)\right]d\mu_d(x) \qquad (27)$$

$$= \frac{4k+d-2}{2k+d-2}A_{2k}^{(d)} - \frac{2k}{2k+d-2}B_{2k-1}^{(d)}. \qquad (28)$$

I claim that

$$A_{2k}^{(d)} = \begin{cases} \frac{Z_d}{d-1} & k=0 \\ (-1)^{k+1}Z_d\frac{(2k-3)!!}{\prod_{j=0}^{k}(d+2j-1)} & k \geq 1 \end{cases} \quad \text{and} \quad B_{2k+1}^{(d)} = (-1)^kZ_d\frac{(2k-1)!!}{\prod_{j=0}^{k}(d+2j-1)}.$$

We proceed by induction on $k$. For the base cases, we first have

$$A_0^{(d)} = \int_0^1 xd\mu_d(x) = \int_0^1 Z_dx(1-x^2)^{\frac{d-3}{2}}$$

$$= \int_0^1 \frac{Z_d}{d-1}du$$

$$= \frac{Z_d}{d-1},$$

where we use the substitution $u = (1-x^2)^{\frac{d-3}{2}}$. Next,

$$B_1^{(d)} = \int_0^1 xd\mu_d(x) = A_0^{(d)} = \frac{Z_d}{d-1}.$$

Next, eq. (25) gives

$$A_2^{(d)} = Z_d \cdot \frac{-1}{(d-1)^2} + \frac{2d}{(d-1)^2} \cdot \frac{Z_d}{d+1}$$

$$= \frac{Z_d}{(d-1)(d+1)}.$$

Finally, eq. (26) gives

$$B_3^{(d)} = \frac{d+2}{d}A_2^{(d)} - \frac{2}{d}B_1^{(d)}$$

$$= \frac{Z_d}{d-1}\left[\frac{d+2}{d(d+1)} - \frac{2}{d}\right]$$

$$= -\frac{Z_d}{(d-1)(d+1)}.$$

Therefore the base case is proven for $k = 0, 1$.

Now, assume that the claim is true for some $k \geq 1$ for all $d$. We first have

$$
A_{2k+2}^{(d)} = Z_d \cdot \frac{(2k+1)!!}{(d-1)\Pi_{j=0}^{k}(d+2j-1)}(-1)^{k+1} + \frac{(2k+2)(2k+d)}{(d-1)^2} \cdot \frac{Z_d}{Z_{d+2}} B_{2k+1}^{(d+2)}
$$

$$
= Z_d \cdot \frac{(2k+1)!!}{(d-1)\Pi_{j=0}^{k}(d+2j-1)}(-1)^{k+1} + Z_d \cdot \frac{(2k+2)(2k+d)}{(d-1)^2} \cdot \frac{(2k-1)!!}{\Pi_{j=0}^{k}(d+2j+1)}(-1)^k
$$

$$
= (-1)^{k+1} Z_d \cdot \frac{(2k-1)!!}{\Pi_{j=0}^{k+1}(d+2j-1)}\left[\frac{(d+2k+1)(2k+1)}{d-1} - \frac{(2k+2)(2k+d)}{d-1}\right]
$$

$$
= (-1)^{k+1} Z_d \cdot \frac{(2k-1)!!}{\Pi_{j=0}^{k+1}(d+2j-1)}\left[\frac{-d+1}{d-1}\right]
$$

$$
= (-1)^{k+2} Z_d \cdot \frac{(2k-1)!!}{\Pi_{j=0}^{k+1}(d+2j-1)}.
$$

Next, we have

$$
B_{2k+3}^{(d)} = \frac{4k+d+2}{2k+d}A_{2k+2}^{(d)} - \frac{2k+2}{2k+d}B_{2k+1}^{(d)}
$$

$$
= (-1)^k Z_d \left[\frac{4k+d+2}{2k+d}\frac{(2k-1)!!}{\Pi_{j=0}^{k+1}(d+2j-1)} - \frac{2k+2}{2k+d}\frac{(2k-1)!!}{\prod_{j=0}^{k}(d+2j-1)}\right]
$$

$$
= (-1)^k Z_d \frac{(2k-1)!!}{\prod_{j=0}^{k+1}(d+2j-1)}\left[\frac{(4k+d+2) - (2k+2)(d+2k+1)}{2k+d}\right]
$$

$$
= (-1)^k Z_d \frac{(2k-1)!!}{\prod_{j=0}^{k+1}(d+2j-1)}\left[\frac{-(2k+1)(2k+d)}{2k+d}\right]
$$

$$
= (-1)^{k+1} Z_d \frac{(2k+1)!!}{\prod_{j=0}^{k+1}(d+2j-1)}.
$$

Therefore by induction the claim holds for all $k, d$.

The Gegenbauer expansion of ReLU is given by

$$
\mathrm{ReLU}(x) = \sum_{i=0}^{\infty}\langle \mathrm{ReLU}, G_i^{(d)}\rangle_{L^2(\mu_d)} B(d,i)G_i^{(d)}(x).
$$

Note that $\mathrm{ReLU}(x) = \frac{1}{2}(x + |x|)$. Since $|x|$ is even, the only nonzero odd Gegenbauer coefficient is for $G_1^{(d)}$. In this case,

$$
\langle \mathrm{ReLU}, G_1^{(d)}\rangle_{L^2(\mu_d)} = \frac{1}{2}\mathbb{E}_{x\sim\mu_d}[x^2] = \frac{1}{2d^2}.
$$

Also, $B(d,1) = d$. Next, we see that

$$
\langle \mathrm{ReLU}, G_{2k}^{(d)}\rangle_{L^2(\mu_d)} = \int_{-1}^{1}\mathrm{ReLU}(x)G_{2k}^{(d)}(x)d\mu_d(x) = \int_{0}^{1}xG_{2k}^{(d)}(x)d\mu_d(x) = A_{2k}^{(d)}.
$$

Plugging in our derivation for $A_{2k}^{(d)}$ gives the desired result. $\qquad\square$

