# OpenReview forum: "Provable Guarantees for Nonlinear Feature Learning in Three-Layer Neural Networks"
_NeurIPS.cc/2023/Conference — NeurIPS 2023 spotlight_

### Official Review · Reviewer_3iYS · 2023-07-06

**Soundness:** 4 excellent
**Presentation:** 4 excellent
**Contribution:** 4 excellent
**Rating:** 8
**Confidence:** 4

**Summary:**

This paper studies the feature learning capability of a 3-layer neural
network, where the bottom layer is random and fixed, the middle layer is
trained for only one step from zero, and the upper layer is trained in
the rest of the gradient descent steps. The paper characterizes the
richer feature learning capability of this 3-layer network. That is, the
one-step training of the middle-layer weights essentially maps the
original random features to another space, which enlarges the type of
functions that can be learned. Two special cases are then considered:
(i) the single index model (similar to [18]); and (ii) more
significantly, the quadratic feature model (more complex than [18]). The
results demonstrate a significant reduction in the sample complexity
compared to kernel methods. Further, the latter class of functions in
(ii) cannot be approximated by 2-layer networks, and thus a
depth-separation between 2-layer and 3-layer networks is established.

**Strengths:**

1. The results are novel and solid. The precisely characterization of
the feature learning capability of this particular 3-layer network
(Theorem 1), and the ability to learn functions of quadratic features,
are both significant contributions to the field.

2. The paper is also very well-written and presents the main intuitions
quite clearly.

**Weaknesses:**

Nothing particular.

**Questions:**

1. Can the authors comment on other possible features that this
approach can learn effectively (beyond quadratic features)?

2. The one-step training of the middel-layer weights seems to be a
crucial element for learning a specific type of features. At the same
time, one-step training could be limiting. Can the authors comment on
possible generalizations for multi-step training on the middle-layer?

Post rebuttal phase:

The reviewer wishes to thank the authors for their response and preliminary thoughts on these questions.

**Limitations:**

The discussion of limitations is adequate.

---

> ### Author Rebuttal · Authors · 2023-08-07
>
> We thank the reviewer for their detailed and thoughtful review, and address specific comments below.
>
> > “Can the authors comment on other possible features that this approach can learn effectively”
> - We hypothesize that our results here can be extended to learn arbitrary polynomial features, and thus allow us to learn hierarchical functions $f^* = g^* \circ h^*$ where $h^*$ is a polynomial, with sample complexity scaling with the degree of $h^*$. However, we don’t currently have any concrete results in this setting and hence defer this to future work.
>
> > “Can the authors comment on possible generalizations for multi-step training on the middle-layer?”
> - While initial works studying feature learning in two-layer neural networks such as [8, 18] relied on the single-step training procedure, later works [2] studied training $W$ for multiple steps and were able to improve the dimension dependence in the sample complexity. For three layer neural networks, it is possible that a more refined analysis of multi-step training could improve the sample complexity. However, analyzing the training dynamics of $W$ for multiple steps introduces a number of new technical challenges which will likely require developing new techniques. As such, we defer this to future work.

---

> > ### Comment · Reviewer_3iYS · 2023-08-11
> >
> > I thank the authors for their response. I will keep my review score.

---

### Official Review · Reviewer_aPJJ · 2023-07-06

**Soundness:** 4 excellent
**Presentation:** 4 excellent
**Contribution:** 3 good
**Rating:** 8
**Confidence:** 3

**Summary:**

The paper theoretically studies the feature learning in three layer neural networks. For the analysis, it considers layer-wise GD; more precisely, the first layer is not trained, the second layer is trained for one step, and then there is the training for last layer. Particularly, they show that three layer neural networks achieve a better bound for learning functions of the form $g(x^TAx)$ comparing to the known bounds for two-layer neural networks. They also show an optimization-based separation showing that three-layer NNs can learn functions that two-layer NNs cannot learn with polynomial width and time (assuming a non-increasing learning rate).

**Strengths:**

- The paper considers the optimization of three-layer NNs with a layer-wise training and provides a fairly general approach for studying their feature learning capabilities (in the paper, the features have also been characterized for the examples considered).
- For functions of the type $g(x^TAx)$, a sample complexity upper bound for three-layer NNs is proved which is more efficient than the known upper bounds for two-layer NNs.
- There is an optimization-based separation result provided between two-layer and three-layer NNs.
- Paper is generally very well written.


**Weaknesses:**

- Generally the limitations of the work have been discussed in the paper. These limitations are usually quite common in the deep learning theory literature. For example, the training is layer-wise, and the first layer is not trained at all and second layer is only trained for one iteration. In this sense, the work is similar to theoretical analysis done on two-layer NNs in many recent works.
- The simulations have been implemented for the settings as the theorems. However, it would have been interesting to also run experiments with the more common setting (e.g., training all parameters together) for both two-layer and three-layer NNs and compare the results with the results of the model in the theoretical settings (e.g., with layer-wise training). This would potentially show if the assumptions made in theory are too restrictive.


**Questions:**

- Can the analysis be done in the SGD setting as well?

Here are also a few typos:
- line 246: $\mathbb{E}[(g^{*})'] = 0$ instead of $\mathbb{E}[g'] = 0$
- line 291: inconsistency regarding $P_2 f^*$ (e.g., with equation 16) (I think also the same problem appeared in the appendix.)
- line 298: is it the intended equation?

**Limitations:**

There is no negative societal impact. The limitations of theory have been discussed in the work and are (unfortunately) quite common in the current literature of deep learning theory.

---

> ### Author Rebuttal · Authors · 2023-08-07
>
> We thank the reviewer for their detailed and thoughtful review, and address specific comments below:
>
> > “However, it would have been interesting to also run experiments with the more common setting (e.g., training all parameters together) for both two-layer and three-layer NNs…”
> - We agree that this is an interesting experiment to run, and will add it to a future revision of our paper.
>
> > “Can the analysis be done in the SGD setting as well?”
> - For the first stage we do need a large batch size, but the second stage can be done in the online SGD setting. Since the loss is convex in $a$, the analysis for this would be a standard application of online convex optimization results.
>
> Thank you for pointing out the typos, we will update these in the next revision of our paper. Line 298 should read $\mathbb{K}f^*(x) = \Theta(d^{-2})\cdot(x^TAx + o_d(1))$.

---

> > ### Comment · Reviewer_aPJJ · 2023-08-16
> >
> > Thank you for your response. I will maintain my score.

---

### Official Review · Reviewer_8d6Z · 2023-07-07

**Soundness:** 3 good
**Presentation:** 3 good
**Contribution:** 3 good
**Rating:** 6
**Confidence:** 3

**Summary:**

In this work the authors show that there is a three layer neural network setup with better provable learning guarantees than the current best bound for two layer setups. The setup involves a randomly initialized layer with frozen weights, which feeds into a two layer network where first the hidden layer weights are trained, and then the output weights are trained. In the case of functions of quadratic features, this three layer (but two learnable layers) network is able to use feature learning to re-weight the initial random feature kernel to improve training efficiency.

**Strengths:**

The model used in the paper is a simple and sensible extension of the two layer network. The learning algorithm is also quite reasonable, both in its connection to previous work as well as to the practical end of deep learning.

The arguments seem correct (though I did not fully validate the detailed proofs in the appendices), and the intuitive explanation relating feature learning and eigenspace weighting helps shed light on the importance (and perhaps mechanisms of) feature learning in deep networks. The bounds in the 3 layer case are a marked improvement over the kernel learning and 2 hidden layer case bounds.

**Weaknesses:**

There are two concerns with this paper. The first is a question of the tightness of the bounds. While the bounds show great improvement over the quoted bounds in the two hidden layer case, it is not clear how tight these bounds are in the various scenarios discussed.

Relatedly, there is a big question as to whether or not the bounds are useful for understanding the success of deep learning systems even on simple problems (e.g. FCN on MNIST). In particular, the proof sketch seems to suggest that just a single step of GD in the middle layer is enough to induce massive improvements in the sample complexity; however even in simple settings, it seems that it is helpful to both learn in multiple layers, as well as over multiple timepoints. I note that this is a general weakness of similar sample complexity analyses and not of this particular paper, and the authors do mention this in the discussion.

**Questions:**

What evidence is there that the difference between the 2 and 3 layer bounds will persist even as theoretical techniques/the bounds themselves are improved?

How do the 3 layer feature learning bounds differ from NTK learning of a similar 3 layer network? (This discussion would be useful to add to the main text to emphasize the benefits of feature learning over depth alone.)

Can the method be used to show anything about the usefulness of running gradient descent over a longer period of time? How does this trade off with the sample complexity?

Is there any intuition about good choices of the function q in the bounds, in a more general setting?

**Limitations:**

Yes

---

> ### Author Rebuttal · Authors · 2023-08-07
>
> We thank the reviewer for their detailed and thoughtful review, and address specific concerns of yours below.
>
> > “While the bounds show great improvement over the quoted bounds in the two hidden layer case, it is not clear how tight these bounds are in the various scenarios discussed.”
> - For the quadratic feature setting (Section 4.2), the information theoretic lower bound for the sample complexity is $d^2$, as there are $O(d^2)$ free parameters in the matrix $A$. Our theorem 3 shows that the three-layer network can learn the target with $d^4$ samples, which is within a polynomial factor of this optimal sample complexity (but crucially still better than kernel methods or two-layer networks). Similarly, the information theoretic lower bound for single-index models is $O(d)$, while our algorithm obtains an $O(d^2)$ sample complexity. Improving the sample complexity to the information theoretic threshold is an interesting direction of future work.
>
> > “...there is a big question as to whether or not the bounds are useful for understanding the success of deep learning systems even on simple problems (e.g. FCN on MNIST)....it seems that it is helpful to both learn in multiple layers, as well as over multiple timepoints.”
> - We certainly agree that there is more work to be done in understanding the effectiveness of deep learning on realistic datasets. In more realistic settings, it is plausible that training all layers jointly and for multiple timesteps leads to improved results over the layerwise training procedure. We view our work as elucidating the capabilities of three-layer networks, in particular in comparison to two-layer networks. Specifically, our results provide a setting in which three-layer networks can efficiently learn a class of functions that cannot be efficiently learned using two-layer networks.
> - Furthermore, the recent work [43] shows that shallow MLPs trained via standard techniques on image datasets do learn features corresponding to the linear feature obtained via a single step of GD for two-layer networks. An interesting direction of future study would be to understand whether this phenomenon also holds for three-layer networks.
>
> > “What evidence is there that the difference between the 2 and 3 layer bounds will persist even as theoretical techniques/the bounds themselves are improved?”
> - We remark that our lower bound in Section 5 shows that no polynomially sized 2-layer network can express $f^*$ below an error threshold. This implies that no algorithm can learn $f^*$ with polynomially many samples / in poly time. Thus even if more refined bounds were proven for 2 layer neural networks, $f^*$ would still not be learnable and the separation would still exist.
>
> > “How do the 3 layer feature learning bounds differ from NTK learning”
> - The NTK is a kernel method, and thus the discussion in Section 3, point 2 also applies to the NTK. Since the NTK for any depth is a rotationally invariant kernel, the lower bound from [27] applies. Therefore, in the quadratic feature example in Section 4.2, $d^{2p}$ samples are needed to learn $g^*(x^TAx)$ for a degree $p$ polynomial $g^*$ via the NTK. We mention this point in line 305, but we will update the exposition to make clear that this lower bound applies to the NTK for a depth 3 network as well.
>
> > “Can the method be used to show anything about the usefulness of running gradient descent over a longer period of time?”
> - Understanding the sample complexity benefit of training $W$ for longer is indeed an interesting direction of future work. While initial works studying feature learning in two-layer neural networks such as [8, 18] relied on the single-step training procedure, later works [2] studied training $W$ for multiple steps and were able to improve the dimension dependence in the sample complexity. For three layer neural networks, it is possible that a more refined analysis of multi-step training could improve the sample complexity even further. However, analyzing the training dynamics of $W$ for multiple steps introduces a number of new technical challenges which will likely require developing new techniques. As such, we defer this to future work.
>
> > “Is there any intuition about good choices of the function q in the bounds”
> - For the general theorem, if $f^*$ possesses the hierarchical structure that $f^* = g^* \circ h^*$ where $h^*$ is the learned feature, then one should choose $q = g^*$ in the main theorem. In the examples in Section 4, we see that a good choice is $q = g^*$; however, we don’t have a clean description of the optimal $q$ in the general setting.

---

> > ### Comment · Reviewer_8d6Z · 2023-08-15
> > **Response to authors**
> >
> > I thank the authors for their detailed responses. Between their comments here and their comments to the rest of the reviewers, my main concerns have been addressed and I will update my review score.

---

### Official Review · Reviewer_oYFv · 2023-07-24

**Soundness:** 3 good
**Presentation:** 3 good
**Contribution:** 2 fair
**Rating:** 6
**Confidence:** 2

**Summary:**

This paper analyzed the features learned by a three-layer network trained with layer-wise gradient descent as existing analyses are largely restricted to two-layer networks. It presented a general purpose theorem that upper bounds the sample complexity and width needed to achieve low test error when the target has a certain hierarchical structure.

**Strengths:**

-	This paper has shown that a three-layer network can learn nonlinear features, although existing work only has shown that two-layer neural networks learn only linear functions of input.
-	It’s good that the paper has an example section (Sec. 4.)


**Weaknesses:**

-	The “Hierarchical function” that this paper deals with is not early enough defined or introduced. It will make the problem that the paper solved more clear if that point is made clear in an earlier part of the paper.
-	This work still does not answer the same questions for convolutional networks.
-	This paper certainly showed that three-layer networks have provably richer feature learning capabilities than two-layer networks. Nonetheless, it is not guaranteed that the findings and proof can still be applied to any general “depth” as it has shown for the change from the two-layer to three-layer (although it might be high-likely.)
-	In other words, this issue is about generalizability on depth scalability.
-	This paper lacks discussion of its own limitations.


**Questions:**

Do the authors think this can be applied to convolutional networks and networks with more depth than three layers?

**Limitations:**

No limitations are discussed in the paper. It would have been nicer if the authors discussed the generalizability of the work to a deeper network.

---

> ### Author Rebuttal · Authors · 2023-08-07
>
> We thank the reviewer for their detailed and thoughtful review, and address specific comments below.
>
> > “It will make the problem that the paper solved more clear if that point is made clear in an earlier part of the paper.”
>
> - Thank you for this feedback; we will add more details about what specifically we mean by a hierarchical function to the introduction section in a future revision of our paper.
>
> > “This work still does not answer the same questions for convolutional networks.”
>
> - While understanding the optimization and generalization properties of convolutional networks (or other modern architectures such as transformers) is indeed an interesting question, it is beyond the scope of our current work. We remark that showing an end-to-end learning guarantee for a two-layer convolutional network beyond the kernel regime is still an open question. Prior guarantees for feature learning in fully connected networks have largely been restricted to two-layer networks, and our work makes progress on extending these results to three-layer networks.
>
> > “Nonetheless, it is not guaranteed that the findings and proof can still be applied to any general “depth””
>
> - We agree that it is also an interesting question to understand what kinds of hierarchical functions can be learned by networks of depth >3. We anticipate that a similar layerwise training procedure and analysis could allow us to show learnability of a class of hierarchical functions with deeper networks, but we leave investigation of this to future work.
>
> > “This paper lacks discussion of its own limitations.”
>
> - We included a limitation section within the supplementary material, but we can certainly move this discussion of limitations to the main text in a future revision of our paper.

---

> > ### Comment · Reviewer_oYFv · 2023-08-16
> >
> > Thanks for the responses. I maintain my rating.

---

### Decision · Program_Chairs · 2023-09-21

**Decision:**

Accept (spotlight)

**Comment:**

This paper analyzes feature learning in three-layer neural networks and proves that there are functions that are efficiently learnable with three layers but not two layers.

The reviewers acknowledge that these results improve the state-of-the-art in the theoretical understanding of feature learning, and generally  found the paper to be well written and clear. Some of the weaknesses brought forward in the discussion include limitations of layer-wise analysis, restriction to simple architectures, and the tightness of the bounds. These concerns were adequately addressed by the authors and overall the consensus is that this is an interesting paper that should be accepted to NeurIPS.